# Overlap-Adaptive Regularization for Conditional Average Treatment Effect Estimation

**Valentyn Melnychuk, Dennis Frauen, Jonas Schweisthal & Stefan Feuerriegel**
LMU Munich & Munich Center for Machine Learning
Munich, Germany
`melnychuk@lmu.de`

## Abstract

The conditional average treatment effect (CATE) is widely used in personalized medicine to inform therapeutic decisions. However, state-of-the-art methods for CATE estimation (so-called meta-learners) often perform poorly in the presence of low overlap. In this work, we introduce a new approach to tackle this issue and improve the performance of existing meta-learners in the low-overlap regions. Specifically, we introduce *Overlap-Adaptive Regularization* (OAR) that regularizes target models proportionally to overlap weights so that, informally, the regularization is higher in regions with low overlap. To the best of our knowledge, our OAR is the first approach to leverage overlap weights in the regularization terms of the meta-learners. Our OAR approach is flexible and works with any existing CATE meta-learner: we demonstrate how OAR can be applied to both parametric and non-parametric second-stage models. Furthermore, we propose debiased versions of our OAR that preserve the Neyman-orthogonality of existing meta-learners and thus ensure more robust inference. Through a series of (semi-)synthetic experiments, we demonstrate that our OAR significantly improves CATE estimation in low-overlap settings in comparison to constant regularization.

## 1 Introduction

Estimating the conditional average treatment effect (CATE) from observational data is a core challenge in causal machine learning (ML). Especially in medical applications, the CATE estimates help to guide personalized therapeutic decisions by predicting how different patients might respond to a given treatment (Feuerriegel et al., 2024).

State-of-the-art methods for CATE estimation are based on *two-stage Neyman-orthogonal meta-learners* (Curth & van der Schaar, 2021b; Morzywolek et al., 2023). As such, meta-learners have several practical benefits. Specifically, they are *model-agnostic* (Künzel et al., 2019) (i.e., they can be instantiated with arbitrary predictive models such as neural networks). Furthermore, by using Neyman-orthogonal risks (Chernozhukov et al., 2017; Foster & Syrgkanis, 2023), meta-learners can achieve favorable theoretical properties. In particular, the second-stage model becomes less sensitive to errors in the nuisance function estimates, which improves robustness.

However, the performance of meta-learners is constrained by the degree of *overlap* in the data (D'Amour et al., 2021; Matsouaka et al., 2024) – that is, the extent to which patients with similar covariates receive different treatments. In our work, overlap is represented as the product of the conditional probabilities of receiving each treatment, namely, *overlap weights*. Overlap is often violated in medicine when patients with certain covariate profiles almost exclusively receive one treatment (e.g., due to adherence to medical guidelines). Hence, the low-overlap regions of a covariate space are sparse in counterfactual outcomes, and, thus, learning CATE gets increasingly challenging.

To address issues from low overlap, existing meta-learners suggested two main approaches: (1) **retargeting** and (2) **constant regularization**. (1) **Retargeting** incorporates the overlap weights into *error terms of the target risks* (Morzywolek et al., 2023; Nie & Wager, 2021; Fisher, 2024), so that the error term is truncated or down-weighted in the low overlap regions. In contrast, (2) **constant regularization** aims to reduce CATE heterogeneity towards more averaged causal quantities (e. g., ATE). While effective to some extent, these strategies have key limitations (as we show later). In the case of (1) retargeting, the fitted target models struggle in low-overlap regions: they either (i) *have*

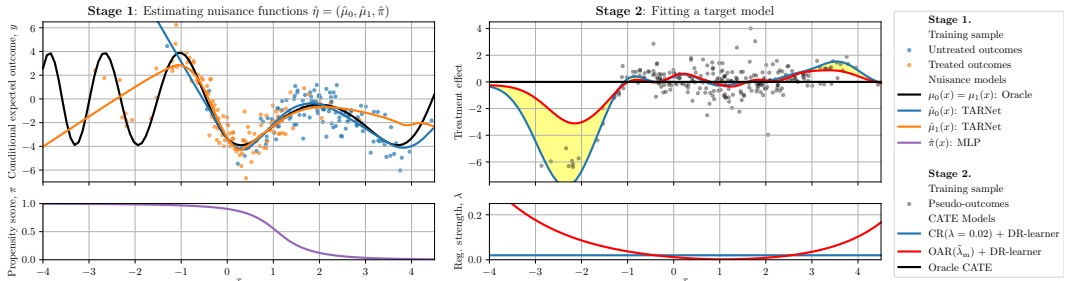

Figure 1: **Motivational example showing how our OAR (in red) performs better in low-overlap regions (in yellow).** Here, we used our OAR together with a DR-learner. We adapted the synthetic data generator from (Melnychuk et al., 2023) ($n_{\text{train}} = 250$; see Appendix F) and used kernel ridge regression (KRR) as a target model. We see that a target model fitted w/ our OAR($\tilde{\lambda}_{\text{m}}$) (shown in red) has a better performance in the low-overlap regions, compared to a target model w/ constant regularization (CR, shown in blue).

*unpredictable behavior* or (ii) *target at a different causal quantity* (e. g., R-/IVW-learners (Nie & Wager, 2021; Fisher, 2024) with the constant regularization yield a weighted average treatment effect [WATE] in the low-overlap regions). Further, (2) constant regularization does not take into account the degree of overlap and "blindly" regularizes all the regions of the covariate space.

In our work, we introduce ***Overlap-Adaptive Regularization*** (OAR), a novel approach that builds on top of existing two-stage meta-learners and tackles the low-overlap issue through adaptive regularization. Our OAR helps to prevent over- and underfitting of the target models by varying the amount of regularization depending on the degree of overlap (see the illustrative example in Fig. 1). As a result, our OAR applies stronger regularization in the regions with low overlap and weaker regularization where overlap is high (i. e., when the propensity scores are close to $0.5$).

Our OAR thus addresses the above limitations of existing meta-learners. First, (1) unlike targeting, our OAR makes the predictions of the OAR-fitted target models *smoother* in low-overlap regions (i. e., it enforces simpler models in those regions). Second, (2) unlike the constant regularization, it allows for more CATE *modeling flexibility* in the overlapping regions (e. g., for DR-learner). Also, unlike R-/IVW-learners, it can yield *the average treatment effect* (ATE) in the low-overlap regions (which is arguably a more meaningful causal quantity than the WATE). To the best of our knowledge, ours is the first approach to address the low-overlap problem by directly adapting the regularization term in the target risk.

Our OAR is flexible and can be applied together with any two-stage meta-learner. We provide several versions of our OAR approach: **(a)** for parametric target models (e.g., neural networks) and **(b)** for non-parametric target models (e.g., kernel ridge regression). For (a), we introduce two practical implementations via: (i) *OAR noise regularization* and (ii) *OAR dropout*. In addition, we propose a one-step bias-corrected (debiased) estimator of our OAR. This correction is important because it makes our OAR first-order insensitive to errors in the estimated overlap weights (which is especially relevant in observational studies where the ground-truth overlap weights are unknown). As a result, when combined with Neyman-orthogonal learners (e. g., DR-, R-, and IVW-learners), our debiased OAR preserves their Neyman-orthogonality. We further provide an extension of our OAR to (b) non-parametric target models (e.g., kernel ridge regression) in Appendix C.

In sum, our contributions are as follows:[1] **(1)** We introduce a novel approach, which we call *Overlap-Adaptive Regularization* (OAR), to address the performance of the existing CATE meta-learners in low-overlap regions. **(2)** We propose several versions of our OAR for both parametric and non-parametric target models, as well as a debiased version that preserves Neyman-orthogonality. **(3)** We show empirically that our OAR improves the performance in CATE estimation over other alternatives.

## 2 RELATED WORK

In the following, we briefly review the existing methods for CATE estimation and the ways they tackle the low-overlap issue. For a more extended overview of related work, we refer to Appendix A.

**Two-stage meta-learners.** State-of-the-art methods for CATE estimation can be broadly divided into two general categories: (a) plug-in learners (also known as model-based methods) and (b) (two-stage) meta-learners (Künzel et al., 2019; Curth & van der Schaar, 2021b; Morzywolek et al., 2023; Frauen

---

[1]Code is available at `https://github.com/Valentyn1997/OAR`.

et al., 2025). Here, we refer to the overview of (a) plug-in learners to Appendix A and rather focus on (b) meta-learners.[2] Meta-learners aim to find the best projection of CATE on a second-stage (target) model class and require the estimation of the nuisance functions at the first stage (Künzel et al., 2019). Importantly, they are *fully model-agnostic* meaning that any ML model can be employed at either of the stages. Also, they can possess many favorable theoretical properties if they are *Neyman-orthogonal* (Chernozhukov et al., 2017; Foster & Syrgkanis, 2023): notable examples of Neyman-orthogonal meta-learners include DR-learner (van der Laan, 2006; Curth et al., 2020; Kennedy, 2023), R-learner (Nie & Wager, 2021), and IVW-learner (Fisher, 2024). Neyman-orthogonality is a property of the target risks that makes it first-order insensitive to the errors of nuisance functions estimation, and, therefore, in this work, we only focus on those.

**How meta-learners deal with low overlap.** Low overlap poses a serious problem for any causal effect estimation (D'Amour et al., 2021; Matsouaka et al., 2024), including CATE. In the low-overlap regions, meta-learners mainly suffer from high variance of the pseudo-outcomes (Morzywolek et al., 2023): it stems either from a bad extrapolation of the first-stage models or from large inverse propensity scores. There are two general ways to tackle low overlap: (1) retargeting and (2) constant regularization.

**(1) Retargeting.** The retargeting approach estimates CATE only for a sub-population by modifying the target risk/loss (Morzywolek et al., 2023; Matsouaka et al., 2024). For example, trimming and truncation (Crump et al., 2009) discard too low propensity scores of the DR-learner loss; and overlap-weighting (Crump et al., 2006; Morzywolek et al., 2023) of the target risk yields either R-learner (Nie & Wager, 2021; Hess et al., 2026) or IVW-learner (Fisher, 2024). However, (1) retargeting on itself does not regulate how target models would generalize beyond the target sub-population. Therefore, the above-mentioned works suggested combining it with a (2) constant regularization.

**(2) Constant regularization.** Constant regularization improves low-overlap predictions by forcing lower CATE heterogeneity (Morzywolek et al., 2023) in *the whole covariate space*. Yet, this approach also has *drawbacks*. For example, when combined with the DR-learner, it does *not* distinguish the variability of pseudo-outcomes in high- and low-overlap regions. DR-learner, thus, can *overfit and underfit* at the same time due to the constant regularization. On the other hand, when combined with R-/IVW-learners, this approach *leads to a different causal quantity (i.e., WATE) when too much regularization is applied*.

**Adaptive regularization in traditional ML.** As discovered by Wager et al. (2013), dropout (Hinton et al., 2012; Srivastava et al., 2014) and noise regularization (Matsuoka, 1992; Bishop, 1995) can be seen as instances of the adaptive regularization. The authors have shown that for generalized linear models, dropout and noise regularization are first-order equivalent to the $l_2$ regularization applied to the features scaled with an inverse diagonal Fisher information matrix. This result was later extended for dropout regularization in NN-based models (Mou et al., 2018; Mianjy et al., 2018; Mianjy & Arora, 2019; Wei et al., 2020; Arora et al., 2021); for noise regularization in NN-based models (Rothfuss et al., 2019; Camuto et al., 2020); and for other types of regularization (Dieng et al., 2018; Mou et al., 2018; LeJeune et al., 2020; Zhang et al., 2021; Nguyen et al., 2021). In our paper, we also draw connections to the seminal results of (Wager et al., 2013). However, *we provide – for the first time – the connection of adaptive regularization to CATE estimation*. To the best of our knowledge, overlap weights have *not* been used to explicitly define regularization for CATE estimation.

## 3 PRELIMINARIES

**Notation.** Random variables are denoted by uppercase letters such as $Z$, their realizations by lowercase letters such as $z$, and their sample spaces by calligraphic symbols such as $\mathcal{Z}$. We write $\mathbb{P}(Z)$, $\mathbb{P}(Z = z)$, and $\mathbb{E}(Z)$ to refer, respectively, to a distribution of $Z$, its probability mass or density at $z$, and its expectation. We denote an $l_2$ norm as $\|x\|_2 = \sqrt{x_1^2 + \cdots + x_d^2}$ for $x \in \mathbb{R}^d$; a reproducing kernel Hilbert space (RKHS) norm as $\|f\|_{\mathcal{H}_K} = \sqrt{\langle f, f \rangle_{\mathcal{H}_K}}$ for $f \in \mathcal{H}_K$, where $\mathcal{H}_K$ is an RKHS induced by a kernel $K(\cdot, \cdot)$. We employ two nuisance functions: the *propensity score* for treatment $A$ is $\pi(x) = \mathbb{P}(A = 1 \mid X = x)$, and a *conditional expected outcome* for the response $Y$ is

---

[2] By naïvely estimating $\hat{\tau}(x) = \hat{\mu}_1(x) - \hat{\mu}_0(x)$, plug-in learners suffer from so-called plug-in bias (Kennedy, 2023) (e.g., $\hat{\mu}_0(x)$ is badly estimated for treated population). This is addressed in (two-stage) meta-learners. Unlike the plug-in learners, meta-learners allow to solve the bias-variance trade-off for the nuisance functions and the target CATE separately (Morzywolek et al., 2023). Hence, we focus on (two-stage) meta-learners throughout our paper.

$\mu_a(x) = \mathbb{E}(Y = y \mid X = x, A = a)$. We also consider a *marginalized conditional expected outcome* $\mu(x) = \mathbb{E}(Y = y \mid X = x)$ and *overlap weights* $\nu(x) = \text{Var}(A \mid X = x)$ as alternative nuisance function (yet, they can be expressed through the former two: $\mu(x) = (1 - \pi(x))\mu_0(x) + \pi(x)\mu_1(x)$ and $\nu(x) = \pi(x)(1 - \pi(x))$). Throughout, we work within the Neyman–Rubin potential outcomes framework (Rubin, 1974). Specifically, $Y[a]$ denotes the *potential outcome* under the intervention.

**Problem setup.** To estimate the CATE, we rely on an observational sample $\mathcal{D} = \{(x^{(i)}, a^{(i)}, y^{(i)})\}_{i=1}^n$, where $X \in \mathcal{X} \subseteq \mathbb{R}^{d_x}$ are high-dimensional covariates, $A \in \{0, 1\}$ is a binary treatment, and $Y \in \mathcal{Y} \subseteq \mathbb{R}$ is a continuous outcome. For example, in cancer care, $Y$ measures tumor growth, $A$ indicates whether chemotherapy is administered, and $X$ records patient attributes such as age and sex. Furthermore, we assume the $n$ triplets in $\mathcal{D}$ are drawn i.i.d. from the joint distribution $\mathbb{P}(X, A, Y)$. We also denote a correlation matrix of the covariates $X$ as $\Sigma = \mathbb{E}[XX^\top]$ and a $\lambda(\cdot)$-weighted correlation matrix as $\Sigma_\lambda = \mathbb{E}[\lambda(A, X)XX^\top]$ for a function $\lambda : \{0, 1\} \times \mathcal{X} \to \mathbb{R}$.

**Causal estimand and assumptions.** We are interested in estimating the *conditional average treatment effect (CATE)*: $\tau(x) = \mathbb{E}[Y[1] - Y[0] \mid X = x]$. To consistently estimate it from the observational data $\mathcal{D}$, we make the standard causal assumptions of the Neyman–Rubin framework (Rubin, 1974): (i) *consistency*: $Y[A] = Y$; (ii) *strong overlap*: $\mathbb{P}(\varepsilon < \pi(X) < 1 - \varepsilon) = 1$ for some $\varepsilon \in (0, 1/2)$; and (iii) *unconfoundedness*: $(Y[0], Y[1]) \perp\!\!\!\perp A \mid X$. Then, under the assumptions (i)–(iii), the CATE is identified from $\mathbb{P}(X, A, Y)$ as $\tau(x) = \mu_1(x) - \mu_0(x)$. In this work, we estimate the CATE from observational data $\mathcal{D}$ with meta-learners (Künzel et al., 2019; Morzywolek et al., 2023).

**Meta-learners for CATE.** Formally, two-stage meta-learners aim to find the best projection $g^*(x)$ of the ground-truth CATE $\tau(x)$ on a pre-specified model class $\mathcal{G} = \{g : \mathcal{X} \to \mathbb{R}\}$ by minimizing a target risk $\mathcal{L}(g, \eta)$ wrt. $g$. Here, $\eta = (\mu_0, \mu_1, \pi)$ are nuisance functions: they are fitted at the first stage and then used at the second stage to learn the optimal $g^* = \arg\min_{g \in \mathcal{G}} L(g, \eta)$. The majority of existing CATE meta-learners can be described by the following target risks, which have the same minimizers given the ground-truth nuisance functions:

$$\text{Original risk:} \quad \mathcal{L}(g, \eta) = \mathbb{E}\Big[w\big(\pi(X)\big)\big(\mu_1(X) - \mu_0(X) - g(X)\big)^2\Big] + \Lambda(g; \mathbb{P}(X)), \quad (1)$$

$$\text{Neyman-orthogonal risk:} \quad \mathcal{L}(g, \eta) = \underbrace{\mathbb{E}\Big[\rho\big(A, \pi(X)\big)\big(\phi(Z, \eta) - g(X)\big)^2\Big]}_{\text{error term } (\mathcal{E})} + \underbrace{\Lambda(g; \mathbb{P}(X))}_{\text{regularization term } (\Lambda)}, \quad (2)$$

where $w(\pi(X)) \geq 0$ is a weighting function, $\rho(A, \pi(X)) = (A - \pi(X))w'(\pi(X)) + w(\pi(X)) \geq 0$ is a debiased weighting function, $\phi(Z, \eta)$ is a pseudo-outcome with a property $\mathbb{E}[\phi(Z, \eta) \mid X = x] = \tau(x)$. While the two risks (original and Neyman-orthogonal) have the same minimizers $g^*$ given the ground-truth nuisance functions $\eta$, they yield significantly different results $\hat{g}$ when the nuisance functions are estimated $\hat{\eta}$.

**Neyman-orthogonal meta-learners.** In the following, we will focus on three Neyman-orthogonal meta-learners (Eq. (2)): DR-learner (Kennedy, 2023), R-learner (Nie & Wager, 2021), and IVW-learner (Fisher, 2024). The DR-learner (Kennedy, 2023) is given by $w(\pi(X)) = p(A, \pi(X)) = 1$ and $\phi(Z, \eta) = (A - \pi(X))(Y - \mu_A(X))/\nu(X) + \mu_1(X) - \mu_0(X)$; the R-learner (Nie & Wager, 2021) by $w(\pi(X)) = \nu(X), p(A, \pi(X)) = (A - \pi(X))^2$ and $\phi(Z, \eta) = (Y - \mu(X))/(A - \pi(X))$; and the IVW-learner (Fisher, 2024) by a combination of the former: the weighting functions of the R-learner and the pseudo-outcome of the DR-learner (see details on meta-learners in Appendix B).

**Low overlap.** We speak of low overlap, whenever either $\pi(x)$ or $(1 - \pi(x))$ (and thus $\nu(x)$) are close to 0. Conversely, perfect overlap regions have $\pi(x) = 0.5$ and $\nu(x) = 1/4$. Importantly, low overlap negatively affects the convergence of any meta-learner. For example, for the DR-learner, it inflates the inverse propensity scores and, for R-/IVW-learners, it retargets the target risk at a different quantity than CATE (Morzywolek et al., 2023).

**Constant regularization.**[3] The regularization term in Eq. (2), $\Lambda = \Lambda(g; \mathbb{P}(X))$, should be specified depending on the target model class $\mathcal{G}$. For example, if the second-stage model is (a) parametric, namely $\mathcal{G} = \{g(\cdot; \beta, c) : \mathcal{X} \to \mathbb{R} \mid \beta \in \mathbb{R}^d, c \in \mathbb{R}\}$, $l_2$-regularization is a popular choice: $\Lambda(g; \mathbb{P}(X)) = \lambda \|\beta\|_2^2$. Here, $c$ is an intercept, and $\lambda > 0$ is a regularization constant. Similarly, for a (b) non-parametric second-stage model (e. g., kernel ridge regression), $\mathcal{G}$ is the

---

[3]Here, the regularization term $\Lambda$ might also depend on $\mathbb{P}(X)$ (e. g., a standard dropout implicitly depends on the correlation matrix $\Sigma$). In our context, we call it constant regularization as it does not depend on the overlap.

RKHS $\mathcal{H}_{K+c}$ and $\Lambda(g; \mathbb{P}(X)) = \lambda \|g\|_{\mathcal{H}_K}^2$. Here (with a slight abuse of notation), $c$ is an added constant kernel. It is easy to see that, in both cases (a) and (b), increasing $\lambda \to \infty$ leads to $g^* \to \mathbb{E}[p(A, \pi(X))\phi(Z, \eta)]/\mathbb{E}[p(A, \pi(X))] = c^*$ (see Remark 1 in Appendix B.2). This happens as the intercept/constant kernel is not regularized. Yet, the constant regularization $\lambda$ (i) does not directly address the low-overlap issue for DR-learner or (ii) leads to WATE in low-overlap regions for R- and IVW-learners. This motivates our core idea of an adaptive regularization that depends on the distribution of the covariates and the treatment $\mathbb{P}(X, A)$, and the level of overlap $\nu(X)$.

## 4 OVERLAP-ADAPTIVE REGULARIZATION

In the following, we introduce our approach of *Overlap-Adaptive Regularization* (OAR) (Sec. 4.1) and several specific versions for (a) parametric target models (Sec. 5). Further, we provide an extension for non-parametric versions of our OAR in Appendix A. Proofs are provided in Appendix D.

### 4.1 GENERAL FRAMEWORK

Here, we define our *Overlap-Adaptive Regularization* (OAR), a novel general approach that (1) addresses the low-overlap issue of existing meta-learners, and (2) is model-agnostic.

---

**Definition 1** (Overlap-adaptive regularization (explicit form)). *For a meta-learner with a second-stage model $g(\cdot) \in \mathcal{G}$ and a target risk $\mathcal{L}(g, \eta) = \mathcal{E} + \Lambda$ (Eq. (2)), **overlap-adaptive regularization (OAR)** in an explicit form is given by*

$$\Lambda_{OAR} = \Lambda(g; \mathbb{P}(X, A); \lambda(\nu(X))), \tag{3}$$

*where $\lambda(\nu) > 0$ is a **regularization function** that defines the amount of the regularization and is proportional to the inverse overlap: $\lambda(\nu) \propto 1/\nu$. We further distinguish three general classes of regularization functions: multiplicative (m), logarithmic (log), and squared multiplicative (m²):*

$$\lambda_{\mathrm{m}}(\nu(x)) = 1/4\nu(x) - 1; \quad \lambda_{\log}(\nu(x)) = -\log(4\nu(x)); \quad \lambda_{\mathrm{m}^2}(\nu(x)) = 1/16\nu(x)^2 - 1. \tag{4}$$

---

Our OAR explicitly depends on the overlap through the regularization function[4], which is the main difference from the constant regularization.

**Interpretation.** Informally, *our OAR increases regularization in the regions of the covariate space $\mathcal{X}$ with low overlap* (namely, $\lambda(\nu) \to \infty$ when $\nu(x) \to 0$). Analogously, the regularization becomes smaller when perfect overlap is achieved (i. e., $\lambda(\nu) \to 0$ when $\nu(x) \to 1/4$). This introduces a desired behavior in a practical application. For example, in a medical context, low-overlap regions imply higher certainty about the treatment decisions (as optimal treatment might already be known there). Our OAR then allows us to focus the model flexibility on the overlapping sub-population (namely, the individuals for whom the CATE/optimal treatment is unknown).

**Implicit form.** Notably, our OAR can also be defined in the implicit form: it can enter through the error term of the target risk $\mathcal{E}_{\mathrm{OAR}}$ (e. g., noise regularization and dropout). Still, as we will demonstrate later, the two formulations are equivalent for linear models. That is, it is possible to find an equivalent form of the target risk with the original error term $\mathcal{E}$ and the regularization term in the explicit form $\Lambda_{\mathrm{OAR}}$. This equivalence can also be partially extended to some deep neural networks: noise regularization (Camuto et al., 2020) and dropout (Wei et al., 2020) were shown to have an explicit, first-order equivalent form.

**Flexibility.** Our OAR can be combined with any (Neyman-orthogonal) meta-learner. Also, we intentionally did not specify the dependency on $g$ and the observational distribution of $\mathbb{P}(X, A)$. This was done as we want our OAR to be *model-agnostic* and fit into a wide range of the second-stage model classes $\mathcal{G}$. For example, many standard regularization techniques like dropout and noise regularization can have very different dependencies on $g$ and $\mathbb{P}(X, A)$ in their explicit forms (Wager et al., 2013; Camuto et al., 2020; Mianjy & Arora, 2019; Wei et al., 2020).

### 4.2 DIFFERENCE FROM THE LITERATURE

**Difference to retargeting.** A natural question arises on whether our OAR is related to retargeting, a standard approach in meta-learners to handle low overlap (see Sec. 2). Specifically, retargeting is

---

[4] Our regularization function can be seen as an example of a selection (or tilting) function (Li et al., 2018; Assaad et al., 2021; Matsouaka et al., 2024). So far, these were only used in the error terms of the target risks (e. g., R-learner uses bias-corrected overlap weights).

Table 1: **Summary of OAR versions.** Here, $\mathcal{E}$ is the error term defined in Eq. (2) and $\lambda(\nu)$ is the regularization function of OAR (see Definition 1).

| Instantiation of OAR | Implicit form ($\mathcal{E}_{\text{OAR}}$) & explicit form ($\Lambda_{\text{OAR}}$) | Equivalent explicit form for a linear model class $\mathcal{G}$ |
|---|---|---|
| Noise regularization | $\mathcal{E}_{\text{OAR}} = \dots$ (Eq. (5))   $\Lambda_{\text{OAR}} = 0$ | $\mathcal{E}_{\text{OAR}} = \mathcal{E}$   $\Lambda_{\text{OAR}} = \|\beta\|_2^2 \, \mathbb{E}\left[\rho(A, \pi(X)) \cdot \lambda(\nu(X))\right]$ |
| Dropout | $\mathcal{E}_{\text{OAR}} = \dots$ (Eq. (7))   $\Lambda_{\text{OAR}} = 0$ | $\mathcal{E}_{\text{OAR}} = \mathcal{E}$   $\Lambda_{\text{OAR}} = \beta^\top \operatorname{diag}\left[\Sigma_{\rho(\cdot,\pi) \cdot \lambda(\nu)}\right]\beta$ |
| RKHS norm | $\mathcal{E}_{\text{OAR}} = \mathcal{E}$   $\Lambda_{\text{OAR}} = \left\|\sqrt{\lambda(\nu)}\, g\right\|_{\mathcal{H}_K}^2$ | Undefined |

implemented in both R- and IVW-learners, as they down-weight their error term of the target risk proportionally to overlap: $\mathbb{E}[\rho(A, \pi(X)) \mid X = x] = \mathbb{E}[(A - \pi(X))^2 \mid X = x] = \nu(x)$. On the other hand, our OAR up-weights the regularization term wrt. overlap. However, both approaches, in general, lead to *different risk minimizers*: while the re-weighted error term incorporates overlap with all the aspects of the observed distribution $\mathbb{P}(X, A, Y)$, our regularization term in OAR combines overlap only with $\mathbb{P}(X, A)$. Interestingly, the two approaches match only in a very simple case (as we will show later) when the propensity score and, thus, the overlap are constant.

**Difference to balancing.** Another way to tackle low overlap was suggested by balancing representations with empirical probability metrics (Johansson et al., 2016; Shalit et al., 2017; Johansson et al., 2022; Assaad et al., 2021) in neural network (NN)-based plug-ins (see Appendix A). Here, the average amount of regularization is proportional to a distributional distance between untreated and treated covariates, $\operatorname{dist}(\mathbb{P}(X \mid A = 0); \mathbb{P}(X \mid A = 1))$. In our case, the average amount of regularization can also be represented through distributional distances, yet *different from the one used in balancing regularization*.

**Proposition 1** (Average regularization function as a distributional distance). *The average amount of overlap-adaptive regularization $\mathbb{E}[\lambda(\nu(X))]$ is equal to or upper-bounded by $f$-divergences between $\mathbb{P}(X)$ and $\mathbb{P}(X \mid A = a)$ for $a \in \{0, 1\}$.*

We immediately see that our OAR is implicitly based on the distributional distances between $\mathbb{P}(X)$ and $\mathbb{P}(X \mid A = a)$, which are different from those used in balancing. Furthermore, our OAR is thus *simpler in implementation than balancing* because we only need to estimate the propensity score but not the distributional distance for a high-dimensional $X$.

## 5   INSTANTIATIONS OF OUR OAR

In the following, we provide several versions of our OAR for **parametric target models**, and we carefully tailor existing regularization techniques so that they become "overlap-adaptive" (see the overview in Table 1). For each version, we also (1) show an equivalent explicit form $\Lambda_{\text{OAR}}$ when the target model is linear and (2) derive a debiased (one-step bias-corrected) version of the regularization. The latter is beneficial to remove the first-order dependency on the estimated propensity score. We also provide a version for a **non-parametric target model** in Appendix C. All proofs are in Appendix D.1.

We consider target models in the following parametric form $\mathcal{G} = \{g(\cdot\,; \beta, c) : \mathcal{X} \to \mathbb{R} \mid \beta \in \mathbb{R}^d, c \in \mathbb{R}\}$, where $\beta$ are parameters to be regularized and $c$ is an intercept. For this very general parametric class, we tailor two general regularization techniques based on *noise injection*: (i) OAR noise regularization and (ii) OAR dropout. We also consider w.l.o.g. that noise is injected into the inputs of $g$ (if $g$ is an NN, then noise can be injected into any layer, see Fig. 3 in Appendix E).

### 5.1   OAR NOISE REGULARIZATION

Our **OAR with Gaussian noise regularization** is given by

$$\mathcal{L}_{\text{OAR}}^{+\xi}(g, \eta) = \mathcal{E}_{\text{OAR}} = \mathbb{E}\left[\mathbb{E}_{\xi \sim N(0, \sqrt{\lambda(\nu(X))}^2)}\left[\rho(A, \pi(X))\big(\phi(Z, \eta) - g(X + \xi)\big)^2\right]\right], \qquad (5)$$

where $N(0, \sigma^2)$ is a normal distribution with variance $\sigma^2$.

Thus, by construction of our OAR noise regularization, the variance of additive noise is proportional to the inverse overlap $\sigma^2 \propto 1/\nu(x)$, and the model $g(x)$ is regularized more in low-overlap regions. We further show an explicit form of OAR noise regularization $\Lambda_{\text{OAR}}$ for linear models $g$.

**Proposition 2** (Explicit form of OAR noise regularization in linear $g$). *For a linear model $g(x) = \beta^\top x + c$, OAR noise regularization has the following explicit form $\Lambda_{OAR}$:*

$$\mathcal{L}_{OAR}^{+\xi}(g, \eta) = \mathcal{E} + \Lambda_{OAR} = \mathcal{E} + \|\beta\|_2^2 \, \mathbb{E}\big[\rho(A, \pi(X)) \cdot \lambda(\nu(X))\big], \tag{6}$$

*where $\mathcal{E}$ is given by the original error term from Eq. (2).*

**Interpretation.** We observe that, for linear models, OAR noise regularization coincides with a ridge regression with the constant regularization $\lambda = \mathbb{E}[\rho(A, \pi(X)) \cdot \lambda(\nu(X))]$. However, for other, more complex parametric models, the explicit form is more complicated and is very different from $l_2$ (e. g., for NNs, noise regularization in explicit form depends on the Jacobians wrt. parameters (Camuto et al., 2020)).

## 5.2 OAR Dropout

> Our **OAR dropout** is given by:
>
> $$\mathcal{L}_{\text{OAR}}^{\circ\xi}(g, \eta) = \mathcal{E}_{\text{OAR}} = \mathbb{E}\Big[\mathbb{E}_{\xi \sim \text{Drop}(p(\nu(X)))}\big[\rho\big(A, \pi(X)\big)\big(\phi(Z, \eta) - g(X \circ \xi)\big)^2\big]\Big], \tag{7}$$
>
> where $\circ$ is an element-wise multiplication; $p(\nu) = \lambda(\nu)/(\lambda(\nu) + 1) \in (0, 1)$ is a dropout probability; $\xi \sim \text{Drop}(p)$ is sampled from a scaled Bernoulli distribution ($\xi = 0$ with probability $p$ and $\xi = 1/(1 - p)$ with probability $1 - p$).

Given the definition of our OAR dropout, it is easy to see that $p \propto 1/\nu(x)$. This means, the dropout probability is $p = 0$ in high-overlap regions ($\nu(x) = 1/4$), and $p \to 1$ in low-overlap regions ($\nu(x) \to 0$).

Interestingly, both the regular (Wager et al., 2013; Bartlett et al., 2019) and OAR dropouts are significantly different from the $l_2$-regularization, even for linear models $g$. We show it with the following proposition.

**Proposition 3** (Explicit form of OAR dropout in linear $g$). *For a linear model $g(x) = \beta^\top x + c$, OAR dropout regularization has the following explicit form $\Lambda_{OAR}$:*

$$\mathcal{L}_{OAR}^{\circ\xi}(g, \eta) = \mathcal{E} + \Lambda_{OAR} = \mathcal{E} + \beta^\top \text{diag}\big[\Sigma_{\rho(\cdot,\pi) \cdot \lambda(\nu)}\big]\beta, \tag{8}$$

*where $\mathcal{E}$ is given by the original error term from Eq. (2), $\lambda(\nu) = p(\nu)/(1 - p(\nu))$, and $\text{diag}[\cdot]$ zeroes out all but the diagonal entries of a matrix.*

**Interpretation.** Proposition 3 motivates our choice of $p(\nu)$ as $\lambda(\nu)/(\lambda(\nu) + 1)$ so that $\lambda(\nu) = p(\nu)/(1 - p(\nu))$. Also, we immediately see that, for linear models $g$, our OAR is *not* an $l_2$-regularization but an overlap-dependent quadratic form for $\beta$. That is, our OAR dropout scales each $\beta_j$ prior to applying $l_2$-regularization. Specifically, our OAR dropout in linear models is equivalent to a ridge regression where each feature is scaled down proportionately to the product of the inverse overlap and its own second moment: $\tilde{X}_j = X_j / \sqrt{\mathbb{E}[\rho(A, \pi(X)) \cdot \lambda(\nu(X)) \cdot X_j^2]}$. For other parametric models, the explicit form of OAR dropout becomes more complex (e. g., the explicit form of the standard dropout in NNs has $l_2$-path regularizers and rescaling invariant sub-regularizers (Mianjy & Arora, 2019; Wei et al., 2020)).

**Implicit and explicit forms.** Importantly, Propositions 2 and 3 also show the effect of OAR applied on top of the retargeted learners if OAR is presented in the implicit form $\mathcal{E}_{\text{OAR}}$. For example, when multiplicative OAR noise regularization is used with R-/IVW-learners, they result in a constant amount of regularization in low-overlap regions (i. e., $\mathbb{E}[\rho(A, \pi(X)) \cdot \lambda_{\text{m}}(\nu(X))] = 1/4 - \nu(X) \to 1/4$ given the ground-truth nuisance functions). This suggests that, if we want to adaptively regularize retargeted learners with OAR noise regularization, we need to employ the squared multiplicative regularization function $\lambda_{\text{m}^2}$ (so that $\mathbb{E}[\rho(A, \pi(X)) \cdot \lambda_{\text{m}^2}(\nu(X))] \to \infty$ in low-overlap regions).

## 5.3 Debiased OAR for parametric models

Here, we provide two debiased (one-step bias-corrected) versions, which we call **dOAR noise regularization** and **dOAR dropout**. Debiasing (van der Vaart, 2000; Kennedy, 2024) is beneficial to remove the first-order errors from the estimated propensity score $\hat{\nu}(x) = \hat{\pi}(x)(1 - \hat{\pi}(x))$. Namely, our original OAR from Eq. (5) and (7) might be overly sensitive to the misspecification of the overlap weights (e. g., when the propensity score $\hat{\pi}$ is badly estimated).

**Proposition 4** (Debiased OAR). *Under the continuous differentiability of $g(x; \beta, c)$, (i) debiased OAR noise regularization and (ii) debiased OAR dropout are given by*

$$\mathcal{L}_{dOAR}^{\diamond}(g, \eta) = \mathcal{L}_{OAR}^{\diamond}(g, \eta) + \mathbb{E}\left[\int_{\mathcal{X}} \mathbb{E}_{\xi}[C^{\diamond}(X; A; \xi; \nabla_{\xi}[g]; \eta)] \, \mathbb{P}(X = x) \, dx\right], \text{ for } \diamond \in \{+\xi, \circ\xi\}, \quad (9)$$

$$C^{+\xi}(X; A; \xi; \nabla_{\xi}[g]; \eta) = -2w(X)(\mu_1(X) - \mu_0(X) - g(X + \xi)) \cdot \nabla_{\xi}[g](X, \xi) \cdot \mathbb{IF}(\lambda(\nu(x)); X, A), \quad (10)$$

$$C^{\circ\xi}(X; A; \xi; \nabla_{\xi}[g]; \eta) = w(X)(\mu_1(X) - \mu_0(X) - g(X \circ \xi))^2 \cdot \frac{1 - \xi}{p(\nu(X))} \cdot \mathbb{IF}(p(\nu(x)); X, A)$$

$$- 2w(X)(\mu_1(X) - \mu_0(X) - g(X \circ \xi)) \cdot \nabla_{\xi}[g](X, \xi) \cdot \mathbb{IF}(p(\nu(x)); X, A), \quad (11)$$

*where $\mathcal{L}_{OAR}^{\diamond}$ are from Eq. (5) and (7); $\nabla_{\xi}[g]$ is a gradient wrt. $\xi$; and $\mathbb{IF}(\cdot; Z)$ is an efficient influence function (see Appendix D.2 for further details). Furthermore, by construction, $\mathcal{L}_{dOAR}^{\diamond}$ is a Neyman-orthogonal risk.*

*Proof.* For debiasing, we derived the efficient influence functions using a chain rule together with reparameterization and REINFORCE tricks. The full proof is in Appendix D. □

Importantly, after debiasing, our OAR recovers the property of Neyman-orthogonality (Chernozhukov et al., 2017; Foster & Syrgkanis, 2023) when combined with the standard Neyman-orthogonal learners. Furthermore, we can show that, under some additional conditions, our OAR/dOAR are guaranteed to outperform the constant regularization (CR).

**Proposition 5** (Excess prediction risk of our OAR/dOAR dropout with linear second-stage model). *The excess prediction risk of the DR-learner with the linear second-stage model and dropout regularization has the following form:*

$$||\hat{g} - g^*||_{L_2}^2 = \mathbb{E}[(\hat{\beta}^T X - \beta^{*T} X)^2] \lesssim \underbrace{\frac{1}{n} \text{tr}\left[\Sigma(\Sigma + \Gamma)^{-1} \Sigma_{\tilde{\phi}(Z, \eta)^2} (\Sigma + \Gamma)^{-1}\right]}_{\text{variance term}} + \underbrace{\beta^{*T} \Gamma \beta^*}_{\text{bias term}} + R(\eta, \hat{\eta}), \quad (12)$$

*where $\Gamma_{CR} = \lambda I$ for the CR, $\Gamma_{OAR} = \text{diag}\left[\Sigma_{\lambda(\nu)}\right]$ for the OAR/dOAR. Then, under (i) a conditional variance assumption (=conditional variance of the outcome is constant), the variance term for OAR/dOAR is less than or equal to the variance term of the CR. Also, under (ii) a low-overlap-low-heterogeneity inductive bias (LOLH-IB), OAR/dOAR do not increase the bias term too much.*

*Proof.* We used a bias-variance decomposition of the excess prediction risk for linear models. Then, we showed how assumptions (i)-(ii) help to reduce each term for our OAR/dOAR in comparison to the CR. The full proof is in Appendix D. □

We provide the full statement and the full proof of Proposition 5 in Appendix D. Arguably, both assumptions of Proposition 5 are reasonable: (i) The conditional variance of the outcome, $\text{Var}(Y \mid X, A)$, becomes nearly constant comparing to the variance of the DR pseudo-outcome; while (ii) LOLH-IB is often assumed to simplify causal ML (Curth & van der Schaar, 2021a; Melnychuk et al., 2026) (see Appendix A). Importantly, Proposition 5 and Eq. (12) apply to any level of overlap. However, specifically for the low-overlap setting (i.e., with larger values of $1/\nu(x)$), it is fair to assume (i) the conditional variance assumption, as the variance of the DR pseudo-outcome can be considered proportional to the inverse overlap.

## 5.4 NON-PARAMETRIC TARGET MODELS: OAR RKHS NORM

In the following, we introduce an instantiation of our OAR for a very general class of non-parametric models that belong to a reproducing kernel Hilbert space $\mathcal{G} = \mathcal{H}_{K+c}$ induced by a sum of an arbitrary kernel $K(\cdot, \cdot)$ and a constant kernel $K(\cdot, \cdot) = c$.

Our **OAR RKHS norm** for a target model $g \in \mathcal{H}_{K+c}$ can be instantiated as a weighted kernel ridge regression (KRR) with a modified, OAR-based RKHS norm:

$$\mathcal{L}_{OAR}^{\mathcal{H}}(g, \eta) = \mathcal{E} + \Lambda_{OAR} = \mathcal{E} + \left\|\sqrt{\lambda(\nu)}g\right\|_{\mathcal{H}_K}^2, \quad (13)$$

where $\mathcal{E}$ is given by the original error term from Eq. (2), and we assume that $\sqrt{\lambda(\nu)}g \in \mathcal{H}_K$ for every $g \in \mathcal{H}_{K+c}$ (this assumption is required so that the modified RKHS norm is well-defined).

Here, the regularization function $\sqrt{\lambda(\nu(x))}$ adaptively regularizes a function $g(x)$ and is known as a *multiplier of the RKHS $\mathcal{H}_K$* (Szafraniec, 2000; Paulsen & Raghupathi, 2016). Then, under special conditions, the weighted KRR with OAR-based RKHS norm has a well-defined solution.

**Proposition 6** (Kernel ridge regression with an OAR-based RKHS norm). *Let $\sqrt{\lambda(\nu)}g \in \mathcal{H}_K$ for every $g \in \mathcal{H}_{K+c}$. Then, the minimizer of the target risk $g^* = \arg\min_{g \in \mathcal{H}_{K+c}}[\mathcal{L}_{OAR}^{\mathcal{H}}(g, \eta)]$ is in $\mathcal{H}_{K+c}$ and has an explicit solution.*

We defer the full formulation of Proposition 6 and further discussions to Appendix C. Furthermore, Proposition 7 in Appendix C shows a result similar to Proposition 5 for our OAR RKHS norm regularization. Specifically, we showed there that our OAR RKHS norm improves over the CR under the analogous conditions for the RKHS: (i) conditional variance assumption and (ii) low-overlap-low-heterogeneity inductive bias.

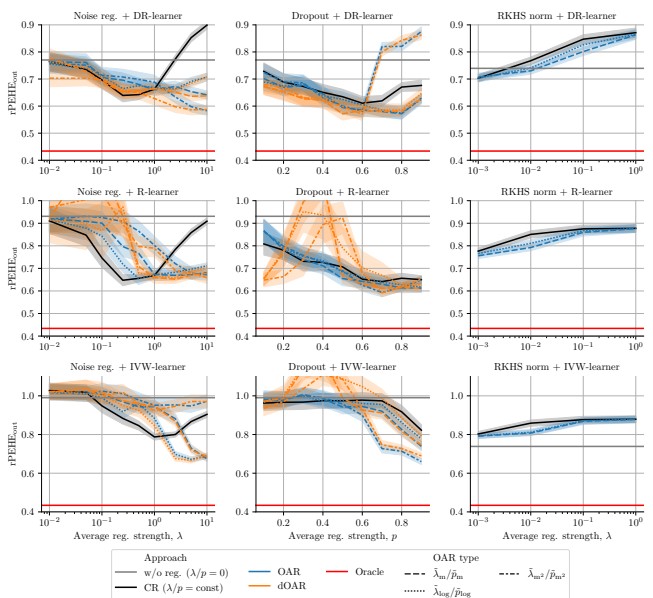

## 5.5 IMPLEMENTATION DETAILS

The implementation of our OAR proceeds in two stages (e. g., see Fig. 3 in Appendix E for the (a) parametric OAR instantiations with NNs). In stage 1, we estimate the nuisance functions $\hat{\eta}$. For this, we fit cross-validated fully-connected

Figure 2: **Results for IHDP dataset experiments.** Reported: median rPEHE$_{\text{out}}$ $\pm$ se over 100 runs.

NNs. Then, in stage 2, we fit a target network using empirical versions of the target risks $\hat{\mathcal{L}}$ in (a) and the KRR solution $\hat{g}$ in (b) to yield a target model (= CATE estimator). Furthermore, in both settings (a) and (b), we rescaled the regularization function $\tilde{\lambda}(\nu)$ (or $\tilde{p}(\nu)$) so that OAR can be comparable with the constant amount of regularization. For our dOAR, we additionally trimmed too large absolute values of the debiasing term $C^\diamond$: this helped to achieve a better stability of training the target models. We provide further implementation details in Appendix E.

## 6 EXPERIMENTS

**Setup.** We follow prior literature (Curth & van der Schaar, 2021b; Melnychuk et al., 2023) and use several (semi-)synthetic datasets where both counterfactual outcomes $Y[0]$ and $Y[1]$ and ground-truth CATE are available. Specifically, we used four datasets for benchmarking (see Appendix F for details). For all four, we report an out-sample root precision in estimating heterogeneous effect (rPEHE$_{\text{out}}$) or an improvement of our OAR over the baseline as a difference of the former ($\Delta$rPEHE$_{\text{out}}$).

**Baselines.** We compare all versions of our **OAR** and our debiased OAR (**dOAR**). As a baseline, we use only a comparable regularization strategy for meta-learners, namely, constant regularization (**CR**) (Morzywolek et al., 2023). In Appendix G, we also report the results of other, not directly comparable baselines (e. g., trimming and balancing). Here, we compare how different amounts of regularization work with three Neyman-orthogonal learners: (i) **DR-learner** (Kennedy, 2023), (ii) **R-learner** (Nie & Wager, 2021), and (iii) **IVW-learner** (Fisher, 2024). For a fair comparison, we rescaled our OAR/dOAR so that they, on average, coincide with the CR values (see Appendix E).

Table 2: **Results for 77 semi-synthetic ACIC 2016 experiments for the DR-learner**. Reported: % of datasets, where our OAR/dOAR significantly outperforms CR ($\alpha = 0.1$ with 15 runs per dataset).

| Reg. | Noise reg. | Dropout |
|---|---|---|
| $\lambda/p =$ Approach | 0.05 | 0.3 |
| OAR($\tilde{\lambda}_{\log}/\tilde{p}_{\log}$) | 14.29% | 29.87% |
| dOAR($\tilde{\lambda}_{\log}/\tilde{p}_{\log}$) | 7.79% | 64.94% |
| OAR($\tilde{\lambda}_{\mathrm{m}}/\tilde{p}_{\mathrm{m}}$) | 31.17% | 41.56% |
| dOAR($\tilde{\lambda}_{\mathrm{m}}/\tilde{p}_{\mathrm{m}}$) | 57.14% | 70.13% |
| OAR($\tilde{\lambda}_{\mathrm{m}^2}/\tilde{p}_{\mathrm{m}^2}$) | 27.27% | 16.88% |
| dOAR($\tilde{\lambda}_{\mathrm{m}^2}/\tilde{p}_{\mathrm{m}^2}$) | 76.62% | 64.94% |

Higher = better (improvement over the baseline in <50% of runs in green)

baselines (e. g., trimming and balancing). Here, we compare how different amounts of regularization work with three Neyman-orthogonal learners: (i) **DR-learner** (Kennedy, 2023), (ii) **R-learner** (Nie & Wager, 2021), and (iii) **IVW-learner** (Fisher, 2024). For a fair comparison, we rescaled our OAR/dOAR so that they, on average, coincide with the CR values (see Appendix E).

**IHDP dataset.** The IHDP dataset ($n = 672 + 75; d_x = 25$) (Hill, 2011; Shalit et al., 2017) is well-known to have severe overlap violations (Curth et al., 2021). Results. We show the results of the experiments with the IHDP dataset in Fig. 2. Therein, our OAR/dOAR are particularly effective for the DR-learner and large regularization values. Notably, the best performance for every meta-learner and regularization type is achieved by some version of our OAR/dOAR.

Table 3: **Results for HC-MNIST experiments for OAR/dOAR($\tilde{\lambda}_\mathrm{m}/\tilde{p}_\mathrm{m}$).** Reported: rPEHE$_\mathrm{out}$ ($\Delta$rPEHE$_\mathrm{out}$); mean $\pm$ std over 30 runs.

| Learner | Reg.
$\lambda/p =$ Approach | Noise reg.
0.05 | 0.1 | 0.25 | Dropout
0.1 | 0.3 | 0.5 |
|---|---|---|---|---|---|---|---|
| DR | CR ($\lambda/p$ = const) | 0.752 ± 0.038 | 0.741 ± 0.037 | 0.711 ± 0.030 | 0.746 ± 0.036 | 0.727 ± 0.032 | 0.711 ± 0.025 |
| | OAR($\tilde{\lambda}_\mathrm{m}/\tilde{p}_\mathrm{m}$) | 0.743 ± 0.039 (−0.009) | 0.726 ± 0.036 (−0.015) | 0.696 ± 0.033 (−0.015) | 0.742 ± 0.038 (−0.004) | 0.713 ± 0.032 (−0.014) | 0.701 ± 0.025 (−0.011) |
| | dOAR($\tilde{\lambda}_\mathrm{m}/\tilde{p}_\mathrm{m}$) | 0.731 ± 0.035 (−0.021) | 0.712 ± 0.033 (−0.029) | 0.684 ± 0.027 (−0.027) | 0.713 ± 0.038 (−0.033) | 0.705 ± 0.031 (−0.021) | 0.702 ± 0.026 (−0.009) |
| R | CR ($\lambda/p$ = const) | 0.715 ± 0.015 | 0.703 ± 0.010 | 0.674 ± 0.007 | 0.720 ± 0.027 | 0.711 ± 0.027 | 0.696 ± 0.018 |
| | OAR($\tilde{\lambda}_\mathrm{m}/\tilde{p}_\mathrm{m}$) | 0.711 ± 0.012 (−0.004) | 0.696 ± 0.009 (−0.007) | 0.673 ± 0.007 (-0.000) | 0.720 ± 0.024 (-0.000) | 0.696 ± 0.013 (−0.015) | 0.685 ± 0.010 (−0.011) |
| | dOAR($\tilde{\lambda}_\mathrm{m}/\tilde{p}_\mathrm{m}$) | **0.705 ± 0.009 (−0.010)** | **0.695 ± 0.010 (−0.007)** | **0.671 ± 0.008 (−0.003)** | **0.689 ± 0.015 (−0.031)** | **0.687 ± 0.013 (−0.024)** | **0.682 ± 0.011 (−0.013)** |
| IVW | CR ($\lambda/p$ = const) | 1.121 ± 0.246 | 1.102 ± 0.235 | 1.028 ± 0.201 | 1.136 ± 0.251 | 1.117 ± 0.259 | 1.113 ± 0.281 |
| | OAR($\tilde{\lambda}_\mathrm{m}/\tilde{p}_\mathrm{m}$) | 1.099 ± 0.237 (−0.021) | 1.071 ± 0.225 (−0.030) | 0.984 ± 0.215 (−0.044) | 1.131 ± 0.259 (−0.005) | 1.061 ± 0.231 (−0.056) | 0.997 ± 0.213 (−0.116) |
| | dOAR($\tilde{\lambda}_\mathrm{m}/\tilde{p}_\mathrm{m}$) | 1.105 ± 0.239 (−0.016) | 1.058 ± 0.217 (−0.044) | 0.978 ± 0.212 (−0.049) | 1.130 ± 0.221 (−0.006) | 1.110 ± 0.293 (−0.006) | 1.027 ± 0.235 (−0.086) |
| Oracle | | 0.513 | | | | | |

Lower = better (best in bold, second best underlined). Change over the baseline in brackets (significant improvement in green, significant worsening in red, $\alpha = 0.05$)

**ACIC 2016 datasets.** ACIC 2016 collection (Dorie et al., 2019) contains 77 semi-synthetic datasets ($n = 4802, d_x = 82$) with varying overlap and CATE heterogeneity. Due to the high-dimensionality of covariates, we exclude RKHS norm regularization from the experiments. Results. Results for the DR-learner are in Table 2, as our OAR/dOAR were most effective in the combination with the DR-learner. Here, different versions of our OAR/dOAR lead to a high percentage of significant improvements over the CR. Furthermore, our dOAR often leads to a significant improvement in more than half of the datasets.

**HC-MNIST dataset.** Finally, we adopted a high-dimensional HC-MNIST dataset ($d_x = 784 + 1$) (Jesson et al., 2021), which naturally suffers from low overlap (due to the dimensionality). Results. Table 3 provides the results for OAR/dOAR with the multiplicative regularization function (results for other regularization functions are in Appendix G). Here, we observe that our OAR/dOAR significantly improves the performance of the CR + DR-/R-/IVW-learners in the majority of cases. Notably, the *best performance for every regularization value is always achieved by some version of our OAR/dOAR*. This proves the effectiveness and scalability of our approach.

**Additional results.** In Appendix G, we additionally report the results for the synthetic data from Fig. 1. Also, we report the results of other, not directly comparable baselines (e. g., trimming and balancing). There, we vary regularization hyperparameters responsible for addressing low overlap.

**On the choice of the regularization function.** Our results support the *multiplicative regularization function* as the most effective choice for our OAR. First, the proof of our Proposition 5 in Appendix D.2 suggests that, under the conditional variance assumption, the variance-optimal shape for adaptive regularization scales as a fractional power of inverse overlap, namely $\lambda(\nu) \propto \nu^{-1/3}$. This lies between the logarithmic dependence (looser penalization) and the multiplicative dependence (stronger penalization), making the latter a practical, more robust variant of the regularization function. This choice of a stronger regularization function can also be particularly relevant for non-linear target models (e. g., neural networks), whose effective variance typically grows with model complexity. Second, Corollary C in Appendix 2 shows that our OAR based on the multiplicative regularization function combined with the DR-learner is equivalent to the CR combined with the R-learner in kernel ridge regression settings. Given the well-studied effectiveness of the R-learner, this offers theoretical support for multiplicative regularization. Finally, our extensive empirical evaluation in Sec. 6 and Appendix G consistently identifies the multiplicative regularization function as the strongest performer across diverse benchmarks. Collectively, these arguments justify our recommendation of the *multiplicative regularization function* as the default regularization strategy for our OAR.

**On the best combination.** We empirically found the combination of our OAR/dOAR noise regularization/dropout with DR-learner to be consistently good across *all the benchmarks*. The main reason for this is that our OAR/dOAR in combination with the DR-learner achieve just the right balance between the high-variability of the pseudo-outcome and the regularization strength (as suggested by the assumptions of Proposition 5). On the other hand, R- and IVW-learners in the combination with our OAR/dOAR might over-regularize the low-overlap areas, as the overlap already downscales the error term of the target loss.

**Conclusion.** In this paper, we introduced a novel approach for regularizing two-stage meta-learners: *Overlap-Adaptive Regularization*. Our OAR adaptively sets the regularization depending on the overlap so that low-overlap regions are regularized more. We showed that this approach is more effective than the existing constant regularization techniques. OAR performs best when low overlap coincides with low CATE heterogeneity (this can be seen as an underlying inductive bias). Such an inductive bias is often meaningful in practice: in the absence of ground-truth counterfactuals (i. e., in low-overlap versions), simpler models for the CATE may be preferred.

ETHICS STATEMENT

*Overlap-Adaptive Regularization* (OAR) is designed to make conditional average treatment effect (CATE) meta-learners more reliable when data exhibit poor treatment-control overlap. In high-stakes domains such as personalized medicine, where CATE estimates inform therapeutic choices, better behavior in low-overlap regions can translate into safer, more effective, and more equitable care decisions. Beyond healthcare, the technique may help policymakers or social-science researchers draw fairer conclusions from observational data by making CATE estimation more stable.

REPRODUCIBILITY STATEMENT

We have taken several measures to ensure the reproducibility of our work:

- *Algorithms.* We provide full algorithmic descriptions, including pseudocode for adaptive regularization methods. Hyperparameters, update rules, and initialization strategies are explicitly detailed.
- *Experimental validation.* We describe datasets, architectures, hyperparameters, and evaluation procedures in detail. We also released the code and experiment scripts to facilitate verification.
- *Resources.* The results from our paper can be fully reproduced using publicly available tools and the released supplementary materials.

Thus, all results can be independently verified based on the text and accompanying resources.

LLMs USAGE STATEMENT

We used ChatGPT during the preparation of this paper in a limited way, primarily for language editing. All statements were verified independently by the authors. ChatGPT was not used to generate new research ideas, algorithms, or experiments.

**Acknowledgments.** This paper is supported by the DAAD program "Konrad Zuse Schools of Excellence in Artificial Intelligence", sponsored by the Federal Ministry of Education and Research. S.F. acknowledges funding via Swiss National Science Foundation Grant 186932. This work has been supported by the German Federal Ministry of Education and Research (Grant: 01IS24082).

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

## A  EXTENDED RELATED WORK

In the following, we briefly discuss plug-in learners and why they are limited in comparison to (two-stage) meta-learners (and, thus, ***not*** relevant baselines for our work). Later, we discuss existing works on inductive biases for CATE estimation and how they relate to the existing strategies to tackle low overlap. Finally, we summarize the strategies to address low overlap for both plug-in and meta-learners in Table 4.

Table 4: Existing approaches for addressing low overlap in CATE estimation through regularization. Most relevant methods are highlighted in yellow.

| Approach | Underlying learner | Underlying IBs | | Effect of over-regularization | MA | NO | OART |
|---|---|---|---|---|---|---|---|
| Balancing (Johansson et al., 2016; Shalit et al., 2017; Hassanpour & Greiner, 2019; Johansson et al., 2022) | NNs (plug-in) | — | LOLH-IB | ✗ ($\hat{\tau} \to$ DiM) | ✗ | ✗ | ✓ |
| "Soft approach" (Curth & van der Schaar, 2021a) | NNs (plug-in) | S-IB | — | ✗ ($\hat{\tau} \to$ const) | ✗ | ✗ | ✗ |
| Const. reg. (Morzywolek et al., 2023) | X,U,RA,IPTW (Künzel et al., 2019; Curth & van der Schaar, 2021b) | S-IB | — | ✓ ($\hat{\tau} \to$ ATE) | ✓ | ✗ | ✗ |
| Const. reg. (Morzywolek et al., 2023) | DR (Kennedy, 2023) | S-IB | — | ✓ ($\hat{\tau} \to$ ATE) | ✓ | ✓ | ✗ |
| Const. reg. + Retargeting (Morzywolek et al., 2023) | R (Nie & Wager, 2021), IVW(Fisher, 2024) | S-IB | LOLH-IB | ⚠ ($\hat{\tau} \to$ WATE) | ✓ | ✓ | ✗ |
| OAR (**Our paper**) | DR (Kennedy, 2023) | S-IB | LOLH-IB | ✓ ($\hat{\tau} \to$ ATE) | ✓ | (✓) | ✓ |
| OAR (**Our paper**) + Retargeting | R (Nie & Wager, 2021), IVW(Fisher, 2024) | S-IB | LOLH-IB | ⚠ ($\hat{\tau} \to$ WATE) | ✓ | (✓) | ✓ |

Legend: model-agnostic (MA), Neyman-orthogonal (NO), overlap-adaptive regularization term (OART).

**Plug-in learners.** Plug-in learners *aim at the conditional expected outcomes* and yield the estimated CATE as the difference of the former. They can be either fully model-agnostic (e. g., S-/T-learner) or model-specific (e. g., causal forest). Specific instantiations of plug-in learners include random forest methods (Wager & Athey, 2018; Tibshirani et al., 2018; Athey et al., 2019; Athey & Wager, 2019), non-parametric kernel methods (Alaa & van der Schaar, 2017; 2018), and NN-based representation learning methods (Johansson et al., 2016; Shalit et al., 2017; Hassanpour & Greiner, 2019; Curth & van der Schaar, 2021b; Assaad et al., 2021; Johansson et al., 2022).

**How plug-in learners deal with low overlap.** In the low-overlap setting, plug-in estimators fail due to imprecise extrapolation wrt. counterfactual treatments (Jesson et al., 2021) (e. g., see Fig. 1, left). To tackle this, several regularization approaches have been proposed. For example, neural-based plug-in learners can employ (i) *balancing representations*[5] with empirical probability metrics (Johansson et al., 2016; Shalit et al., 2017; Johansson et al., 2022; Assaad et al., 2021; Melnychuk et al., 2026). Alternatively, one can use a (ii) *"soft approach"* of Curth & van der Schaar (2021a) which effectively forces the estimated conditional expected outcomes to be similar in low-overlap regions. Yet, both (i) and (ii) might have a detrimental effect on the estimated CATE when too much regularization is applied. For example, too much balancing leads to the estimation of a difference in means (DiM), also known as representation-induced confounding bias (Melnychuk et al., 2024; 2026); and the "soft approach" of Curth & van der Schaar (2021a) can force the estimated CATE to be constant. Therefore, we do *not* consider plug-in learners (and their regularization strategies) as relevant baselines.

**Addressing low overlap through model class choice.** An alternative to the regularization approach for addressing low overlap is a choice of a model class / NN architecture. This approach was primarily studied for plug-in learners as it is tailored to a specific model. For example, both estimated conditional expected outcomes can be forced to be similar, both (a) implicitly with an NN-based S-learner (= S-Net) (Curth & van der Schaar, 2021b) and (b) explicitly with neural architecture design

---

[5]Melnychuk et al. (2026) suggested a hypothetical way to incorporate balancing representations into a target model of meta-learners. In Sec. 4.2, we show that our instantiations of OAR are related to balancing of target models but, unlike balancing, are simpler to implement, as they do not require the evaluation of empirical distributional distances in the representation space.

as suggested in the "hard approach" of Curth & van der Schaar (2021a). Furthermore, the low-overlap issue is implicitly addressed in random forests with overlap-dependent depth (Wager & Athey, 2018; Tibshirani et al., 2018). However, in our paper, we focus on fully model-agnostic approaches to address low overlap that are based on *regularizing target models* and not on a model class choice.[6]

**Inductive biases for CATE estimation.** Regularization in ML is explicitly connected to inductive biases: increased regularization prioritizes simpler models. An inductive bias (IB) can be thus defined as any (non-causal) *assumption* a learning algorithm makes to generalize beyond the training data (Abu-Mostafa et al., 2012). In the context of CATE estimation, inductive biases are important due to the fundamental problem of causal inference (counterfactual outcomes are not observable, especially in low-overlap regions) and, thus, the impossibility of the exact data-driven model selection (Curth & van der Schaar, 2023). In the related work on CATE estimation, we outlined two main inductive biases: **smoothness inductive bias** (S-IB) and **low-overlap-low-heterogeneity inductive bias** (LOLH-IB). S-IB assumes that the ground-truth CATE is strictly simpler than both of the conditional expected outcomes (Curth & van der Schaar, 2021a; Morzywolek et al., 2023). By enforcing this inductive bias, we can improve low-overlap predictions by forcing lower CATE heterogeneity in *the whole covariate space*. LOLH-IB then extends S-IB further by assuming simpler models *specifically in low-overlap regions* (Melnychuk et al., 2026). In practice, both S-IB and LOLH-IB can be implemented in a model-agnostic fashion via regularization. We summarize the connections between different regularization approaches and the underlying inductive biases in Table 4.

---

[6]We acknowledge that this categorization is somewhat arbitrary, as some types of regularization might implicitly change the model class.

# B BACKGROUND MATERIALS

In the following, we provide background information on Neyman-orthogonality and two-stage meta-learners.

## B.1 NEYMAN-ORTHOGONALITY

We use the following additional notation: $a \lesssim b$ means there exists $C \geq 0$ such that $a \leq C \cdot b$, and $X_n = o_{\mathbb{P}}(r_n)$ means $X_n / r_n \xrightarrow{p} 0$.

**Definition 2** (Neyman-orthogonality (Chernozhukov et al., 2017; Foster & Syrgkanis, 2023; Morzywolek et al., 2023)). *A target risk $\mathcal{L}$ is called* Neyman-orthogonal *if its pathwise cross-derivative is zero:*

$$D_\eta D_g \mathcal{L}(g^*, \eta)[g - g^*, \hat{\eta} - \eta] = 0 \quad \text{for all } g \in \mathcal{G} \text{ and } \hat{\eta} \in \mathcal{H}, \tag{14}$$

*where $D_f F(f)[h] = \frac{d}{dt} F(f + th)|_{t=0}$ and $D_f^k F(f)[h_1, \ldots, h_k] = \frac{\partial^k}{\partial t_1 \ldots \partial t_k} F(f + t_1 h_1 + \cdots + t_k h_k)|_{t_1 = \cdots = t_k = 0}$ are pathwise derivatives (Foster & Syrgkanis, 2023); $g^* = \arg\min_{g \in \mathcal{G}} \mathcal{L}(g, \eta)$; and $\eta$ are the ground-truth nuisance functions.*

The definition of Neyman-orthogonality informally means that a target risk is first-order insensitive with respect to the misspecification of the nuisance functions.

## B.2 TWO-STAGE META-LEARNERS

To address the shortcomings of plug-in learners, two-stage meta-learners were proposed. These proceed in three steps as follows.

**(i)** First, one chooses a *target working model class* $\mathcal{G} = \{g(\cdot) : \mathcal{X} \to \mathbb{R}\}$ such as, for example, neural networks.

Then, **(ii)** the two-stage meta-learners define a specific *(original) target risk* for $g$. Several possible target risks can be selected, and each option bears distinct interpretations and ramifications for population and finite-sample two-stage CATE estimation. For example, one can use a regular MSE risk:

$$\mathcal{L}(g, \eta) = \mathbb{E}\left[\left(\mu_1(X) - \mu_0(X) - g(X)\right)^2\right] + \Lambda(g; \mathbb{P}(X)), \tag{15}$$

or an overlap-weighted MSE risk:

$$\mathcal{L}(g, \eta) = \mathbb{E}\left[\nu(X)\left(\mu_1(X) - \mu_0(X) - g(X)\right)^2\right] + \Lambda(g; \mathbb{P}(X)), \tag{16}$$

where $\Lambda(g; \mathbb{P}(X))$ is a constant regularization term with a magnitude $\lambda$. The latter (the overlap-weighted MSE risk) implements retargeting, but it *only focuses on the overlapping regions of the population*.

Finally, **(iii)** two-stage meta-learners minimize an empirical target risk (target loss) $\hat{\mathcal{L}}(g, \hat{\eta})$ estimated from the observational sample and using the first-stage nuisance estimates $\hat{\eta}$. When this empirical risk is built from semi-parametrically efficient estimators, the resulting method is known as a *Neyman-orthogonal learner* (Robins & Rotnitzky, 1995; Foster & Syrgkanis, 2023).

Notably, the constant regularization term does not have a detrimental effect on the CATE estimator. Notably, when too much regularization is applied, a non-regularized intercept of the target model yields ATE/WATE.

**Remark 1** (Over-regularized meta-learners (Morzywolek et al., 2023)). *Consider a target model class with a non-regularized intercept c. When $\lambda \to \infty$, the minimizer of Eq. (16) is given by WATE*

$$g^* = \arg\min_{g \in \mathcal{G}} \mathcal{L}(g, \eta) \to c^* = \frac{\mathbb{E}[\nu(X)(\mu_1(X) - \mu_0(X))]}{\mathbb{E}[\nu(X)]}. \tag{17}$$

## C  NON-PARAMETRIC TARGET MODELS: OAR RKHS NORM

**Proposition 6** (Kernel ridge regression with an OAR-based RKHS norm). *Let $\sqrt{\lambda(\nu)}g \in \mathcal{H}_K$ for every $g \in \mathcal{H}_{K+c}$. Then, the minimizer of the target risk $g^* = \arg\min_{g \in \mathcal{H}_{K+c}}[\mathcal{L}_{OAR}^{\mathcal{H}}(g, \eta)]$ is in $\mathcal{H}_{K+c}$ and has the following form:*

$$g^*(x) = (T_{\rho,K} + M_{\lambda(\nu)})^{-1} S_{\rho,K}(x) + c^*, \quad c^* = \mathbb{E}[p(A, \pi(X))\phi(Z, \eta)]/\mathbb{E}[p(A, \pi(X))], \quad (18)$$

*where $(T_{\rho,K}g)(x) = \mathbb{E}[\rho(A, \pi(X))K(x, X)g(X)]$ is a weighted covariance operator $(T_{\rho,K} : \mathcal{H}_K \to \mathcal{H}_K)$, $(S_{\rho,K})(x) = \mathbb{E}[\rho(A, \pi(X))K(x, X)\tilde{\phi}(Z, \eta)]$ is a weighted cross-covariance functional, $\tilde{\phi}(Z, \eta) = \phi(Z, \eta) - c^*$ is a centered pseudo-outcome, and $(M_{\lambda(\nu)}g)(x) = \lambda(\nu(x))g(x)$ is a bounded multiplication operator on $(M_{\lambda(\nu)} : \mathcal{H}_K \to \mathcal{H}_K)$.*

*Proof.*  See Appendix D.3.

Proposition 6 suggests that under the special conditions, our OAR-based RKHS norm yields a KRR solution with a varying regularization term $\lambda(\nu)$. For a finite-sample version of $g^*$, we refer to the following corollary.

**Corollary 1.**  *Consider that the assumptions (i)–(ii) of Proposition 6 hold and denote $\mathbf{K}_{XX} \in \mathbb{R}^{n \times n} = [K(x^{(i)}, x^{(j)})]_{i,j=1,\ldots,n}$; $\mathbf{K}_{xX} \in \mathbb{R}^{1 \times n} = [K(x, x^{(j)})]_{j=1,\ldots,n}$; $\mathbf{R}(\pi) \in \mathbb{R}^{n \times n} = [\rho(a^{(i)}, \pi(x^{(i)}))]_{i=1,\ldots,n} \circ \mathbf{I}_n$; $\mathbf{\Lambda}(\nu) \in \mathbb{R}^{n \times n} = [\lambda(\nu(x^{(i)}))]_{i=1,\ldots,n} \circ \mathbf{I}_n$; and $\mathbf{\Phi}(\eta) \in \mathbb{R}^{n \times 1} = [\phi(z^{(i)}, \eta)]_{i=1,\ldots,n}$. Then, a finite-sample KRR solution from Proposition 6 has the following form:*

$$\hat{g}(x) = \mathbf{K}_{xX} \left( \mathbf{R}(\hat{\pi}) \mathbf{K}_{XX} + n\mathbf{\Lambda}(\hat{\nu}) \right)^{-1} \mathbf{R}(\hat{\pi}) \mathbf{\Phi}(\hat{\eta}) + \hat{c}. \quad (19)$$

Also, Proposition 6 shows that, although our OAR-based RKHS-norm is generally undefined for a linear kernel, it works well for more flexible, infinite-dimensional kernels (e. g., RBF and Matérn). In practice, assumption (i) can be satisfied by either assuming a sufficiently smooth $\sqrt{\lambda(\nu)}$ (e. g., when the propensity score is smooth itself and bounded away from zero), or by approximating $\sqrt{\lambda(\nu)}$ with some element $\hat{g}$ from $\mathcal{H}_K$. This approximation can be done arbitrarily well with the infinite-dimensional kernels if they are dense in many smooth functional classes (e. g., RBF and Matérn are dense in a compact class of continuously differentiable functions).

Finally, in the following corollary, we show the connection between KRR with retargeted learners (R-/IVW-learners) and our OAR-based RKHS norm.

**Corollary 2.**  *A solution of (i) the KRR with constant RKHS norm regularization with $\lambda = 1$ for the original risks of the retargeted learners (R-/IVW-learners) coincides with a solution of (ii) the KRR with our OAR-based RKHS norm regularization with $\lambda(\nu(x)) = 1/\nu(x)$ for the original risk of the DR-learner, given the ground-truth nuisance functions $\eta$:*

$$\hat{g}(x) = \underbrace{\mathbf{K}_{xX} \left( \mathbf{W}(\pi) \mathbf{K}_{XX} + n\mathbf{I}_n \right)^{-1} \mathbf{W}(\pi) \mathbf{T}(\eta)}_{(i)} + \hat{c} = \underbrace{\mathbf{K}_{xX} \left( \mathbf{K}_{XX} + n\mathbf{\Lambda}(\nu) \right)^{-1} \mathbf{T}(\eta)}_{(ii)} + \hat{c}, \quad (20)$$

*where $\mathbf{W}(\pi) \in \mathbb{R}^{n \times n} = [\pi(x^{(i)})(1 - \pi(x^{(i)}))]_{i=1,\ldots,n} \circ \mathbf{I}_n$ and $\mathbf{T}(\eta) \in \mathbb{R}^{n \times 1} = [\mu_1(x^{(i)}) - \mu_0(x^{(i)})]_{i=1,\ldots,n}$.*

Corollary 2 thus hints that our OAR-based RKHS norm is equivalent to retargeting with the constant regularization only in a special (unnatural) case (i. e., when the ground-truth nuisance functions are known). That is, when $\mathbf{R}(\pi)/\mathbf{\Phi}(\eta)$ are used instead of $\mathbf{W}(\pi)/\mathbf{T}(\eta)$, the equality (i) = (ii) does not hold anymore.

Based on Corollary 2, we make another important observation for linear kernels $K(x, x') = x^\top x'$, namely that linear KRR can be formulated simultaneously as a parametric and a non-parametric model. Interestingly, while Corollary 2 still holds, the expression (ii) is, in general, *not* a solution to the OAR-based KRR. This happens, as the RKHS norm $\left\| \sqrt{\lambda(\nu)}g \right\|_{\mathcal{H}_K}^2$ is not defined for linear kernels when $\sqrt{\lambda(\nu)}$ is a non-linear function. Consequently, for linear target models, the approach

of retargeting *cannot*, in general (e. g., when the propensity score is not constant), be represented as a version of our general OAR (Sec. 4.1).

**Debiased OAR for RKHS norm**. Unlike OAR for parametric models, debiasing OAR-based RKHS norm is less intuitive. For example, the expected efficient influence function of the OAR-based RKHS norm cannot be expressed as an RKHS norm itself. This can be seen after applying a Mercer representation theorem (Theorem 4.51 in Steinwart & Christmann (2008) implies that $\left\|\sqrt{\lambda(\nu)}g\right\|_{\mathcal{H}_K}^2 = \mathbb{E}[\lambda(\nu(X))g(X)(T_K^{-1}g)(X)]$):

$$\mathbb{E}\Big[\mathbb{IF}\big(\left\|\sqrt{\lambda(\nu)}g\right\|_{\mathcal{H}_K}^2\big)\Big] = \mathbb{E}\Big[\mathbb{E}\big[\mathbb{IF}(\lambda(\nu);X,A)\big]g(X)(T_K^{-1}g)(X)\Big] \neq \left\|\sqrt{\mathbb{E}\big[\mathbb{IF}(\lambda(\nu);x,A)\big]}g\right\|_{\mathcal{H}_K}^2,$$
(21)

where the last equality does not hold as $\mathbb{E}\big[\mathbb{IF}(\lambda(\nu);x,A)\big]$ is not a proper RKHS multiplier (it depends on $A$ now). Therefore, we leave the debiasing of OAR-based RKHS norm for future work.

**Note on the squared multiplicative regularization.** Our main motivation for the introduction of the squared multiplicative regularization was to counteract the weights of the R- and IVW-learners when noise regularization and dropout are used. Specifically, noise regularization and dropout in their explicit form are down-scaled by $\rho$. In this way, the multiplicative regularization effectively results in a constant regularization, as $\mathbb{E}[\rho(A,\pi(X)),\lambda_m(\pi(X))] = 1$. Then, to preserve the overlap-adaptivity, we introduced the squared multiplicative regularization. The nonparametric models (namely, kernel ridge regressions), on the other hand, have their regularizations in explicit form and without a scaler $\rho$. Therefore, squared multiplicative regularization is not used with the RKHS norm.

**Excess prediction risk.** Finally, we show that, under some additional conditions, our OAR RKHS norm is guaranteed to outperform the constant regularization (CR) (similarly to linear target models as described in Proposition 5).

**Proposition 7** (Excess prediction risk of our OAR RKHS norm). *Let $\sqrt{\lambda(\nu)}g \in \mathcal{H}_K$ for every $g \in \mathcal{H}_{K+c}$. Then, the excess prediction risk of the DR-learner with the RKHS second-stage model and RKHS norm regularization has the following form:*

$$||\hat{g} - g^*||_{L_2}^2 \lesssim \underbrace{\frac{1}{n}\operatorname{tr}\big[(T_K+\Gamma)^{-1}T_K(T_K+\Gamma)^{-1}T_{\tilde{\phi}(Z,\eta)^2,K}\big]}_{\text{variance term}} + \underbrace{\langle g^*, \Gamma g^*\rangle_{\mathcal{H}_K}}_{\text{bias term}} + R(\eta,\hat{\eta}), \quad (22)$$

*where $(T_K g)(x) = \mathbb{E}[K(x,X)g(X)]$ and $(T_{\tilde{\phi}(Z,\eta)^2,K}g)(x) = \mathbb{E}[\tilde{\phi}(Z,\eta)^2 K(x,X)g(X)]$ are (weighted) covariance operators $(T_K, T_{\tilde{\phi}(Z,\eta)^2,K} : \mathcal{H}_K \to \mathcal{H}_K)$; $(\Gamma_{CR}g)(x) = \lambda g(x)$ is a constant scaling operator for the CR; and $(\Gamma_{OAR}g)(x) = (M_{\lambda(\nu)}g)(x) = \lambda(\nu(x))g(x)$ is a bounded multiplication operator on $(M_{\lambda(\nu)} : \mathcal{H}_K \to \mathcal{H}_K)$ for the OAR. Then, under (i) a conditional variance assumption (=conditional variance of the outcome is constant), the variance term for OAR is less than or equal to the variance term of the CR. Also, under (ii) a low-overlap-low-heterogeneity inductive bias (LOLH-IB), OAR does not increase the bias term too much.*

*Proof.* See Appendix D.3.

We provide the full statement and the full proof of Proposition 7 in Appendix D.

# D THEORETICAL RESULTS

## D.1 GENERAL FRAMEWORK OF OAR

**Proposition 1** (Average regularization function as a distributional distance). *The average amount of overlap-adaptive regularization is upper-bounded by the following $f$-divergences:*

$$\mathbb{E}[\lambda_{\mathrm{m}}(\nu(X))] \leq C_{\mathrm{m}} \sqrt{D_{f_{\mathrm{m}}}\big(\mathbb{P}(X) \,\|\, \mathbb{P}(X \mid A = 0)\big) + 1} \sqrt{D_{f_{\mathrm{m}}}\big(\mathbb{P}(X) \,\|\, \mathbb{P}(X \mid A = 1)\big) + 1}, \qquad (23)$$

$$\text{with } C_{\mathrm{m}} = 1/(4\pi_0\pi_1) \quad \text{and} \quad f_{\mathrm{m}}(t) = 1/t^2 - 1,$$

$$\mathbb{E}[\lambda_{\log}(\nu(X))] = C_{\log} + \mathrm{KL}\big(\mathbb{P}(X) \,\|\, \mathbb{P}(X \mid A = 0)\big) + \mathrm{KL}\big(\mathbb{P}(X) \,\|\, \mathbb{P}(X \mid A = 1)\big), \qquad (24)$$

$$\text{with } C_{\log} = -\log(4\pi_0\pi_1),$$

$$\mathbb{E}[\lambda_{\mathrm{m}^2}(\nu(X))] \leq C_{\mathrm{m}^2} \sqrt{D_{f_{\mathrm{m}^2}}\big(\mathbb{P}(X) \,\|\, \mathbb{P}(X \mid A = 0)\big) + 1} \sqrt{D_{f_{\mathrm{m}^2}}\big(\mathbb{P}(X) \,\|\, \mathbb{P}(X \mid A = 1)\big) + 1}, \quad (25)$$

$$\text{with } C_{\mathrm{m}^2} = 1/(16\pi_0^2\pi_1^2) \quad \text{and} \quad f_{\mathrm{m}^2}(t) = 1/t^4 - 1,$$

*where $\pi_a = \mathbb{P}(A = a)$, $D_f$ is an $f$-divergence $D_f(\mathbb{P}_1 \,\|\, \mathbb{P}_2) = \int f(\mathbb{P}_1(X = x)/\mathbb{P}_2(X = x))\mathbb{P}_1(X = x)\,\mathrm{d}x$; and $\mathrm{KL}$ is a KL-divergence.*

*Proof.* By the definitions of the regularization functions (Eq. (4)), the following holds:

$$\mathbb{E}[\lambda_{\mathrm{m}}(\nu(X))] \leq \mathbb{E}\left[\frac{1}{4\,\mathbb{P}(A = 0 \mid X)\,\mathbb{P}(A = 1 \mid X)}\right] = \frac{1}{4\pi_0\pi_1} \mathbb{E}\left[\frac{(\mathbb{P}(X))^2}{\mathbb{P}(X \mid A = 0)\,\mathbb{P}(X \mid A = 1)}\right] \qquad (26)$$

$$\overset{(*)}{\leq} C_{\mathrm{m}} \sqrt{\mathbb{E}\left[\left(\frac{\mathbb{P}(X)}{\mathbb{P}(X \mid A = 0)}\right)^2\right]} \sqrt{\mathbb{E}\left[\left(\frac{\mathbb{P}(X)}{\mathbb{P}(X \mid A = 1)}\right)^2\right]} \qquad (27)$$

$$= C_{\mathrm{m}} \sqrt{\int_{\mathcal{X}} \left[\left(\frac{\mathbb{P}(X = x)}{\mathbb{P}(X = x \mid A = 0)}\right)^2 - 1\right]\mathbb{P}(X = x)\,\mathrm{d}x + 1} \qquad (28)$$

$$\cdot \sqrt{\int_{\mathcal{X}} \left[\left(\frac{\mathbb{P}(X = x)}{\mathbb{P}(X = x \mid A = 1)}\right)^2 - 1\right]\mathbb{P}(X = x)\,\mathrm{d}x + 1}$$

$$= C_{\mathrm{m}} \sqrt{D_{f_{\mathrm{m}}}\big(\mathbb{P}(X) \,\|\, \mathbb{P}(X \mid A = 0)\big) + 1} \sqrt{D_{f_{\mathrm{m}}}\big(\mathbb{P}(X) \,\|\, \mathbb{P}(X \mid A = 1)\big) + 1}, \qquad (29)$$

where $(*)$ holds due to a Cauchy–Schwarz inequality, $C_{\mathrm{m}} = 1/(4\pi_0\pi_1)$, and $f_{\mathrm{m}}(t) = 1/t^2 - 1$. Analogously, it is easy to see that

$$\mathbb{E}[\lambda_{\mathrm{m}^2}(\nu(X))] \leq C_{\mathrm{m}^2} \sqrt{D_{f_{\mathrm{m}^2}}\big(\mathbb{P}(X) \,\|\, \mathbb{P}(X \mid A = 0)\big) + 1} \sqrt{D_{f_{\mathrm{m}^2}}\big(\mathbb{P}(X) \,\|\, \mathbb{P}(X \mid A = 1)\big) + 1}, \quad (30)$$

where $C_{\mathrm{m}} = 1/(16\pi_0^2\pi_1^2)$, and $f_{\mathrm{m}}(t) = 1/t^4 - 1$.

Similarly, we can show that the average logarithmic regularization function equals

$$\mathbb{E}[\lambda_{\log}(\nu(X))] = -\log(4) - \mathbb{E}\big[\log\mathbb{P}(A = 0 \mid X)\big] - \mathbb{E}\big[\log\mathbb{P}(A = 1 \mid X)\big] \qquad (31)$$

$$= -\log(4\pi_0\pi_1) - \mathbb{E}\left[\log\frac{\mathbb{P}(A = 0 \mid X)}{\mathbb{P}(X)}\right] - \mathbb{E}\left[\log\frac{\mathbb{P}(A = 1 \mid X)}{\mathbb{P}(X)}\right] \qquad (32)$$

$$= C_{\log} + \mathrm{KL}\big(\mathbb{P}(X) \,\|\, \mathbb{P}(X \mid A = 0)\big) + \mathrm{KL}\big(\mathbb{P}(X) \,\|\, \mathbb{P}(X \mid A = 1)\big), \qquad (33)$$

where $C_{\log} = -\log(4\pi_0\pi_1)$. □

### D.2 PARAMETRIC TARGET MODELS: OAR NOISE REGULARIZATION & OAR DROPOUT

**Proposition 2** (Explicit form of OAR noise regularization in linear $g$). *For a linear model $g(x) = \beta^\top x + c$, OAR noise regularization has the following explicit form $\Lambda_{OAR}$:*

$$\mathcal{L}_{OAR}^{+\xi}(g, \eta) = \mathcal{E} + \Lambda_{OAR} = \mathcal{E} + \|\beta\|_2^2 \, \mathbb{E}\big[\rho(A, \pi(X)) \cdot \lambda(\nu(X))\big], \tag{34}$$

*where $\mathcal{E}$ is given by the original error term from Eq. (2).*

*Proof.* The implicit OAR noise regularization of a linear target model has the following form:

$$\mathcal{L}_{OAR}^{+\xi}(g, \eta) = \mathbb{E}\Big[\mathbb{E}_{\xi \sim N(0, \sqrt{\lambda(\nu(X))}^2)}\big[\rho\big(A, \pi(X)\big)\big(\phi(Z, \eta) - \beta^\top(X + \xi) - c\big)^2\big)\big]\Big] \tag{35}$$

$$= \underbrace{\mathbb{E}\Big[\mathbb{E}_{\xi \sim N(0, \sqrt{\lambda(\nu(X))}^2)}\big[\rho\big(A, \pi(X)\big)\big(\phi(Z, \eta) - \beta^\top X - c\big)^2\big)\big]\Big]}_{\mathcal{E}}$$

$$- 2\underbrace{\mathbb{E}\Big[\mathbb{E}_{\xi \sim N(0, \sqrt{\lambda(\nu(X))}^2)}\big[\rho\big(A, \pi(X)\big)\big(\phi(Z, \eta) - \beta^\top X - c\big)\beta^\top \xi\big]\Big]}_{=0} \tag{36}$$

$$+ \mathbb{E}\Big[\mathbb{E}_{\xi \sim N(0, \sqrt{\lambda(\nu(X))}^2)}\big[\rho\big(A, \pi(X)\big)\big((\beta^\top \xi)^2\big)\big]\Big]$$

$$= \mathcal{E} + \beta^\top \mathbb{E}\big[\mathbb{E}_{\xi \sim N(0, \sqrt{\lambda(\nu(X))}^2)}[\rho\big(A, \pi(X)\big)\xi\xi^\top]\big]\beta \tag{37}$$

$$= \mathcal{E} + \|\beta\|_2^2 \, \mathbb{E}[\rho(A, \pi(X)) \, \mathrm{Var}[\xi \mid X]], \tag{38}$$

where $\mathrm{Var}[\xi \mid X = x] = \lambda(\nu(x))$. □

**Proposition 3** (Explicit form of OAR dropout in linear $g$). *For a linear model $g(x) = \beta^\top x + c$, OAR noise regularization has the following explicit form $\Lambda_{OAR}$:*

$$\mathcal{L}_{OAR}^{\circ\xi}(g, \eta) = \mathcal{E} + \Lambda_{OAR} = \mathcal{E} + \beta^\top \mathrm{diag}\big[\Sigma_{\rho(\cdot, \pi) \cdot \lambda(\nu)}\big]\beta, \tag{39}$$

*where $\mathcal{E}$ is given by the original error term from Eq. (2), $\lambda(\nu) = p(\nu)/(1 - p(\nu))$, and $\mathrm{diag}[\cdot]$ zeroes out all but the diagonal entries of a matrix.*

*Proof.* The implicit OAR dropout of a linear target model has the following form:

$$\mathcal{L}_{OAR}^{\circ\xi}(g, \eta) = \mathbb{E}\Big[\mathbb{E}_{\xi \sim \mathrm{Drop}(p(\nu(X)))}\big[\rho\big(A, \pi(X)\big)\big(\phi(Z, \eta) - \beta^\top(X \circ \xi) - c\big)^2\big]\Big] \tag{40}$$

$$= \underbrace{\mathbb{E}\Big[\mathbb{E}_{\xi \sim \mathrm{Drop}(p(\nu(X)))}\big[\rho\big(A, \pi(X)\big)\big(\phi(Z, \eta) - \beta^\top X - c\big)^2\big]\Big]}_{\mathcal{E}}$$

$$- 2\underbrace{\mathbb{E}\Big[\mathbb{E}_{\xi \sim \mathrm{Drop}(p(\nu(X)))}\big[\rho\big(A, \pi(X)\big)\big(\phi(Z, \eta) - \beta^\top X - c\big)\big(\beta^\top(X \circ \xi) - \beta^\top X\big)\big]\Big]}_{=0} \tag{41}$$

$$+ \mathbb{E}\Big[\mathbb{E}_{\xi \sim \mathrm{Drop}(p(\nu(X)))}\big[\rho\big(A, \pi(X)\big)\big(\beta^\top(X \circ \xi) - \beta^\top X\big)^2\big]\Big]$$

$$\overset{(*)}{=} \mathcal{E} + \mathbb{E}\Big[\rho\big(A, \pi(X)\big) \mathrm{Var}\big[\beta^\top(X \circ \xi) \mid X\big]\Big] = \mathcal{E} + \mathbb{E}\Big[\rho\big(A, \pi(X)\big) \mathrm{Var}\big[\sum_{j=1}^{d_x} \beta_j X_j \xi_j \mid X\big]\Big] \tag{42}$$

$$= \mathcal{E} + \mathbb{E}\Big[\rho\big(A, \pi(X)\big) \sum_{j=1}^{d_x} \frac{p(\nu(X))}{1 - p(\nu(X))} \beta_j^2 X_j^2\Big] = \mathcal{E} + \mathbb{E}\Big[\rho\big(A, \pi(X)\big)\lambda(\nu(X)) \sum_{j=1}^{d_x} \beta_j^2 X_j^2\Big] \tag{43}$$

$$= \mathcal{E} + \beta^\top \mathrm{diag}\big[\Sigma_{\rho(\cdot, \pi) \cdot \lambda(\nu)}\big]\beta, \tag{44}$$

where the equality $(*)$ holds as $\mathbb{E}_{\xi \sim \mathrm{Drop}(p(\nu(X)))}[\beta^\top(X \circ \xi)] = \beta^\top X$. □

**Proposition 4** (Debiased OAR). *Assume that the parametric model $g(x; \beta, c)$ is continuously differentiable wrt. $x$. Then, (i) debiased OAR noise regularization and (ii) debiased OAR dropout are as*

*follows:*

$$\mathcal{L}_{dOAR}^{\diamond}(g,\eta) = \mathcal{L}_{OAR}^{\diamond}(g,\eta) + \mathbb{E}\left[\int_{\mathcal{X}} \mathbb{E}_{\xi}[C^{\diamond}(X; A; \xi; \nabla_{\xi}[g]; \eta)]\, \mathbb{P}(X = x)\, \mathrm{d}x\right], \; for \diamond \in \{+\xi, \circ\xi\}, \quad (45)$$

$$C^{+\xi}(X; A; \xi; \nabla_{\xi}[g]; \eta) = -2w(X)(\mu_1(X) - \mu_0(X) - g(X + \xi)) \cdot \nabla_{\xi}[g](X, \xi) \cdot \mathbb{IF}(\lambda(\nu(x)); X, A), \quad (46)$$

$$C^{\circ\xi}(X; A; \xi; \nabla_{\xi}[g]; \eta) = w(X)(\mu_1(X) - \mu_0(X) - g(X \circ \xi))^2 \cdot \frac{1 - \xi}{p(\nu(X))} \cdot \mathbb{IF}(p(\nu(x)); X, A)$$
$$- 2w(X)(\mu_1(X) - \mu_0(X) - g(X \circ \xi)) \cdot \nabla_{\xi}[g](X, \xi) \cdot \mathbb{IF}(p(\nu(x)); X, A), \quad (47)$$

*where $\mathcal{L}_{OAR}^{\diamond}$ are from Eq. (5) and (7); $\nabla_{\xi}[g]$ is a gradient wrt. $\xi$; and $\mathbb{IF}(\cdot; X, A)$ are efficient influence functions of the regularization functions. The latter are given as follows:*

$$\mathbb{IF}(\lambda_{\mathrm{m}}(\nu(x)); X, A) = \frac{\delta\{X - x\}}{\mathbb{P}(X = x)} \frac{(A - \pi(x))(2\pi(x) - 1)}{4\nu(x)^2}, \quad (48)$$

$$\mathbb{IF}(p_{\mathrm{m}}(\nu(x)); X, A) = \frac{\delta\{X - x\}}{\mathbb{P}(X = x)}\, 4\,(A - \pi(x))(2\pi(x) - 1), \quad (49)$$

$$\mathbb{IF}(\lambda_{\mathrm{log}}(\nu(x)); X, A) = \frac{\delta\{X - x\}}{\mathbb{P}(X = x)} \frac{(A - \pi(x))(2\pi(x) - 1)}{\nu(x)}, \quad (50)$$

$$\mathbb{IF}(p_{\mathrm{log}}(\nu(x)); X, A) = \frac{\delta\{X - x\}}{\mathbb{P}(X = x)} \frac{(A - \pi(x))(1 - 2\pi(x))}{(1 - \log(4\nu(x)))^2}, \quad (51)$$

$$\mathbb{IF}(\lambda_{\mathrm{m}^2}(\nu(x)); X, A) = \frac{\delta\{X - x\}}{\mathbb{P}(X = x)} \frac{(A - \pi(x))(2\pi(x) - 1)}{8\nu(x)^3}, \quad (52)$$

$$\mathbb{IF}(p_{\mathrm{m}^2}(\nu(x)); X, A) = \frac{\delta\{X - x\}}{\mathbb{P}(X = x)}\, 32\,\nu(x)\,(A - \pi(x))(2\pi(x) - 1), \quad (53)$$

*where $\delta\{\cdot\}$ is a Dirac delta function. Furthermore, by construction, $\mathcal{L}_{dOAR}^{\diamond}$ is a Neyman-orthogonal risk.*

*Proof.* We follow a standard technique for constructing Neyman-orthogonal risks (Foster & Syrgkanis, 2023) by using a one-step bias-correction with efficient influence functions (Kennedy, 2024; Luedtke, 2026):

$$\mathcal{L}_{\mathrm{d}}(g,\eta) = \mathcal{L}(g,\eta) + \mathbb{E}\big[\mathbb{IF}(\mathcal{L}(g,\eta); Z)\big], \quad (54)$$

where $\mathcal{L}(g,\eta)$ is an original target risk, $\mathcal{L}_{\mathrm{d}}(g,\eta)$ is a Neyman-orthogonal risk, and $\mathbb{IF}(\mathcal{L}(g,\eta); Z)$ is an efficient influence function of the original target risk.

To construct a debiased (one-step bias-corrected) version of our OAR, we consider the OAR applied on top of the original target risk from Eq. (2):

$$\mathcal{L}_{\mathrm{OAR}}^{\diamond}(g,\eta) = \mathbb{E}\Big[\mathbb{E}_{\xi}\big[w(\pi(X))(\mu_1(X) - \mu_0(X) - g(\tilde{X}_{\xi}))^2\big]\Big], \quad (55)$$

where $\xi \sim N(0, \sqrt{\lambda(\nu(X))}^2)$ or $\xi \sim \mathrm{Drop}(p(\nu(X)))$, and $\tilde{X}_{\xi} = X + \xi$ or $\tilde{X}_{\xi} = X \circ \xi$, correspondingly, depending on the OAR version. Then, the efficient influence function of $\mathcal{L}_{\mathrm{OAR}}(g,\eta)$ is as follows

$$\mathbb{IF}(\mathcal{L}_{\mathrm{OAR}}^{\diamond}(g,\eta); Z) = \int_{\mathcal{X}} \mathbb{IF}\Big(\mathbb{E}_{\xi}\big[w(\pi(X))(\mu_1(X) - \mu_0(X) - g(\tilde{X}_{\xi}))^2\big]\Big)\mathbb{P}(X = x)\, \mathrm{d}x \quad (56)$$
$$+ \mathbb{E}_{\xi}\big[w(\pi(X))(\mu_1(X) - \mu_0(X) - g(\tilde{X}_{\xi}))^2\big] - \mathcal{L}_{\mathrm{OAR}}^{\diamond}(g,\eta).$$

Therefore, per Eq. (54), the debiased version of our OAR has a following form:

$$\mathcal{L}_{\mathrm{dOAR}}^{\diamond}(g,\eta) = \mathcal{L}_{\mathrm{OAR}}^{\diamond}(g,\eta) + \mathbb{E}\left[\int_{\mathcal{X}} \mathbb{IF}\Big(w(\pi(X))\mathbb{E}_{\xi}\big[(\mu_1(X) - \mu_0(X) - g(\tilde{X}_{\xi}))^2\big]\Big)\mathbb{P}(X = x)\, \mathrm{d}x\right]. \quad (57)$$

The second term can then be found with a product rule:

$$\mathbb{E}\left[\int_{\mathcal{X}} \mathbb{IF}\left(w\big(\pi(X)\big)\mathbb{E}_\xi\left[\big(\mu_1(X) - \mu_0(X) - g(\tilde{X}_\xi)\big)^2\right]\right)\mathbb{P}(X = x)\,\mathrm{d}x\right] \tag{58}$$

$$=\mathbb{E}\left[\int_{\mathcal{X}} \mathbb{IF}\big(w\big(\pi(X)\big)\big)\mathbb{E}_\xi\left[\big(\mu_1(X) - \mu_0(X) - g(\tilde{X}_\xi)\big)^2\right]\mathbb{P}(X = x)\,\mathrm{d}x\right] \tag{59}$$

$$+ \mathbb{E}\left[\int_{\mathcal{X}} w\big(\pi(X)\big)\mathbb{IF}\left(\mathbb{E}_\xi\left[\big(\mu_1(X) - \mu_0(X) - g(\tilde{X}_\xi)\big)^2\right]\right)\mathbb{P}(X = x)\,\mathrm{d}x\right]$$

$$=\mathbb{E}\left[(A - \pi(X))\,w'\big(\pi(X)\big)\mathbb{E}_\xi\left[\big(\mu_1(X) - \mu_0(X) - g(\tilde{X}_\xi)\big)^2\right]\right] \tag{60}$$

$$+ \mathbb{E}\left[\int_{\mathcal{X}} w\big(\pi(X)\big)\mathbb{IF}\left(\mathbb{E}_\xi\left[\big(\mu_1(X) - \mu_0(X) - g(\tilde{X}_\xi)\big)^2\right]\right)\mathbb{P}(X = x)\,\mathrm{d}x\right].$$

Hence, the debiased version of our OAR is

$$\mathcal{L}_{\text{dOAR}}^\diamond(g, \eta) = \mathbb{E}\left[\rho(A, \pi(X))\,\mathbb{E}_\xi\left[\big(\mu_1(X) - \mu_0(X) - g(\tilde{X}_\xi)\big)^2\right]\right] \tag{61}$$

$$+ \mathbb{E}\left[\int_{\mathcal{X}} w\big(\pi(X)\big)\mathbb{IF}\left(\mathbb{E}_\xi\left[\big(\mu_1(X) - \mu_0(X) - g(\tilde{X}_\xi)\big)^2\right]\right)\mathbb{P}(X = x)\,\mathrm{d}x\right],$$

where $\rho(A, \pi(X)) = (A - \pi(X))\,w'\big(\pi(X)\big) + w\big(\pi(X)\big)$. Now we focus on the second term of Eq. (61): it differs depending on the OAR version.

First, we consider **OAR noise regularization**. Let $\varepsilon \sim N(0, 1)$ and consider a reparametrization trick $\xi = \varepsilon \cdot \lambda(\nu(X))$, then

$$\mathbb{IF}\left(\mathbb{E}_\xi\left[\big(\mu_1(X) - \mu_0(X) - g(\tilde{X}_\xi)\big)^2\right]\right) = \mathbb{IF}\left(\mathbb{E}_\varepsilon\left[\big(\mu_1(X) - \mu_0(X) - g(X + \varepsilon \cdot \lambda(\nu(X)))\big)^2\right]\right) \tag{62}$$

$$= 2\mathbb{E}_\varepsilon\left[\big(\mu_1(X) - \mu_0(X) - g(X + \varepsilon \cdot \lambda(\nu(X)))\big)\mathbb{IF}\big(\mu_1(X) - \mu_0(X)\big)\right] \tag{63}$$

$$- 2\mathbb{E}_\varepsilon\left[\big(\mu_1(X) - \mu_0(X) - g(X + \varepsilon \cdot \lambda(\nu(X)))\big)\mathbb{IF}\big(g(X + \varepsilon \cdot \lambda(\nu(X)))\big)\right].$$

Given that $\mathbb{IF}\big(\mu_1(x) - \mu_0(x); Z\big) = \frac{\delta\{X - x\}}{\mathbb{P}(X = x)}\left(\frac{A - \pi(x)}{\nu(x)}\big(Y - \mu_A(x)\big)\right)$, the debiased version of our OAR becomes

$$\mathcal{L}_{\text{dOAR}}^{+\xi}(g, \eta) = \mathbb{E}\left[\rho(A, \pi(X))\,\mathbb{E}_\xi\left[\big(\mu_1(X) - \mu_0(X) - g(\tilde{X}_\xi)\big)^2\right]\right] \tag{64}$$

$$+ 2\mathbb{E}\left[\rho(A, \pi(X))\,\mathbb{E}_\xi\left[\frac{w(\pi(X))}{\rho(A, \pi(X))}\left(\mu_1(X) - \mu_0(X) - g(\tilde{X}_\xi)\right)\left(\frac{A - \pi(X)}{\nu(X)}\big(Y - \mu_A(X)\big)\right)\right]\right]$$

$$+ \mathbb{E}\left[\int_{\mathcal{X}} \mathbb{E}_\varepsilon\left[\underbrace{-2w\big(\pi(X)\big)\big(\mu_1(X) - \mu_0(X) - g(X + \varepsilon \cdot \lambda(\nu(X)))\big)\mathbb{IF}\big(g(X + \varepsilon \cdot \lambda(\nu(X)))\big)}_{C^{+\xi}(X; A; \xi; \nabla_\xi[g]; \eta)}\right]\mathbb{P}(X = x)\,\mathrm{d}x\right].$$

By completing a square, the latter target risk is equivalent in minimization to the following:

$$\mathcal{L}_{\text{dOAR}}^{+\xi}(g, \eta) = \mathbb{E}\left[\mathbb{E}_\xi\left[\rho(A, \pi(X))\underbrace{\left(\frac{w(\pi(X))}{\rho(A, \pi(X))}\left(\frac{A - \pi(X)}{\nu(X)}\big(Y - \mu_A(X)\big)\right) + \mu_1(X) - \mu_0(X) - g(\tilde{X}_\xi)\right)^2}_{\phi(Z, \eta)}\right]\right]$$

$$+ \mathbb{E}\left[\int_{\mathcal{X}} \mathbb{E}_\varepsilon\left[C^{+\xi}(X; A; \xi; \nabla_\xi[g]; \eta)\right]\mathbb{P}(X = x)\,\mathrm{d}x\right], \tag{65}$$

where $\phi(Z, \eta)$ is a pseudo-outcome (Morzywolek et al., 2023), and, thus, the first term recovers our OAR applied to a Neyman-orthogonal target risk from Eq. (5). Note that, to recover IVW-learner (Fisher, 2024), we need to set $\frac{w(\pi(X))}{\rho(A, \pi(X))} = 1$. This is a reasonable choice when $w(\pi) = \nu$ (as in the case of the IVW-learner) as $\mathbb{E}\left[\frac{w(\pi(X))}{\rho(A, \pi(X))} \mid x\right] = \mathbb{E}\left[\frac{\nu(X)}{(A - \pi(X))^2} \mid x\right] = 1$. Also, it is easy to see that, after this modification, the IVW-learner is still Neyman-orthogonal.

To derive $C^{+\xi}(X; A; \xi; \nabla_\xi[g]; \eta)$, we use chain rule for $\mathbb{IF}\big(g(X + \varepsilon \cdot \lambda(\nu(X)))\big)$:

$$\mathbb{IF}\big(g(X + \varepsilon \cdot \lambda(\nu(X)))\big) = g'(X + \varepsilon \cdot \lambda(\nu(X)))\, \varepsilon\, \mathbb{IF}\big(\lambda(\nu(X))\big) = \nabla_\xi[g](X, \xi)\, \mathbb{IF}\big(\lambda(\nu(X))\big), \quad (66)$$

where $\mathbb{IF}\big(\lambda(\nu(X))\big) = \mathbb{IF}\big(\lambda(\nu(x)); X, A\big)$ is the efficient influence function of the regularization function (will be derived later).

Similarly, we can derive a debiasing term for our **OAR dropout**. Let $\varepsilon \sim \mathrm{Bern}(1 - p(\nu(X)))$ so that $\xi = \varepsilon/(1 - p(\nu(X)))$ and consider a log-derivative trick:

$$\mathbb{IF}\Big(\mathbb{E}_\xi\big[\big(\mu_1(X) - \mu_0(X) - g(\tilde{X}_\xi)\big)^2\big]\Big) = \mathbb{IF}\Big(\mathbb{E}_\varepsilon\Big[\Big(\mu_1(X) - \mu_0(X) - g\Big(\frac{X \circ \varepsilon}{1 - p(\nu(X))}\Big)\Big)^2\Big]\Big) \quad (67)$$

$$= \int \mathbb{IF}\Big(\Big(\mu_1(X) - \mu_0(X) - g\Big(\frac{X \circ \varepsilon}{1 - p(\nu(X))}\Big)\Big)^2\Big)\mathbb{P}(\varepsilon)\,\mathrm{d}\varepsilon \quad (68)$$

$$+ \int \Big(\mu_1(X) - \mu_0(X) - g\Big(\frac{X \circ \varepsilon}{1 - p(\nu(X))}\Big)\Big)^2 \mathbb{IF}\Big(\varepsilon \log[1 - p(\nu(X))] + (1 - \varepsilon) \log[p(\nu(X))]\Big)\mathbb{P}(\varepsilon)\,\mathrm{d}\varepsilon$$

$$= 2\mathbb{E}_\xi\Big[\big(\mu_1(X) - \mu_0(X) - g(X \circ \xi)\big)\mathbb{IF}\big(\mu_1(X) - \mu_0(X)\big)\Big]$$

$$- 2\mathbb{E}_\varepsilon\Big[\Big(\mu_1(X) - \mu_0(X) - g\Big(\frac{X \circ \varepsilon}{1 - p(\nu(X))}\Big)\Big)\mathbb{IF}\Big(g\Big(\frac{X \circ \varepsilon}{1 - p(\nu(X))}\Big)\Big)\Big] \quad (69)$$

$$+ \mathbb{E}_\varepsilon\Big[\Big(\mu_1(X) - \mu_0(X) - g\Big(\frac{X \circ \varepsilon}{1 - p(\nu(X))}\Big)\Big)^2 \Big(\frac{\varepsilon}{p(\nu(X)) - 1} + \frac{1 - \varepsilon}{p(\nu(X))}\Big)\mathbb{IF}\big(p(\nu(X))\big)\Big]$$

$$= 2\mathbb{E}_\xi\Big[\big(\mu_1(X) - \mu_0(X) - g(X \circ \xi)\big)\mathbb{IF}\big(\mu_1(X) - \mu_0(X)\big)\Big]$$

$$- 2\mathbb{E}_\varepsilon\Big[\Big(\mu_1(X) - \mu_0(X) - g\Big(\frac{X \circ \varepsilon}{1 - p(\nu(X))}\Big)\Big)g'\Big(\frac{X \circ \varepsilon}{1 - p(\nu(X))}\Big)\frac{\mathbb{IF}\big(p(\nu(X))\big)}{(1 - p(\nu(X)))^2}\Big] \quad (70)$$

$$+ \mathbb{E}_\varepsilon\Big[\Big(\mu_1(X) - \mu_0(X) - g\Big(\frac{X \circ \varepsilon}{1 - p(\nu(X))}\Big)\Big)^2 \Big(\frac{\varepsilon}{p(\nu(X)) - 1} + \frac{1 - \varepsilon}{p(\nu(X))}\Big)\mathbb{IF}\big(p(\nu(X))\big)\Big]$$

$$= 2\mathbb{E}_\xi\Big[\big(\mu_1(X) - \mu_0(X) - g(X \circ \xi)\big)\mathbb{IF}\big(\mu_1(X) - \mu_0(X)\big)\Big]$$

$$- 2\mathbb{E}_\xi\Big[\big(\mu_1(X) - \mu_0(X) - g(X \circ \xi)\big)\nabla_\xi[g](X, \xi)\,\mathbb{IF}\big(p(\nu(X))\big)\Big] \quad (71)$$

$$+ \mathbb{E}_\xi\Big[\big(\mu_1(X) - \mu_0(X) - g(X \circ \xi)\big)^2 \frac{1 - \xi}{p(\nu(X))}\mathbb{IF}\big(p(\nu(X))\big)\Big],$$

where $\mathbb{IF}\big(p(\nu(X))\big) = \mathbb{IF}\big(p(\nu(x)); X, A\big)$ is the efficient influence function of the regularization function (will be derived later). Now, we can complete the square, similarly to Eq. (65), which yields the following debiased target risk:

$$\mathcal{L}^{\circ\xi}_{\mathrm{dOAR}}(g, \eta) = \mathbb{E}\Big[\mathbb{E}_\xi\Big[\rho(A, \pi(X))\underbrace{\Big(\frac{w(\pi(X))}{\rho(A, \pi(X))}\Big(\frac{A - \pi(X)}{\nu(X)}\big(Y - \mu_A(X)\big)\Big) + \mu_1(X) - \mu_0(X) - g(\tilde{X}_\xi)\Big)^2}_{\phi(Z, \eta)}\Big]\Big]$$

$$(72)$$

$$+ \mathbb{E}\Big[\int_{\mathcal{X}} \mathbb{E}_\varepsilon\Big[C^{\circ\xi}(X; A; \xi; \nabla_\xi[g]; \eta)\Big]\mathbb{P}(X = x)\,\mathrm{d}x\Big],$$

where the second term is

$$C^{\circ\xi}(X; A; \xi; \nabla_\xi[g]; \eta) = w(\pi(X))\big(\mu_1(X) - \mu_0(X) - g(X \circ \xi)\big)^2 \frac{1 - \xi}{p(\nu(X))}\mathbb{IF}\big(p(\nu(X))\big) \quad (73)$$

$$- 2w(\pi(X))\big(\mu_1(X) - \mu_0(X) - g(X \circ \xi)\big)\nabla_\xi[g](X, \xi)\,\mathbb{IF}\big(p(\nu(X))\big).$$

Finally, we aim to derive $\mathbb{IF}\big(\lambda(\nu(x)); X, A\big)$ and $\mathbb{IF}\big(p(\nu(x)); X, A\big)$. For the **multiplicative regularization function**, namely:

$$\lambda_{\mathrm{m}}(\nu(x)) = 1/(4\nu(x)) - 1 \quad \text{and} \quad p_{\mathrm{m}}(\nu(x)) = 1 - 4\nu(x), \quad (74)$$

the efficient influence functions are

$$\mathbb{IF}\big(1/(4\nu(x)) - 1; X, A\big) = -\frac{\mathbb{IF}(\nu(x))}{4\nu(x)^2} = \frac{\delta\{X - x\}}{\mathbb{P}(X = x)}\frac{(A - \pi(x))\,(2\pi(x) - 1)}{4\nu(x)^2}, \qquad (75)$$

$$\mathbb{IF}\big(1 - 4\nu(x); X, A\big) = \frac{\delta\{X - x\}}{\mathbb{P}(X = x)}\,4\,(A - \pi(x))\,(2\pi(x) - 1). \qquad (76)$$

For the **logarithmic regularization function**, namely:

$$\lambda_{\log}(\nu(x)) = -\log(4\nu(x)) \quad\text{and}\quad p_{\mathrm{m}}(\nu(x)) = 1 - \frac{1}{1 - \log(4\nu(x))}, \qquad (77)$$

the efficient influence functions are

$$\mathbb{IF}\big(-\log(4\nu(x)); X, A\big) = -\frac{\mathbb{IF}(\nu(x))}{\nu(x)} = \frac{\delta\{X - x\}}{\mathbb{P}(X = x)}\frac{(A - \pi(x))\,(2\pi(x) - 1)}{\nu(x)}, \qquad (78)$$

$$\mathbb{IF}\left(1 - \frac{1}{1 - \log(4\nu(x))}; X, A\right) = \frac{1}{(1 - \log(4\nu(x)))^2}\frac{\mathbb{IF}(\nu(x))}{\nu(x)} \qquad (79)$$

$$= \frac{\delta\{X - x\}}{\mathbb{P}(X = x)}\frac{(A - \pi(x))\,(1 - 2\pi(x))}{(1 - \log(4\nu(x)))^2}. \qquad (80)$$

Then, for the **squared multiplicative regularization function**, namely:

$$\lambda_{\mathrm{m}^2}(\nu(x)) = 1/16\nu(x)^2 - 1 \quad\text{and}\quad p_{\mathrm{m}^2}(\nu(x)) = 1 - 16\nu(x)^2, \qquad (81)$$

the efficient influence functions are

$$\mathbb{IF}\big(1/16\nu(x)^2 - 1; X, A\big) = -\frac{\mathbb{IF}(\nu(x))}{8\nu(x)^3} = \frac{\delta\{X - x\}}{\mathbb{P}(X = x)}\frac{(A - \pi(x))\,(2\pi(x) - 1)}{8\nu(x)^3}, \qquad (82)$$

$$\mathbb{IF}\big(1 - 16\nu(x)^2; X, A\big) = \frac{\delta\{X - x\}}{\mathbb{P}(X = x)}\,32\,\nu(x)\,(A - \pi(x))\,(2\pi(x) - 1). \qquad (83)$$

Notably, Dirac delta functions are later smoothed out with the integration $\int_{\mathcal{X}} \cdot \,\mathbb{P}(X = x)\,\mathrm{d}x$ in the final formula of $\mathcal{L}_{\mathrm{dOAR}}(g, \eta)$. Thus, they do not appear in the practical implementation. $\qquad\square$

**Proposition 5** (Excess prediction risk of our OAR/dOAR dropout with linear second-stage model). *Let $g^*(x) = \beta^{*T}x$ denote the best linear predictor for the second-stage risk with oracle nuisance functions ($\beta^* = \arg\min_\beta \mathbb{E}\big[(\phi(Z, \eta) - \beta^T X)^2\big]$); and let $\hat{g}(x) = \hat{\beta}^T x$ denote the finite-sample linear predictor based on CR/OAR/dOAR with the estimated nuisance functions ($\hat{\beta} = \arg\min_\beta \hat{\mathcal{L}}_\diamond^{\circ\xi}(g, \hat{\eta})$, $\diamond \in \{CR, OAR, dOAR\}$). Also, we consider the following reformulation of the DR pseudo-outcome $\phi(Z, \eta) = \beta^{*T}X + \tilde{\phi}(Z, \eta)$, where $\tilde{\phi}(Z, \eta)$ is a non-linear term with $\mathbb{E}[\tilde{\phi}(Z, \eta) \mid X] = 0$ (without the loss of generality).*

*Then, the excess prediction risk of the DR-learner with the linear second-stage model and dropout regularization has the following form:*

$$||\hat{g} - g^*||_{L_2}^2 = \mathbb{E}\big[(\hat{\beta}^T X - \beta^{*T} X)^2\big] \lesssim \underbrace{\frac{1}{n}\operatorname{tr}\big[\Sigma(\Sigma + \Gamma)^{-1}\Sigma_{\tilde{\phi}(Z,\eta)^2}(\Sigma + \Gamma)^{-1}\big]}_{variance\ term} + \underbrace{\beta^{*T}\Gamma\beta^*}_{bias\ term} + R(\eta, \hat{\eta}), \qquad (84)$$

*where $\Gamma_{CR} = \lambda I$ for the CR, $\Gamma_{OAR} = \operatorname{diag}\big[\Sigma_{\lambda(\nu)}\big]$ for the OAR/dOAR. Given this bias-variance decomposition, the following holds:*

- *For the CR and our dOAR, the remainder term $R(\eta, \hat{\eta})$ only contains higher-order errors of the nuisance functions (thus, the CR and our dOAR are less sensitive to the nuisance functions' misspecification). For example, the CR contains doubly-robust terms $||\hat{\mu}_a - \mu_a||_{L_4}^2||\hat{\pi} - \pi||_{L_4}^2$; our OAR contains doubly-robust terms $||\hat{\mu}_a - \mu_a||_{L_4}^2||\hat{\pi} - \pi||_{L_4}^2$ and a same-order propensity error $||\hat{\pi} - \pi||_{L_2}^2$; and our dOAR contains both doubly-robust terms $||\hat{\mu}_a - \mu_a||_{L_4}^2||\hat{\pi} - \pi||_{L_4}^2$ and a higher-order propensity error $||\hat{\pi} - \pi||_{L_4}^4$.*

- *Under a **conditional variance assumption** ($\mathrm{Var}[\tilde{\phi}(Z,\eta) \mid X] = \sigma^2/\nu(X)$, where $\sigma^2 = \mathrm{Var}[Y \mid X, A]$ is assumed to be constant), our OAR/dOAR reduces the **variance term** in comparison to the CR (given that OAR/dOAR is properly rescaled, i.e., $\mathbb{E}(\tilde{\lambda}(\nu(X))) = \lambda$, see Appendix E). That is,*

$$\mathrm{tr}\left[\Sigma(\Sigma + \Gamma_{OAR})^{-1}\Sigma_{\tilde{\phi}(Z,\eta)^2}(\Sigma + \Gamma_{OAR})^{-1}\right] \leq \mathrm{tr}\left[\Sigma(\Sigma + \Gamma_{CR})^{-1}\Sigma_{\tilde{\phi}(Z,\eta)^2}(\Sigma + \Gamma_{CR})^{-1}\right]. \tag{85}$$

- *Under a mild **low-overlap–low-heterogeneity (LOLH-lB)** condition, OAR/dOAR does not increase **the bias term** too much. This means that the terms $\beta^{*T}\Gamma_{OAR}\beta^*$ and $\beta^{*T}\Gamma_{CR}\beta^*$ only differ insignificantly. This is the case, as the **LOLH-lB** assumes small values for $\beta_j^*$ if some values of $X_j$ lead to the low overlap.*

*Proof.* Our proof follows in several steps.

**1. Bias term.** We start by defining an oracle regularized estimator, $\beta^\circ = \arg\min_\beta \mathcal{L}_\diamond^{\circ\xi}(g, \eta)$, that relates to the oracle unregularized estimator $\beta^*$ with a shrinkage error $b$:

$$b = \beta^\circ - \beta^* = -(\Sigma + \Gamma)^{-1}\Gamma\beta^*. \tag{86}$$

Then, the excess risk between $\beta^\circ$ and $\beta^*$ can be then upper-bounded by the bias term:

$$\|g^\circ - g^*\|_{L_2}^2 = \mathbb{E}\left[(\beta^{\circ T}X - \beta^{*T}X)^2\right] = b^T\Sigma b = \beta^{*T}\Gamma(\Sigma + \Gamma)^{-1}\Sigma(\Sigma + \Gamma)^{-1}\Gamma\beta^* \leq \beta^{*T}\Sigma\beta^*. \tag{87}$$

**2. Variance term.** The finite-sample linear predictor with the estimated nuisance is given by the following:

$$(\hat{\Sigma} + \hat{\Gamma})\hat{\beta} = \hat{c}(\hat{\eta}), \tag{88}$$

where $\hat{c}(\hat{\eta})$ is given by a finite-sample estimator of $c(\hat{\eta}) = \mathbb{E}[\phi(Z, \hat{\eta})X]$ (the formula for $\beta^\circ$ is analogous with $c(\eta)$). Then, the following holds asymptotically:

$$\hat{\beta} - \beta^\circ \approx (\Sigma + \Gamma)^{-1}(\hat{c}(\hat{\eta}) - c(\eta)). \tag{89}$$

Here, if $\hat{\beta}$ is based on the dOAR, an additional second-order remainder has to be added $R(\eta, \hat{\eta})$.

Also, the following holds due to the Neyman-orthogonality:

$$c(\hat{\eta}) - c(\eta) = \mathbb{E}[\tilde{\phi}(Z, \eta)X] + R(\eta, \hat{\eta}) \quad \text{and} \quad \mathrm{Cov}[\hat{c}(\hat{\eta}) - c(\hat{\eta})] = \frac{1}{n}\mathbb{E}[\tilde{\phi}(Z, \eta)^2 XX^T] + R(\eta, \hat{\eta}). \tag{90}$$

Finally, the excess risk between $\hat{\beta}$ and $\beta^\circ$ recovers our variance term:

$$\|\hat{g} - g^\circ\|_{L_2}^2 = \mathbb{E}\left[(\hat{\beta}^T X - \beta^{\circ T}X)^2\right] \approx \frac{1}{n}\mathrm{tr}\left[\Sigma(\Sigma + \Gamma)^{-1}\mathrm{Cov}[\hat{c}(\hat{\eta}) - c(\hat{\eta})](\Sigma + \Gamma)^{-1}\right] + R(\eta, \hat{\eta}) \tag{91}$$

$$= \frac{1}{n}\mathrm{tr}\left[\Sigma(\Sigma + \Gamma)^{-1}\Sigma_{\tilde{\phi}(Z,\eta)^2}(\Sigma + \Gamma)^{-1}\right] + R(\eta, \hat{\eta}). \tag{92}$$

**3.** Now, we combine the bias and variance terms by decomposing $\hat{\beta} - \beta^* = \hat{\beta} - \beta^\circ + \beta^\circ - \beta^*$ and formulate the final excess risk:

$$\|\hat{g} - g^*\|_{L_2}^2 = \mathbb{E}\left[(\hat{\beta}^T X - \beta^{*T}X)^2\right] \lesssim \frac{1}{n}\mathrm{tr}\left[\Sigma(\Sigma + \Gamma)^{-1}\Sigma_{\tilde{\phi}(Z,\eta)^2}(\Sigma + \Gamma)^{-1}\right] + \beta^{*T}\Gamma\beta^* + R(\eta, \hat{\eta}). \tag{93}$$

**4.** To see why our OAR/dOAR improves the CR, under the conditional variance assumption, we show the following. Assuming the correlation matrix $\Sigma$ is diagonal and has $\mathrm{Var}(X_j) = 1$ (w.l.o.g.), the variance term has the following form:

$$V(\Gamma) = \frac{1}{n}\sum_{j=1}^{d_x}\frac{m_j}{(1 + s_j)^2} \tag{94}$$

where $m_j = \mathrm{diag}[\Sigma_{\tilde{\phi}(Z,\eta)^2}]_j = \mathrm{diag}[\Sigma_{\sigma^2/\nu}]_j = \mathbb{E}[\sigma^2/\nu(X) \cdot X_j^2]$ and $s_j = \mathrm{diag}\left[\Sigma_{\lambda(\nu)}\right]_j = \mathbb{E}[\lambda(\nu) \cdot X_j^2]$. If we then compare penalties with the same average strength ($\sum s_j = $ const), the minimal value of $V(\Gamma)$ is achieved when $s_j \propto m_j^{1/3} - 1$ (namely, a KKT solution). The latter can be achieved by choosing the regularization function $\lambda(\nu)$ as described in Definition 1 (e. g., multiplicative, logarithmic, or squared multiplicative). $\qquad\square$

### D.3 Non-parametric target models: OAR RKHS norm

**Proposition 6** (Kernel ridge regression with an OAR-based RKHS norm). *Let $\sqrt{\lambda(\nu)}g \in \mathcal{H}_K$ for every $g \in \mathcal{H}_{K+c}$. Then, the minimizer of the target risk $g^* = \arg\min_{g \in \mathcal{H}_{K+c}}[\mathcal{L}_{OAR}^{\mathcal{H}}(g, \eta)]$ is in $\mathcal{H}_{K+c}$ and has the following form:*

$$g^*(x) = (T_{\rho,K} + M_{\lambda(\nu)})^{-1} S_{\rho,K}(x) + c^*, \quad c^* = \mathbb{E}[p(A, \pi(X))\phi(Z, \eta)]/\mathbb{E}[p(A, \pi(X))], \quad (95)$$

*where $(T_{\rho,K}g)(x) = \mathbb{E}[\rho(A, \pi(X))K(x, X)g(X)]$ is a weighted covariance operator ($T_{\rho,K} : \mathcal{H}_K \to \mathcal{H}_K$), $(S_{\rho,K})(x) = \mathbb{E}[\rho(A, \pi(X))K(x, X)\tilde{\phi}(Z, \eta)]$ is a weighted cross-covariance functional, $\tilde{\phi}(Z, \eta) = \phi(Z, \eta) - c^*$ is a centered pseudo-outcome, and $(M_{\lambda(\nu)}g) = \lambda(\nu(x))g(x)$ is a bounded multiplication operator on ($M_{\lambda(\nu)} : \mathcal{H}_K \to \mathcal{H}_K$).*

*Proof.* The OAR-based KRR aims to minimize the following objective:

$$g^* = \underset{g \in \mathcal{H}_{K+c}}{\arg\min} \left[ \mathbb{E}\Big[\rho\big(A, \pi(X)\big)\big(\phi(Z, \eta) - g(X)\big)^2\Big] + \left\|\sqrt{\lambda(\nu)}g\right\|_{\mathcal{H}_K}^2 \right], \quad (96)$$

which is equivalent to the minimization of the following objective:

$$g^* = c^* + \underset{g \in \mathcal{H}_K}{\arg\min} \underbrace{\left[ \mathbb{E}\Big[\rho\big(A, \pi(X)\big)\big(\tilde{\phi}(Z, \eta) - g(X)\big)^2\Big] + \left\|\sqrt{\lambda(\nu)}g\right\|_{\mathcal{H}_K}^2 \right]}_{\tilde{\mathcal{L}}_{OAR}^{\mathcal{H}}(g, \eta)}, \quad (97)$$

$\tilde{\phi}(Z, \eta) = \phi(Z, \eta) - c^*$ is a centered pseudo-outcome.

Under the strong overlap assumption (ii), $\sqrt{\lambda(\nu)}$ is a bounded kernel multiplier and we can define two self-adjoint multiplication operators (Szafraniec, 2000; Paulsen & Raghupathi, 2016), $M_{\sqrt{\lambda(\nu)}}$ and $M_{\lambda(\nu)}$, that act from $\mathcal{H}_K$ onto $\mathcal{H}_K$:

$$(M_{\sqrt{\lambda(\nu)}}g) = \sqrt{\lambda(\nu(x))}g(x) \quad \text{and} \quad (M_{\lambda(\nu)}g) = \lambda(\nu(x))g(x). \quad (98)$$

These two operators have the following property:

$$\left\|\sqrt{\lambda(\nu)}g\right\|_{\mathcal{H}_K}^2 = \langle M_{\sqrt{\lambda(\nu)}}g, M_{\sqrt{\lambda(\nu)}}g\rangle_{\mathcal{H}_K} = \langle g, M_{\sqrt{\lambda(\nu)}}^* M_{\sqrt{\lambda(\nu)}}g\rangle_{\mathcal{H}_K} = \langle g, M_{\lambda(\nu)}g\rangle_{\mathcal{H}_K}, \quad (99)$$

where $M^*$ is an adjoint operator.

Then, to find a minimizer of the centered OAR-based KRR objective in Eq. (97), we take a path-wise (Gâteaux) derivative in an arbitrary direction $h \in \mathcal{H}_K$ (Zhang et al., 2023):

$$\mathcal{D}_g\tilde{\mathcal{L}}_{OAR}^{\mathcal{H}}(g, \eta)[h] = \frac{d}{dt}\left[\mathbb{E}\Big[\rho\big(A, \pi(X)\big)\big(\tilde{\phi}(Z, \eta) - g(X) - th(X)\big)^2\Big] + \langle g + th, M_{\lambda(\nu)}(g + th)\rangle_{\mathcal{H}_K}\right]\Bigg|_{t=0} \quad (100)$$

$$= -2\mathbb{E}\Big[\rho\big(A, \pi(X)\big)\big(\tilde{\phi}(Z, \eta) - g(X)\big)h(X)\Big] + \langle g, M_{\lambda(\nu)}h\rangle_{\mathcal{H}_K} + \langle h, M_{\lambda(\nu)}g\rangle_{\mathcal{H}_K} \quad (101)$$

$$= -2\mathbb{E}\Big[\rho\big(A, \pi(X)\big)\tilde{\phi}(Z, \eta)h(X)\Big] + 2\mathbb{E}\Big[\rho\big(A, \pi(X)\big)g(X)h(X)\Big] + 2\langle h, M_{\lambda(\nu)}g\rangle_{\mathcal{H}_K} \quad (102)$$

$$\overset{(*)}{=} -2\langle h, S_{\rho,K}\rangle_{\mathcal{H}_K} + 2\langle h, T_{\rho,K}g\rangle_{\mathcal{H}_K} + 2\langle h, M_{\lambda(\nu)}g\rangle_{\mathcal{H}_K} \quad (103)$$

$$= 2\langle h, (T_{\rho,K} + M_{\lambda(\nu)})g - S_{\rho,K}\rangle_{\mathcal{H}_K}, \quad (104)$$

where $(T_{\rho,K}g)(x) = \mathbb{E}[\rho(A, \pi(X))K(x, X)g(X)]$ is a weighted covariance operator ($T_{\rho,K} : \mathcal{H}_K \to \mathcal{H}_K$); $(S_{\rho,K})(x) = \mathbb{E}[\rho(A, \pi(X))K(x, X)\tilde{\phi}(Z, \eta)]$ is a weighted cross-covariance functional; and the equality $(*)$ holds due to a Mercer representation theorem (Theorem 4.51 in Steinwart & Christmann (2008)). Namely, the Mercer representation theorem connects $L_2$ dot product and the RKHS dot product: $\mathbb{E}[g(X)h(X)] = \langle h, T_K g\rangle_{\mathcal{H}_K}$.

Then, the centered OAR-based KRR objective in Eq. (97) is optimized when for every $h \in \mathcal{H}_K$:

$$\mathcal{D}_g\tilde{\mathcal{L}}_{OAR}^{\mathcal{H}}(\tilde{g}^*, \eta)[h] = 0 \quad \Longleftrightarrow \quad \tilde{g}^* = (T_{\rho,K} + M_{\lambda(\nu)})^{-1} S_{\rho,K}. \quad (105)$$

The latter then recovers the desired optimizer $g^* = \tilde{g}^* + c^*$. $\square$

**Corollary 1.** *Consider that the assumptions (i)-(ii) of Proposition 6 hold and denote* $\mathbf{K}_{XX} \in \mathbb{R}^{n \times n} = [K(x^{(i)}, x^{(j)})]_{i,j=1,\ldots,n}$; $\mathbf{K}_{xX} \in \mathbb{R}^{1 \times n} = [K(x, x^{(j)})]_{j=1,\ldots,n}$; $\mathbf{R}(\pi) \in \mathbb{R}^{n \times n} = [\rho(a^{(i)}, \pi(x^{(i)}))]_{i=1,\ldots,n} \circ \mathbf{I}_n$; $\mathbf{\Lambda}(\nu) \in \mathbb{R}^{n \times n} = [\lambda(\nu(x^{(i)}))]_{i=1,\ldots,n} \circ \mathbf{I}_n$; *and* $\mathbf{\Phi}(\eta) \in \mathbb{R}^{n \times 1} = [\phi(z^{(i)}, \eta)]_{i=1,\ldots,n}$. *Then, a finite-sample KRR solution from Proposition 6 has the following form:*

$$\hat{g}(x) = \mathbf{K}_{xX} \left( \mathbf{R}(\hat{\pi}) \mathbf{K}_{XX} + n \mathbf{\Lambda}(\hat{\nu}) \right)^{-1} \mathbf{R}(\hat{\pi}) \mathbf{\Phi}(\hat{\eta}) + \hat{c}. \tag{106}$$

*Proof.* The finite-sample KRR solution immediately follows from Proposition 6. Specifically, we use a plug-in estimator of the weighted-covariance operator $\hat{T}_{\rho,K} = \frac{1}{n} \mathbf{R}(\hat{\pi}) \mathbf{K}_{XX}$; a plug-in estimator of the weighted cross-covariance operator $\hat{S}_{\rho,K} = \frac{1}{n} \mathbf{R}(\hat{\pi}) \mathbf{\Phi}(\hat{\eta}) \mathbf{K}_{xX}$; and a plug-in estimator of the multiplication operator $\hat{M}_{\lambda(\hat{\nu})} = \mathbf{\Lambda}(\hat{\nu})$. Then, the finite-sample KRR solution is as follows:

$$\hat{g}(x) = \mathbf{K}_{xX} \left( \frac{1}{n} \mathbf{R}(\hat{\pi}) \mathbf{K}_{XX} + \mathbf{\Lambda}(\hat{\nu}) \right)^{-1} \frac{1}{n} \mathbf{R}(\hat{\pi}) \mathbf{\Phi}(\hat{\eta}) + \hat{c} \tag{107}$$

$$= \mathbf{K}_{xX} \left( \mathbf{R}(\hat{\pi}) \mathbf{K}_{XX} + n \mathbf{\Lambda}(\hat{\nu}) \right)^{-1} \mathbf{R}(\hat{\pi}) \mathbf{\Phi}(\hat{\eta}) + \hat{c}. \tag{108}$$

$\square$

**Corollary 2.** *A solution of (i) the KRR with constant RKHS norm regularization with* $\lambda = 1$ *for the original risks of the retargeted learners (R-/IVW-learners) coincides with a solution of (ii) the KRR with our OAR-based RKHS norm regularization with* $\lambda(\nu(x)) = 1/\nu(x)$ *for the original risk of the DR-learner, given the ground-truth nuisance functions* $\eta$:

$$\hat{g}(x) = \underbrace{\mathbf{K}_{xX} \left( \mathbf{W}(\pi) \mathbf{K}_{XX} + n \mathbf{I}_n \right)^{-1} \mathbf{W}(\pi) \mathbf{T}(\eta)}_{(i)} + \hat{c} = \underbrace{\mathbf{K}_{xX} \left( \mathbf{K}_{XX} + n \mathbf{\Lambda}(\nu) \right)^{-1} \mathbf{T}(\eta)}_{(ii)} + \hat{c}, \tag{109}$$

*where* $\mathbf{W}(\pi) \in \mathbb{R}^{n \times n} = [\pi(x^{(i)}) (1 - \pi(x^{(i)}))]_{i=1,\ldots,n} \circ \mathbf{I}_n$ *and* $\mathbf{T}(\eta) \in \mathbb{R}^{n \times 1} = [\mu_1(x^{(i)}) - \mu_0(x^{(i)})]_{i=1,\ldots,n}$.

*Proof.* Corollary 2 follows from Corollary 1 and a push-through identity:

$$(\mathbf{P}\mathbf{Q} + \mathbf{I})^{-1} \mathbf{P} = \mathbf{P}(\mathbf{Q}\mathbf{P} + \mathbf{I})^{-1}, \tag{110}$$

where $\mathbf{P}$ and $\mathbf{Q}$ are conformable matrices, and $\mathbf{I}$ is an identity matrix. Hence, by setting $\mathbf{P} = \frac{1}{n} \mathbf{W}(\pi)$ and $\mathbf{Q} = \mathbf{K}_{XX}$, the following holds:

$$\hat{g}(x) = \underbrace{\mathbf{K}_{xX} \left( \frac{1}{n} \mathbf{W}(\pi) \mathbf{K}_{XX} + \mathbf{I}_n \right)^{-1} \frac{1}{n} \mathbf{W}(\pi) \mathbf{T}(\eta)}_{(i)} + \hat{c} \tag{111}$$

$$= \mathbf{K}_{xX} \frac{1}{n} \mathbf{W}(\pi) \left( \mathbf{K}_{XX} \frac{1}{n} \mathbf{W}(\pi) + \mathbf{I}_n \right)^{-1} \mathbf{T}(\eta) + \hat{c} \tag{112}$$

$$= \mathbf{K}_{xX} \frac{1}{n} \mathbf{W}(\pi) \left( \left( \mathbf{K}_{XX} + n \mathbf{I}_n (\mathbf{W}(\pi))^{-1} \right) \frac{1}{n} \mathbf{W}(\pi) \right)^{-1} \mathbf{T}(\eta) + \hat{c} \tag{113}$$

$$= \underbrace{\mathbf{K}_{xX} \left( \mathbf{K}_{XX} + n \mathbf{\Lambda}(\nu) \right)^{-1} \mathbf{T}(\eta)}_{(ii)} + \hat{c}. \tag{114}$$

$\square$

**Proposition 7** (Excess prediction risk of our OAR RKHS norm). *Let* $g^*$ *denote the best RKHS predictor for the second-stage risk with oracle nuisance functions (* $g^* = \arg\min_{g \in \mathcal{H}_{K+c}} \mathbb{E}\left[ (\phi(Z, \eta) - g(X))^2 \right]$ *); and let* $\hat{g}$ *denote the finite-sample RKHS predictor based on CR/OAR with the estimated nuisance functions (* $\hat{g} = \arg\min_{g \in \mathcal{H}_{K+c}} \hat{\mathcal{L}}_\diamond^{\mathcal{H}}(g, \hat{\eta})$, $\diamond \in \{CR, OAR\}$ *). Also, we consider the following reformulation of the DR pseudo-outcome* $\phi(Z, \eta) = g^*(X) + \tilde{\phi}(Z, \eta)$, *where* $\tilde{\phi}(Z, \eta)$ *is a RKHS approximation error term with* $\mathbb{E}[\tilde{\phi}(Z, \eta) \mid X] = 0$. *We assume* $c^* = 0$ *w.l.o.g.*

*Then, the excess prediction risk of the DR-learner with the RKHS second-stage model and RKHS norm regularization has the following form:*

$$||\hat{g} - g^*||_{L_2}^2 \lesssim \underbrace{\frac{1}{n} \operatorname{tr}\left[(T_K + \Gamma)^{-1} T_K (T_K + \Gamma)^{-1} T_{\tilde{\phi}(Z,\eta)^2, K}\right]}_{\text{variance term}} + \underbrace{\langle g^*, \Gamma g^* \rangle_{\mathcal{H}_K}}_{\text{bias term}} + R(\eta, \hat{\eta}), \quad (115)$$

*where $(T_K g)(x) = \mathbb{E}[K(x,X)g(X)]$ and $(T_{\tilde{\phi}(Z,\eta)^2, K} g)(x) = \mathbb{E}[\tilde{\phi}(Z,\eta)^2 K(x,X)g(X)]$ are (weighted) covariance operators $(T_K, T_{\tilde{\phi}(Z,\eta)^2, K} : \mathcal{H}_K \to \mathcal{H}_K)$; $(\Gamma_{CR} g)(x) = \lambda g(x)$ is a constant scaling operator for the CR; and $(\Gamma_{OAR} g)(x) = (M_{\lambda(\nu)} g)(x) = \lambda(\nu(x))g(x)$ is a bounded multiplication operator on $(M_{\lambda(\nu)} : \mathcal{H}_K \to \mathcal{H}_K)$ for the OAR.*

*Given this bias-variance decomposition, the following holds:*

- *For the CR, the remainder term $R(\eta, \hat{\eta})$ only contains higher-order errors of the nuisance functions (thus, the CR is less sensitive to the nuisance functions' misspecification). Specifically, the CR contains doubly-robust terms $||\hat{\mu}_a - \mu_a||_{L_4}^2 ||\hat{\pi} - \pi||_{L_4}^2$. At the same time, our OAR contains doubly-robust terms $||\hat{\mu}_a - \mu_a||_{L_4}^2 ||\hat{\pi} - \pi||_{L_4}^2$ and a same-order propensity error $||\hat{\pi} - \pi||_{L_2}^2$.*

- *Under a **conditional variance assumption** ($\operatorname{Var}[\tilde{\phi}(Z,\eta) \mid X] = \sigma^2/\nu(X)$, where $\sigma^2 = \operatorname{Var}[Y \mid X, A]$ is assumed to be constant), our OAR/dOAR reduces the **variance term** in comparison to the CR (given that OAR/dOAR is properly rescaled, i.e., $\mathbb{E}(\tilde{\lambda}(\nu(X))) = \lambda$, see Appendix E). That is,*

$$\operatorname{tr}\left[(T_K + \Gamma_{OAR})^{-1} T_K (T_K + \Gamma_{OAR})^{-1} T_{\tilde{\phi}(Z,\eta)^2, K}\right] \leq \operatorname{tr}\left[(T_K + \Gamma_{CR})^{-1} T_K (T_K + \Gamma_{CR})^{-1} T_{\tilde{\phi}(Z,\eta)^2, K}\right]. \quad (116)$$

- *Under a mild **low-overlap–low-heterogeneity (LOLH-IB)** condition, OAR/dOAR does not increase **the bias term** too much. This means that the terms $\langle g^*, \Gamma_{OAR} g^* \rangle_{\mathcal{H}_K}$ and $\langle g^*, \Gamma_{CR} g^* \rangle_{\mathcal{H}_K}$ only differ insignificantly. This is the case, as the **LOLH-IB** assumes a small norm for $g^*$ in the low-overlap regions.*

*Proof.* Our proof proceeds in 4 steps, similarly to Proposition 5.

**1. Bias term.** We start by defining an oracle regularized estimator, $g^\circ = \arg\min_{g \in \mathcal{H}_{K+c}} \mathcal{L}_\diamond^{\mathcal{H}}(g, \eta)$. Then, the excess risk between $g^\circ$ and $g^*$ can be upper-bounded by the bias term:

$$||g^\circ - g^*||_{L_2}^2 = \mathbb{E}\left[(g^\circ(X) - g^*(X))^2\right] = ||g^\circ - \phi(\cdot, \eta)||_{L_2}^2 - ||g^* - \phi(\cdot, \eta)||_{L_2}^2 \quad (117)$$

$$= \mathcal{L}_\diamond^{\mathcal{H}}(g^\circ, \eta) - \langle g^\circ, \Gamma g^\circ \rangle_{\mathcal{H}_K} - \mathcal{L}_\diamond^{\mathcal{H}}(g^*, \eta) + \langle g^*, \Gamma g^* \rangle_{\mathcal{H}_K} \quad (118)$$

$$\overset{(*)}{\leq} -\langle g^\circ, \Gamma g^\circ \rangle_{\mathcal{H}_K} + \langle g^*, \Gamma g^* \rangle_{\mathcal{H}_K} \overset{(**)}{\leq} \langle g^*, \Gamma g^* \rangle_{\mathcal{H}_K}, \quad (119)$$

where $(*)$ holds as $g^\circ$ is a minimizer of the corresponding loss, and $(**)$ holds as $\Gamma$ is positive semi-definite operator.

**2. Variance term.** Similarly to Proposition 5, it can be shown that for the finite-sample KRR solution, the variance term is approximately

$$||\hat{g} - g^\circ||_{L_2}^2 = \mathbb{E}\left[(\hat{g}(X) - g^\circ(X))^2\right] \approx \frac{1}{n} \operatorname{tr}\left[(T_K + \Gamma)^{-1} T_K (T_K + \Gamma)^{-1} T_{\tilde{\phi}(Z,\eta)^2, K}\right] + R(\eta, \hat{\eta}). \quad (120)$$

**3.** Now, we combine the bias and variance terms by decomposing $\hat{g} - g^* = \hat{g} - g^\circ + g^\circ - g^*$ and formulate the final excess risk:

$$||\hat{g} - g^*||_{L_2}^2 \lesssim \frac{1}{n} \operatorname{tr}\left[(T_K + \Gamma)^{-1} T_K (T_K + \Gamma)^{-1} T_{\tilde{\phi}(Z,\eta)^2, K}\right] + \langle g^*, \Gamma g^* \rangle_{\mathcal{H}_K} + R(\eta, \hat{\eta}). \quad (121)$$

**4.** Under (i) the conditional-variance assumption, the following holds approximately:

$$T_{\tilde{\phi}(Z,\eta)^2, K} \approx T_K^{1/2} M_{1/\nu} T_K^{1/2}. \quad (122)$$

.

Thus, the operator $(T_K + \Gamma)^{-1} T_K (T_K + \Gamma)^{-1}$ would suppresses the high-variance low-overlap eigenmodes more effectively when $\Gamma = \Gamma_{\text{OAR}}$ in comparison with $\Gamma = \Gamma_{\text{CR}}$ (given the same average regularization, namely $\text{tr}[\Gamma_{\text{OAR}}] = \text{tr}[\Gamma_{\text{CR}}]$). Hence, we obtain the desired inequality:

$$\text{tr}\left[(T_K + \Gamma_{\text{OAR}})^{-1} T_K (T_K + \Gamma_{\text{OAR}})^{-1} T_{\tilde{\phi}(Z,\eta)^2,K}\right] \leq \text{tr}\left[(T_K + \Gamma_{\text{CR}})^{-1} T_K (T_K + \Gamma_{\text{CR}})^{-1} T_{\tilde{\phi}(Z,\eta)^2,K}\right].$$
(123)

$\square$

# E OAR IMPLEMENTATION DETAILS

## E.1 RESCALING

In all the experiments, we performed the rescaling of our OAR so that it can be compared with the constant amount of regularization $\lambda > 0$ (or $p \in (0, 1)$):

$$\tilde{\lambda}(\nu(x)) = \lambda + \gamma \cdot \frac{\lambda}{\mathbb{E}[\lambda(\nu(X))]} \big( \lambda(\nu(x)) - \mathbb{E}[\lambda(\nu(X))] \big), \tag{124}$$

$$\tilde{p}(\nu(x)) = p + \gamma \cdot \min \left\{ \frac{p}{\mathbb{E}[p(\nu(X))]}; \frac{1-p}{1 - \mathbb{E}[p(\nu(X))]} \right\} \big( p(\nu(x)) - \mathbb{E}[p(\nu(X))] \big), \tag{125}$$

where $\gamma \in [0, 1]$ is an adaptivity coefficient. Here, $\gamma = 1$ leads to a full OAR, and $\gamma = 0$ is a constant regularization. In our experiments, we set $\gamma = 1$ for (a) parametric target models and $\gamma = 0.9$ for (b) non-parametric target KRR (to bound away the RKHS norm regularization from zero). The rescaling is crucial, as now we can ensure that $\tilde{\lambda}_\gamma(\nu)$ (1) is on average $\lambda$, (2) varies depending on overlap, and (3) lies in the admissible bounds ($\tilde{\lambda}_\gamma(\nu) > 0$ and $\tilde{p}_\gamma(\nu) \in (0, 1)$).

Notably, after rescaling our OAR, we also need to *adjust our debiased OAR (dOAR)*. Specifically, now we need to use $\mathbb{IF}(\tilde{\lambda}(\nu(x)); X, A)$ instead of $\mathbb{IF}(\lambda(\nu(x)); X, A)$ (and $\mathbb{IF}(\tilde{p}(\nu(x)); X, A)$ instead of $\mathbb{IF}(p(\nu(x)); X, A)$, respectively) in Eq. (10)–(11). The influence functions of the rescaled OAR can be found by using the chain rule (Kennedy, 2024; Luedtke, 2026) and are thus given by the following expressions:

$$\mathbb{IF}(\tilde{\lambda}(\nu(x)); X, A) = \gamma\lambda \left( \frac{\mathbb{IF}(\lambda(\nu(x)); X, A)}{\mathbb{E}[\lambda(\nu(X))]} - \frac{\lambda(\nu(X)) \, \mathbb{IF}(\mathbb{E}[\lambda(\nu(X))]; X, A)}{(\mathbb{E}[\lambda(\nu(X))])^2} \right), \tag{126}$$

$$\mathbb{IF}(\mathbb{E}[\lambda(\nu(X))]; X, A) = \int_{\mathcal{X}} \mathbb{IF}(\lambda(\nu(x)); X, A)\mathbb{P}(X = x) \, \mathrm{d}x + \lambda(\nu(X)) - \mathbb{E}[\lambda(\nu(X))], \tag{127}$$

$$\mathbb{IF}(\tilde{p}(\nu(x)); X, A) = \begin{cases} \gamma p \left( \frac{\mathbb{IF}(p(\nu(x)); X, A)}{\mathbb{E}[p(\nu(X))]} - \frac{p(\nu(X)) \, \mathbb{IF}(\mathbb{E}[p(\nu(X))]; X, A)}{(\mathbb{E}[p(\nu(X))])^2} \right), & \text{if } \frac{p}{\mathbb{E}[p(\nu(X))]} < \frac{1-p}{1 - \mathbb{E}[p(\nu(X))]}, \\ \gamma(1-p) \left( \frac{\mathbb{IF}(p(\nu(x)); X, A)}{1 - \mathbb{E}[p(\nu(X))]} - \frac{(1 - p(\nu(X))) \, \mathbb{IF}(\mathbb{E}[p(\nu(X))]; X, A)}{(1 - \mathbb{E}[p(\nu(X))])^2} \right), & \text{else,} \end{cases} \tag{128}$$

$$\mathbb{IF}(\mathbb{E}[p(\nu(X))]; X, A) = \int_{\mathcal{X}} \mathbb{IF}(p(\nu(x)); X, A)\mathbb{P}(X = x) \, \mathrm{d}x + p(\nu(X)) - \mathbb{E}[p(\nu(X))], \tag{129}$$

where $\mathbb{IF}(\lambda(\nu(x)); X, A)$ are $\mathbb{IF}(p(\nu(x)); X, A)$ provided in Proposition 4.

## E.2 OTHER IMPLEMENTATION DETAILS

We implemented our OAR/dOAR in PyTorch and Pyro. It proceeds in two stages as follows (see all the details in Algorithm 1).

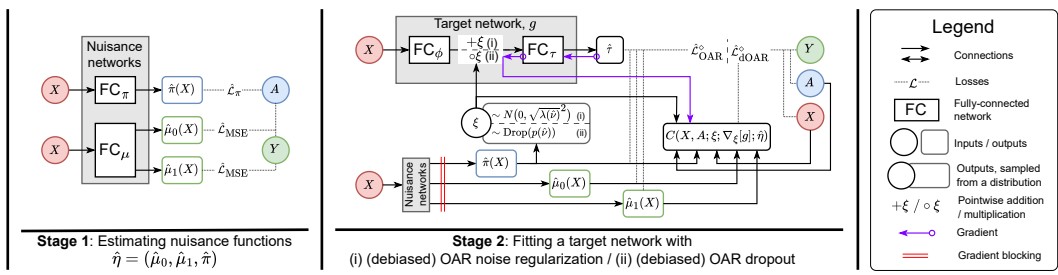

Figure 3: **An overview of our OAR for a neural network as a (a) parametric target model** $g$. Our OAR/debiased OAR are used at the second stage of the meta-learner to regularize the target network proportionally to the level of overlap (lower overlap leads to stronger regularization). Here, we instantiate OAR with noise injection for the middle layer of $g$: (i) OAR noise regularization and (ii) OAR dropout.

**Stage 1.** At the first stage, we used neural networks (NNs) with fully-connected (FC) layers and exponential linear units (ELUs) as activation functions. Specifically, to estimate the propensity score, we employed a multi-layer perceptron (MLP) $FC_\pi$ consisting of $L$ FC layers (with a tunable number of hidden units in each layer). For the conditional expected outcomes, we used a TARNet (Shalit et al., 2017) $FC_\mu$ consisting of a representation sub-network $FC_{\mu,\phi}$ and outcomes sub-networks $FC_{\mu,a}$ (again, each sub-network has a tunable number of layers). We trained the propensity network $FC_\pi$ and the outcomes network $FC_{\mu,a}$ with AdamW (Loshchilov & Hutter, 2019) and $n_{\text{epochs}} = 200$ ($n_{\text{epochs}} = 20$ for HC-MNIST dataset).

**Stage 2 (parametric target models).** For a second stage model, we used a target MLP $g$ with two sub-networks $FC_\phi$ and $FC_\tau$ and ELU activations. Both $FC_\phi$ and $FC_\tau$ have a fixed number of hidden units, matching the hidden units of the TARNet $FC_\mu$ from the first stage. Again, for training, we employed AdamW (Loshchilov & Hutter, 2019) with $n_{\text{epochs}} = 200$ ($n_{\text{epochs}} = 20$ for HC-MNIST dataset). For all the experiments with the parametric models, we set the adaptivity coefficient $\gamma = 1$. To further stabilize training of the target network, we (i) used exponential moving average (EMA) of model weights (Polyak & Juditsky, 1992) with a smoothing hyperparameter ($\kappa = 0.995$); (ii) trimmed too low propensity scores for both the pseudo-outcomes and our OAR/dOAR ($0.05 \leq \hat{\pi}(X) \leq 0.95$); and (iii) clipped too large values of the bias-correction term with a threshold $\alpha = 1.0$ ($|C^\diamond| \leq \alpha$ & $|C^\diamond| \leq \hat{\mathcal{L}}^\diamond_{\text{OAR}}(g, \hat{\eta})$).

**Stage 2 (non-parametric target models).** We used KRR with a radial basis function (RBF) kernel. An RBF bandwidth $h$ is fixed individually for each dataset (see Table 5). For all the experiments with the non-parametric models, we set the adaptivity coefficient $\gamma = 0.9$.

We use the same training data $\mathcal{D}$ for two stages of learning, as the (regularized) NNs belong to the Donsker class of estimators (van der Vaart, 2000; Kennedy, 2024).

**Computational complexity.** Importantly, the computation of the OAR/dOAR weights does not introduce a lot of computational burden: Given that we have the estimated propensity scores, both OAR/dOAR are functions of the latter. The dOAR additionally requires the evaluation of the gradient wrt. the target model inputs. Yet, this operation also scales linearly wrt. the minibatch size. Therefore, both OAR/dOAR can be evaluated in linear time depending on the minibatch size.

---

**Algorithm 1** Pseudocode of our OAR/dOAR with meta-learners

---

1: **Input:** Training dataset $\mathcal{D}$; OAR/dOAR version $\diamond \in \{+\xi, \circ\xi, \mathcal{H}\}$; average regularization strength $\lambda > 0 / p \in (0, 1)$
2: **Stage 1**: Estimate nuisance functions $\hat{\eta} = (\hat{\mu}_0, \hat{\mu}_1, \hat{\pi})$
3:      Fit a propensity network $\text{FC}_\pi$ (MLP) by minimizing a BCE loss, $\hat{\mathcal{L}}_\pi = \mathbb{P}_n\{\text{BCE}(\text{FC}_\pi(X), A)\}$
4:      Fit an outcomes network $\text{FC}_\mu$ (TARNet) by minimizing an MSE loss, $\hat{\mathcal{L}}_{\text{MSE}} = \mathbb{P}_n\{(Y - \text{FC}_\mu(X, A))^2\}$
5: **Output:** Nuisance functions estimators $\hat{\eta} = (\text{FC}_\mu(x, 0), \text{FC}_\mu(x, 1), \text{FC}_\pi(x))$
6: **Stage 2**: Fit a target model $\hat{g} = \arg\min_{g \in \mathcal{G}} \hat{\mathcal{L}}^\diamond_{\text{OAR}}(g, \hat{\eta})$    /    $\hat{g} = \arg\min_{g \in \mathcal{G}} \hat{\mathcal{L}}^\diamond_{\text{dOAR}}(g, \hat{\eta})$
7:      $\lambda(\hat{\nu}(X)) \leftarrow \ldots$ (see Eq. (4)); $p(\hat{\nu}(X)) \leftarrow \lambda(\hat{\nu}(X))/(\lambda(\hat{\nu}(X)) + 1)$
8:      $I(X) \leftarrow \mathbb{1}\{0.05 \le \hat{\pi}(X) \le 0.95\}$                               $\triangleright$ Trimming indicator
9:      $\mathbb{E}[\widehat{\lambda(\nu(X))}] \leftarrow \mathbb{P}_n\{I(X) \cdot \lambda(\hat{\nu}(X))\}/\mathbb{P}_n\{I(X)\}$
10:     $\mathbb{E}[\widehat{p(\nu(X))}] \leftarrow \mathbb{P}_n\{I(X) \cdot p(\hat{\nu}(X))\}/\mathbb{P}_n\{I(X)\}$
11:     $\tilde{\lambda}(\hat{\nu}(X)) \leftarrow \lambda + \gamma \cdot I(X) \cdot \frac{\lambda}{\mathbb{E}[\widehat{\lambda(\nu(X))}]}\left(\lambda(\hat{\nu}(x)) - \mathbb{E}[\widehat{\lambda(\nu(X))}]\right)$ (see Eq. (124))      $\triangleright$ Rescaling
12:     $\tilde{p}(\hat{\nu}(X)) \leftarrow p + \gamma \cdot I(X) \cdot \min\left(\frac{p}{\mathbb{E}[\widehat{p(\nu(X))}]}; \frac{1-p}{1-\mathbb{E}[\widehat{p(\nu(X))}]}\right)\left(p(\hat{\nu}(x)) - \mathbb{E}[\widehat{p(\nu(X))}]\right)$ (see Eq. (125))
13:     **if** Target model $==$ MLP **then**                          $\triangleright$ Parametric target models
14:         **for** $i = 0$ **to** $\lceil n_{\text{epochs}} \cdot n/b_{\text{T}}\rceil$ **do**
15:             Draw a minibatch $\mathcal{B} = \{X, A, Y, \tilde{\lambda}(\hat{\nu}(X)), \tilde{p}(\hat{\nu}(X)), I(X)\}$ of size $b_{\text{T}}$ from $\mathcal{D}$
16:             $\Phi \leftarrow \text{FC}_\phi(X)$
17:             $\xi \sim N\left(0, \sqrt{\tilde{\lambda}(\hat{\nu}(X))}^2\right)$    /    $\xi \sim \text{Drop}(\tilde{p}(\hat{\nu}(X)))$         $\triangleright$ Noise regularization / dropout
18:             $g(X) \leftarrow \text{FC}_\tau(\Phi + \xi)$    /    $g(X) \leftarrow \text{FC}_\tau(\Phi \circ \xi)$
19:             $\hat{\mathcal{L}}^\diamond_{\text{OAR}}(g, \hat{\eta}) \leftarrow \mathbb{P}_{b_{\text{T}}}\left\{\rho(A, \hat{\pi}(X))\left(I(X) \cdot \phi(Z, \hat{\eta}) - g(X)\right)^2\right\}$
20:             **if** dOAR **then**
21:                 $C^\diamond \leftarrow \mathbb{P}_{b_{\text{T}}}\left\{I(X) \cdot C^\diamond(X; A; \xi; \nabla_\xi[g]; \hat{\eta})\right\}$ (see Eq. (10)-(11) and Eq. (126),(128))
22:                 $\hat{\mathcal{L}}^\diamond_{\text{dOAR}}(g, \hat{\eta}) \leftarrow \hat{\mathcal{L}}^\diamond_{\text{OAR}}(g, \hat{\eta}) + C^\diamond \cdot \mathbb{1}\{|C^\diamond| \le \alpha \ \& \ |C^\diamond| \le \hat{\mathcal{L}}^\diamond_{\text{OAR}}(g, \hat{\eta})\}$
23:             **end if**
24:             Gradient & EMA update of the target network $g$ wrt. $\hat{\mathcal{L}}^\diamond_{\text{OAR}}(g, \hat{\eta})/\hat{\mathcal{L}}^\diamond_{\text{dOAR}}(g, \hat{\eta})$
25:         **end for**
26:         $\hat{g}(x) \leftarrow \text{FC}_\tau(\text{FC}_\phi(x))$
27:     **else if** Target model $==$ KRR **then**                    $\triangleright$ Non-parametric target models
28:         **if** OAR **then**
29:             $\hat{g}(x) \leftarrow \mathbf{K}_{xX}\left(\mathbf{R}(\hat{\pi})\,\mathbf{K}_{XX} + n\mathbf{\Lambda}(\hat{\nu})\right)^{-1}\mathbf{R}(\hat{\pi})\,\mathbf{\Phi}(\hat{\eta})$ (see Eq. (21))
30:         **else**
31:             Undefined (see discussion in Appendix C)
32:         **end if**
33:     **end if**
34: **Output:** CATE estimator $\hat{g}$

---

### E.3 HYPERPARAMETER TUNING

We performed hyperparameter tuning of the first-stage models based on five-fold cross-validation using the training subset. For the second stage, we used fixed hyperparameters for all the experiments, as an exact hyperparameter search is not possible for target CATE models solely with the observational data (Curth & van der Schaar, 2023). Table 5 provides all the details on hyperparameter tuning. For reproducibility, we made tuned hyperparameters available in our GitHub.[7]

Table 5: Hyperparameter tuning for our OAR/dOAR with meta-learners.

| Stage | Model | Hyperparameter | Range / Value |
|---|---|---|---|
| **Stage 1** | Propensity network (MLP) | Learning rate | 0.001, 0.005, 0.01 |
| | | Minibatch size, $b_N$ | 32, 64, 128 |
| | | Weight decay | 0.0, 0.001, 0.01, 0.1 |
| | | Hidden layers in $FC_\pi$ | $L$ |
| | | Hidden units in $FC_\pi$ | $R\,d_x$, $1.5\,R\,d_x$, $2\,R d_x$ |
| | | Tuning strategy | random grid search with 50 runs |
| | | Tuning criterion | factual BCE loss |
| | | Optimizer | AdamW |
| | Outcomes network (TARNet) | Learning rate | 0.001, 0.005, 0.01 |
| | | Minibatch size, $b_N$ | 32, 64, 128 |
| | | Hidden units in $FC_{\mu,\phi}$ | $R\,d_x$, $1.5\,R\,d_x$, $2\,R\,d_x$ |
| | | Dimensionality of $\Phi$, $d_\phi$ | $R\,d_x$, $1.5\,R\,d_x$, $2\,R\,d_x$ |
| | | Hidden units in $FC_{\mu,a}$ | $R\,d_\phi$, $1.5\,R\,d_\phi$, $2\,R\,d_\phi$ |
| | | Weight decay | 0.0, 0.001, 0.01, 0.1 |
| | | Tuning strategy | random grid search with 50 runs |
| | | Tuning criterion | factual MSE loss |
| | | Optimizer | AdamW |
| **Stage 2** | Target network (MLP) | Learning rate | 0.005 |
| | | Minibatch size, $b_T$ | 64 |
| | | EMA of model weights, $\kappa$ | 0.995 |
| | | Hidden units in $g$ | Hidden units in $FC_\mu$ |
| | | Tuning strategy | no tuning |
| | | Optimizer | AdamW |
| | | Adaptivity coefficient, $\gamma$ | 1 |
| | Target KRR | RBF bandwidth, $h$ | $h$ |
| | | Adaptivity coefficient, $\gamma$ | 0.9 |

$L = 1$ (synthetic data, IHDP dataset, ACIC 2016 datasets), $L = 2$ (HC-MNIST dataset)
$R = 2$ (synthetic data), $R = 1$ (IHDP dataset), $R = 0.25$ (ACIC 2016 datasets, HC-MNIST dataset)
$h = 0.1$ (synthetic data), $h = 2$ (ACIC 2016 datasets), $h = 5$ (IHDP dataset)

---

[7]https://github.com/Valentyn1997/OAR.

## F  DATASET DETAILS

### F.1  SYNTHETIC DATA

We adapted the synthetic dataset from Melnychuk et al. (2023) where the amount of overlap can be varied. Specifically, we took the original generative mechanisms for the covariate $X$ and the treatment $A$ but modified the data-generating process of the outcome $Y$:

$$\begin{cases} X \sim & \text{Mixture}\big(0.5N(0,1) + 0.5N(b,1)\big), \\ A := & \begin{cases} 1, & -U_A < \log\big(\pi(X)/(1-\pi(X))\big) \\ 0, & \text{otherwise} \end{cases}, \quad \pi(x) = \frac{N(x;0,1^2)}{N(x;0,1^2)+N(x;b,1^2)}, \quad U_A \sim \text{Logistic}(0,1), \\ Y \sim & N(3\cos(3X^2 - 2X + 0.5) - 2.5\sin(3X^2 - 2X + 0.5), 1^2), \end{cases}$$

(130)

where $N(x;\mu,\sigma^2)$ is a density of a normal distribution $N(\mu,\sigma^2)$ with a mean $\mu$ and a standard deviation $\sigma$; and a parameter $b \in \mathbb{R}$ regulates the amount of overlap ($b = 0$ implies a perfect overlap). In our synthetic experiments, we set $b = 2$.

### F.2  IHDP DATASET

The Infant Health and Development Program (IHDP) dataset (Hill, 2011; Shalit et al., 2017) is a standard semi-synthetic benchmark for assessing treatment effect estimators. It comes with 100 predefined train–test splits, each containing $n_{\text{train}} = 672$, $n_{\text{test}} = 75$, and $d_x = 25$. The ground-truth CAPOs are then given by an exponential function ($\mu_0(x)$) and a linear function ($\mu_1(x)$). Notably, the IHDP dataset has a well-known drawback of a severe lack of overlap, which causes instability for approaches that depend on propensity-score re-weighting (Curth & van der Schaar, 2021b; Curth et al., 2021).

### F.3  ACIC 2016 DATASETS

The ACIC 2016 benchmark (Dorie et al., 2019) builds its covariates from the extensive Collaborative Perinatal Project on developmental disorders (Niswander, 1972). Its datasets differ in (i) the number of ground-truth confounders, (ii) the degree of covariate overlap, and (iii) the smoothness and the functional form of the CAPOs. In total, ACIC 2016 supplies **77 different data-generating processes**, each paired with 100 identically sized samples. After one-hot encoding categorical variables, every sample contains $n = 4,802$ observations and $d_X = 82$ features.

### F.4  HC-MNIST DATASET

The HC-MNIST benchmark was proposed as a high-dimensional, semi-synthetic dataset (Jesson et al., 2021), derived from the original MNIST digit images (LeCun, 1998). It contains $n_{\text{train}} = 60,000$ training images and $n_{\text{test}} = 10,000$ test images. HC-MNIST compresses each high-resolution image into a single latent coordinate, $\phi$, so that the potential outcomes are complex functions of both the image's mean pixel intensity and its digit label. Treatment assignment is determined by this one-dimensional summary $\phi$ together with an additional latent (synthetic) confounder $U$, which we treat as an observed covariate. Hence, HC-MNIST is characterized by the following data-generating process:

$$\begin{cases} U \sim \text{Bern}(0.5), \\ X \sim \text{MNIST-image}(\cdot), \\ \phi := \left(\text{clip}\left(\frac{\mu_{N_x} - \mu_c}{\sigma_c}; -1.4, 1.4\right) - \text{Min}_c\right)\frac{\text{Max}_c - \text{Min}_c}{1.4 - (-1.4)}, \\ \alpha(\phi;\Gamma^*) := \frac{1}{\Gamma^* \text{sigmoid}(0.75\phi+0.5)} + 1 - \frac{1}{\Gamma^*}, \quad \beta(\phi;\Gamma^*) := \frac{\Gamma^*}{\text{sigmoid}(0.75\phi+0.5)} + 1 - \Gamma^*, \\ A \sim \text{Bern}\left(\frac{u}{\alpha(\phi;\Gamma^*)} + \frac{1-u}{\beta(\phi;\Gamma^*)}\right), \\ Y \sim N\big((2A-1)\phi + (2A-1) - 2\sin(2(2A-1)\phi) - 2(2U-1)(1+0.5\phi), 1\big), \end{cases}$$

(131)

where $c$ is a label of the digit from the sampled image $X$; $\mu_{N_x}$ is the average intensity of the sampled image; $\mu_c$ and $\sigma_c$ are the mean and standard deviation of the average intensities of the images with

the label $c$; and $\text{Min}_c = -2 + \frac{4}{10}c, \text{Max}_c = -2 + \frac{4}{10}(c+1)$. The parameter $\Gamma^*$ defines what factor influences the treatment assignment to a larger extent, i.e., the additional confounder or the one-dimensional summary. We set $\Gamma^* = \exp(1)$. For further details, we refer to Jesson et al. (2021).

# G    ADDITIONAL EXPERIMENTAL RESULTS

## G.1    SYNTHETIC DATA

We adapted a fully-synthetic dataset ($d_x = 1$) from Melnychuk et al. (2023) where the amount of overlap can be varied. Here, the ground truth CATE is 0, yet both conditional expected outcomes are highly non-linear. We simulated a low-overlap setting with $n_{\text{train}} = 250$ (see Fig. 1).

**Results.** Results are shown in Fig. 4. We see that *our OAR/dOAR noise regularization improves the performance of the CR + DR-Learner*. At the same time, both OAR/dOAR dropout and RKHS norm improve the performance of (almost) all the learners + CR. This hints at a more flexible nature of dropout and RKHS norm, as they depend on the covariates and, thus, are more adaptive.

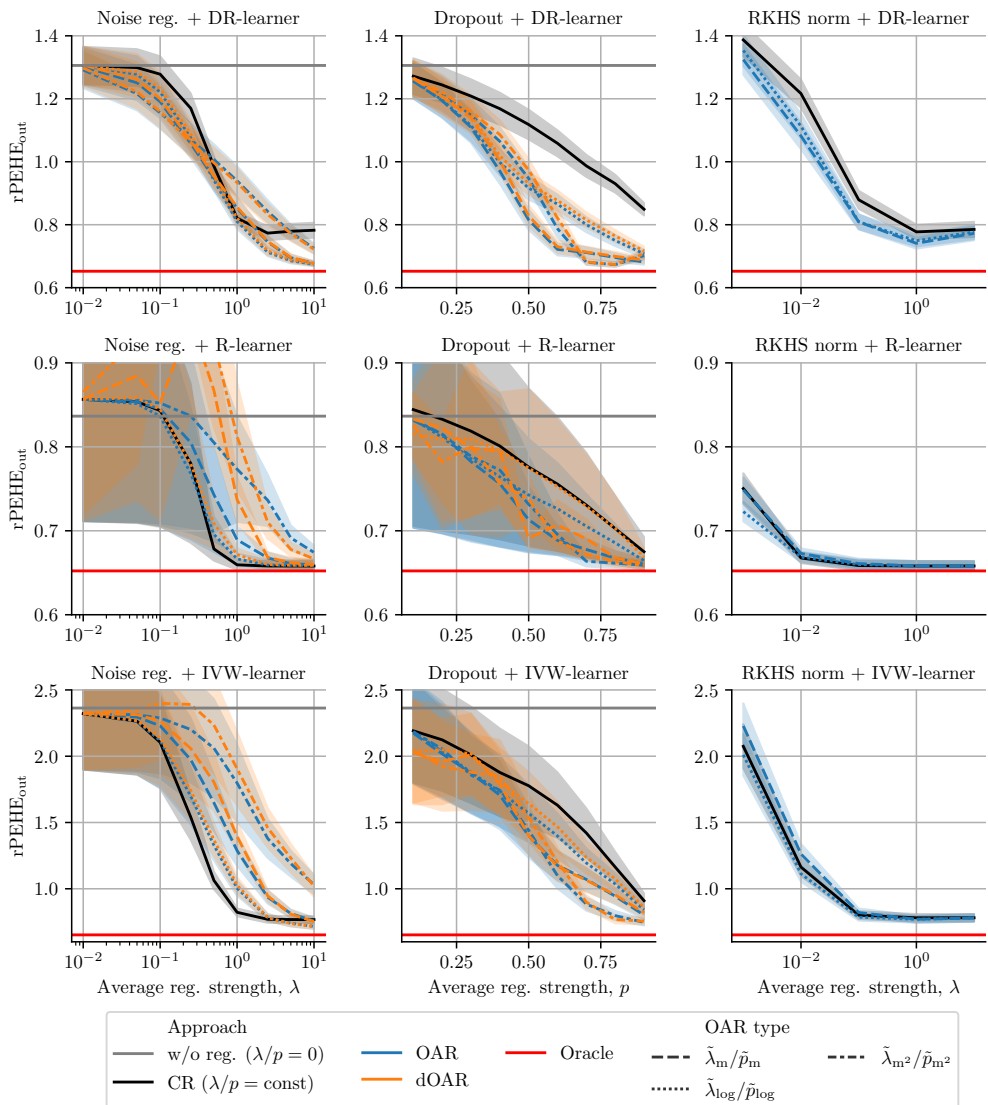

Figure 4: **Results for synthetic experiments.** Reported: rPEHE$_{\text{out}}$; mean $\pm$ se over 40 runs. Lower is better.

## G.2 HC-MNIST DATASET

In Table 6, we demonstrate the remaining[8] experiments for the HC-MNIST dataset. Therein, our OAR/dOAR improves over CR + DR-/R-/IVW-learners in the majority of combinations, often significantly. Furthermore, the best performance for almost every regularization value is achieved by our OAR/dOAR.

Table 6: **Results for HC-MNIST experiments for OAR/dOAR($\tilde{\lambda}_{\log}/\tilde{p}_{\log}$) and OAR/dOAR($\tilde{\lambda}_{\mathrm{m}^2}/\tilde{p}_{\mathrm{m}^2}$)** . Reported: rPEHE$_{\mathrm{out}}$ ($\Delta$rPEHE$_{\mathrm{out}}$); mean $\pm$ std over 30 runs.

| Learner | Reg. $\lambda/p =$ Approach | Noise reg. 0.05 | 0.1 | 0.25 | Dropout 0.1 | 0.3 | 0.5 |
|---|---|---|---|---|---|---|---|
| DR | CR ($\lambda/p =$ const) | $0.741 \pm 0.038$ | $0.729 \pm 0.037$ | $0.702 \pm 0.030$ | $0.735 \pm 0.036$ | $0.716 \pm 0.032$ | $0.704 \pm 0.025$ |
| | OAR($\tilde{\lambda}_{\log}/\tilde{p}_{\log}$) | $0.736 \pm 0.039$ ($-0.006$) | $0.724 \pm 0.035$ ($-0.005$) | $0.686 \pm 0.031$ ($-0.016$) | $0.730 \pm 0.039$ ($-0.005$) | $0.713 \pm 0.031$ ($-0.004$) | $0.700 \pm 0.025$ ($-0.005$) |
| | dOAR($\tilde{\lambda}_{\log}/\tilde{p}_{\log}$) | $0.737 \pm 0.038$ ($-0.004$) | $0.725 \pm 0.036$ ($-0.004$) | $0.687 \pm 0.030$ ($-0.015$) | $0.703 \pm 0.038$ ($-0.032$) | $0.710 \pm 0.029$ ($-0.007$) | $0.702 \pm 0.025$ ($-0.002$) |
| | OAR($\tilde{\lambda}_{\mathrm{m}^2}/\tilde{p}_{\mathrm{m}^2}$) | $0.719 \pm 0.037$ ($-0.023$) | $0.713 \pm 0.063$ ($-0.017$) | $0.710 \pm 0.068$ ($+0.008$) | $0.731 \pm 0.038$ ($-0.004$) | $0.708 \pm 0.033$ ($-0.009$) | $0.701 \pm 0.026$ ($-0.003$) |
| | dOAR($\tilde{\lambda}_{\mathrm{m}^2}/\tilde{p}_{\mathrm{m}^2}$) | $0.703 \pm 0.026$ ($-0.039$) | $0.699 \pm 0.041$ ($-0.030$) | $0.709 \pm 0.214$ ($+0.007$) | $0.700 \pm 0.040$ ($-0.035$) | $0.692 \pm 0.036$ ($-0.025$) | $0.699 \pm 0.026$ ($-0.005$) |
| R | CR ($\lambda/p =$ const) | $0.714 \pm 0.015$ | $0.705 \pm 0.010$ | $\mathbf{0.673 \pm 0.007}$ | $0.715 \pm 0.027$ | $0.705 \pm 0.027$ | $0.692 \pm 0.018$ |
| | OAR($\tilde{\lambda}_{\log}/\tilde{p}_{\log}$) | $0.715 \pm 0.015$ ($+0.001$) | $0.702 \pm 0.011$ ($-0.003$) | $\underline{0.674 \pm 0.008}$ ($+0.001$) | $0.713 \pm 0.022$ ($-0.002$) | $0.700 \pm 0.015$ ($-0.006$) | $0.687 \pm 0.014$ ($-0.006$) |
| | dOAR($\tilde{\lambda}_{\log}/\tilde{p}_{\log}$) | $0.711 \pm 0.013$ ($-0.003$) | $0.701 \pm 0.010$ ($-0.004$) | $0.676 \pm 0.008$ ($+0.003$) | $\underline{0.688 \pm 0.020}$ ($-0.027$) | $0.699 \pm 0.024$ ($-0.006$) | $0.688 \pm 0.018$ ($-0.004$) |
| | OAR($\tilde{\lambda}_{\mathrm{m}^2}/\tilde{p}_{\mathrm{m}^2}$) | $0.699 \pm 0.070$ ($-0.015$) | $0.699 \pm 0.145$ ($-0.005$) | $0.695 \pm 0.448$ ($+0.022$) | $0.717 \pm 0.022$ ($+0.002$) | $0.696 \pm 0.020$ ($-0.010$) | $0.684 \pm 0.020$ ($-0.008$) |
| | dOAR($\tilde{\lambda}_{\mathrm{m}^2}/\tilde{p}_{\mathrm{m}^2}$) | $\mathbf{0.694 \pm 0.031}$ ($-0.020$) | $\mathbf{0.696 \pm 0.072}$ ($-0.008$) | $0.731 \pm 0.341$ ($+0.058$) | $\mathbf{0.680 \pm 0.011}$ ($-0.035$) | $\mathbf{0.680 \pm 0.013}$ ($-0.025$) | $\mathbf{0.680 \pm 0.011}$ ($-0.012$) |
| IVW | CR ($\lambda/p =$ const) | $1.062 \pm 0.246$ | $1.043 \pm 0.235$ | $0.987 \pm 0.201$ | $1.089 \pm 0.251$ | $1.050 \pm 0.259$ | $1.047 \pm 0.281$ |
| | OAR($\tilde{\lambda}_{\log}/\tilde{p}_{\log}$) | $1.038 \pm 0.242$ ($-0.025$) | $1.007 \pm 0.219$ ($-0.036$) | $0.893 \pm 0.193$ ($-0.094$) | $1.099 \pm 0.266$ ($+0.010$) | $1.027 \pm 0.242$ ($-0.023$) | $0.975 \pm 0.220$ ($-0.072$) |
| | dOAR($\tilde{\lambda}_{\log}/\tilde{p}_{\log}$) | $1.051 \pm 0.236$ ($-0.011$) | $1.022 \pm 0.218$ ($-0.021$) | $0.904 \pm 0.186$ ($-0.083$) | $1.070 \pm 0.257$ ($-0.019$) | $1.098 \pm 0.311$ ($+0.048$) | $1.075 \pm 0.294$ ($+0.028$) |
| | OAR($\tilde{\lambda}_{\mathrm{m}^2}/\tilde{p}_{\mathrm{m}^2}$) | $1.039 \pm 0.231$ ($-0.023$) | $1.030 \pm 0.226$ ($-0.013$) | $1.053 \pm 0.431$ ($+0.066$) | $1.087 \pm 0.256$ ($-0.002$) | $1.039 \pm 0.252$ ($-0.011$) | $0.820 \pm 0.189$ ($-0.227$) |
| | dOAR($\tilde{\lambda}_{\mathrm{m}^2}/\tilde{p}_{\mathrm{m}^2}$) | $1.009 \pm 0.244$ ($-0.053$) | $1.047 \pm 0.224$ ($+0.004$) | $1.136 \pm 0.295$ ($+0.149$) | $1.036 \pm 0.231$ ($-0.052$) | $1.093 \pm 0.319$ ($+0.043$) | $0.810 \pm 0.240$ ($-0.238$) |
| Oracle | | 0.513 | | | | | |

Lower = better (best in bold, second best underlined). Change over the baseline in brackets (significant improvement in green, significant worsening in red, $\alpha = 0.05$)

## G.3 ADDITIONAL BASELINES

**Baselines.** In the following, we provide an absolute comparison between our OAR/dOAR approach and other existing approaches that tackle low overlap (see Sec. 2). Here, we included **trimming** of the IPTW weights (with the threshold $t \in \{0.05, 0.1, 0.2\}$). Note that OAR/dOAR also uses a default amount of trimming $t = 0.05$ to stabilize the training. Also, we added **balancing representations** with different empirical probability metrics (Johansson et al., 2016; Shalit et al., 2017) with $\alpha \in \{0.5, 5.0\}$, namely Wasserstein metric (WM) and maximum mean discrepancy (MMD). For a fair comparison, we implemented balancing for the target models (as it was suggested by Melnychuk et al. (2026)).

**Results.** Results for synthetic and the HC-MNIST datasets are shown in Tables 7 and 8, respectively. For the synthetic data, our OAR/dOAR with the multiplicative regularization function in combination with the DR-learner outperforms other approaches. For the HC-MNIST dataset, our OAR/dOAR with the multiplicative regularization function outperforms all the other baselines. The main reason that other baselines fall short of our approach in the HC-MNIST is that (i) trimming simply discards datapoints with low overlap; and (ii) balancing representations becomes highly unstable with the high-dimensional data (as it gets increasingly harder to estimate empirical probability metrics).

---

[8] Notably, we do not report the results for OAR/dOAR RKHS norm, as kernel ridge regression does not scale well with the dimensionality of the covariates (here, $d_x = 784 + 1$).

Table 7: **Results for synthetic experiments for OAR/dOAR($\tilde{\lambda}_m/\tilde{p}_m$) and other baselines that tackle low overlap**. Reported: rPEHE$_{\text{out}}$; mean $\pm$ std over 40 runs.

| | DR-learner | R-learner |
|---|---|---|
| Trimming ($t = 0.05$) | $1.306 \pm 0.375$ | $0.837 \pm 0.861$ |
| Trimming ($t = 0.1$) | $1.339 \pm 0.374$ | $0.837 \pm 0.861$ |
| Trimming ($t = 0.2$) | $1.353 \pm 0.376$ | $0.837 \pm 0.861$ |
| Balancing ($\alpha = 0.5$; MMD) | $1.032 \pm 0.406$ | $\mathbf{0.658 \pm 0.037}$ |
| Balancing ($\alpha = 0.5$; WM) | $1.222 \pm 0.370$ | $\mathbf{0.658 \pm 0.037}$ |
| Balancing ($\alpha = 5.0$; MMD) | $0.878 \pm 0.289$ | $\mathbf{0.658 \pm 0.037}$ |
| Balancing ($\alpha = 5.0$; WM) | $1.156 \pm 0.359$ | $\mathbf{0.658 \pm 0.037}$ |
| OAR Dropout ($\tilde{p}_m$; $p = 0.5$) | $\mathbf{0.815 \pm 0.203}$ | $0.713 \pm 0.249$ |
| dOAR Dropout ($\tilde{p}_m$; $p = 0.5$) | $\underline{0.828 \pm 0.211}$ | $0.691 \pm 0.072$ |
| OAR Noise reg. ($\tilde{\lambda}_m$; $\lambda = 1.0$) | $0.853 \pm 0.211$ | $\underline{0.689 \pm 0.099}$ |
| dOAR Noise reg. ($\tilde{\lambda}_m$; $\lambda = 1.0$) | $0.856 \pm 0.207$ | $0.737 \pm 0.172$ |
| Oracle | 0.652 | |

Lower = better (best in bold, second best underlined)

Table 8: **Results for HC-MNIST experiments for OAR/dOAR($\tilde{\lambda}_m/\tilde{p}_m$) and other baselines that tackle low overlap**. Reported: rPEHE$_{\text{out}}$; mean $\pm$ std over 30 runs.

| | DR-learner | R-learner |
|---|---|---|
| Trimming ($t = 0.05$) | $0.754 \pm 0.040$ | $0.731 \pm 0.029$ |
| Trimming ($t = 0.1$) | $0.736 \pm 0.019$ | $0.731 \pm 0.029$ |
| Trimming ($t = 0.2$) | $0.714 \pm 0.010$ | $0.731 \pm 0.029$ |
| Balancing ($\alpha = 0.5$; MMD) | $0.721 \pm 0.027$ | $0.780 \pm 0.045$ |
| Balancing ($\alpha = 0.5$; WM) | $0.908 \pm 0.167$ | $0.946 \pm 0.033$ |
| Balancing ($\alpha = 5.0$; MMD) | $0.842 \pm 0.230$ | $1.210 \pm 0.007$ |
| Balancing ($\alpha = 5.0$; WM) | $1.171 \pm 0.177$ | $1.124 \pm 0.031$ |
| OAR Dropout ($\tilde{p}_m$; $p = 0.3$) | $0.713 \pm 0.032$ | $0.696 \pm 0.013$ |
| dOAR Dropout ($\tilde{p}_m$; $p = 0.3$) | $\mathbf{0.705 \pm 0.031}$ | $\mathbf{0.687 \pm 0.013}$ |
| OAR Noise reg. ($\tilde{\lambda}_m$; $\lambda = 0.1$) | $0.726 \pm 0.036$ | $0.696 \pm 0.009$ |
| dOAR Noise reg. ($\tilde{\lambda}_m$; $\lambda = 0.1$) | $\underline{0.712 \pm 0.033}$ | $\underline{0.695 \pm 0.010}$ |
| Oracle | 0.513 | |

Lower = better (best in bold, second best underlined)

## G.4 RUNTIME

Table 9 provides the duration of one run on the IHDP dataset. Each run includes two stages: in stage 1, we fit nuisance functions; and, in stage 2, we fit all three meta-learners, namely, DR-/R-/IVW-learners (see Algorithm 1). Here, we compared the constant regularization strategy (CR) with our OAR and dOAR w/ the multiplicative regularization function, $\tilde{\lambda}_m/\tilde{p}_m$ (runtimes for the logarithmic and squared multiplicative regularization functions are analogous). We observe that our OAR has almost the same runtime as the constant regularization. On the other hand, our dOAR has slightly longer training times that are attributed to the calculation of the gradient $\nabla_\xi[g]$ in the debiasing term, see Eq.(10)–(11).

Table 9: **Total runtime (in minutes) for different regularization strategies**. Reported: mean $\pm$ std over 100 runs. Lower is better. Experiments were carried out on 2 GPUs (NVIDIA A100-PCIE-40GB) with Intel Xeon Silver 4316 CPUs @ 2.30GHz.

| Reg. Approach | Noise reg. | Dropout | RKHS norm |
|---|---|---|---|
| CR ($\lambda/p = $ const) | $1.85 \pm 0.03$ | $1.88 \pm 0.04$ | $1.40 \pm 0.21$ |
| OAR ($\tilde{\lambda}_m/\tilde{p}_m$) | $1.87 \pm 0.04$ | $1.89 \pm 0.03$ | $1.35 \pm 0.25$ |
| dOAR ($\tilde{\lambda}_m/\tilde{p}_m$) | $2.96 \pm 0.09$ | $3.86 \pm 0.08$ | — |

