# OpenReview forum: "Overlap-Adaptive Regularization for Conditional Average Treatment Effect Estimation"
_ICLR.cc/2026/Conference — ICLR 2026 Poster_

### Official Review · Reviewer_EKxX · 2025-10-28

**Soundness:** 3
**Presentation:** 2
**Contribution:** 2
**Rating:** 4
**Confidence:** 3

**Summary:**

This paper introduces an overlap-adaptive regularization approach for estimating Conditional Average Treatment Effects (CATE). The key idea is to adjust the regularization strength based on the degree of overlap: regions with strong overlap receive lower regularization, while regions with weak overlap are penalized more heavily to mitigate instability. Additionally, the authors propose a debiased version of the estimator to address bias arising from estimation errors in propensity scores.

**Strengths:**

The idea of overlap-adaptive regularization appears to be a reasonable strategy for CATE estimation. The authors introduce two practical implementation techniques—noise injection and dropout—to facilitate ease of application in real-world settings. They provide theoretical analysis under a linear model framework. Numerical experiments are conducted to illustrate the performance of the proposed estimator.

**Weaknesses:**

1. The theoretical analysis provided in the paper appears inadequate. The results for the linear model can be considered preliminary, as the simplicity of the linear setting limits the relevance of regularization—making it less critical in practice. It would be more valuable to establish theoretical guarantees for nonparametric models, which are more representative of real-world applications. Furthermore, the numerical experiments primarily involve neural networks and kernel-based methods. While Appendix C presents theoretical results for RKHS, these should ideally be included in the main paper to highlight their importance. Moreover, Appendix C primarily focuses on deriving the closed-form solution for the RKHS estimator, without providing any discussion of the statistical properties of the final estimator. This omission limits the theoretical contribution and leaves readers uncertain about the estimator’s theoretical improvement over the traditional CR RKHS estimator.


2. The simulation results are based on only 10 runs, which raises concerns. Moreover, the improvement of the proposed overlap-adaptive regularization (OAR) over CR does not appear to be statistically significant when accounting for the reported standard errors. Similarly, for the synthetic data example.

**Questions:**

I am curious about the different performance of the noise-injection and dropout methods shown in Figure 2. These two regularization strategies appear to behave quite differently under constant regularization. Specifically, for noise injection, the conventional regularization (CR) performs relatively well, whereas for dropout, CR performs poorly.

**Details Of Ethics Concerns:**

N/A.

---

> ### Author Response · Authors · 2025-11-20
> **Rebuttal [1 / 2]**
>
> Thank you for your in-depth review! We are glad to hear that you appreciate our strategy to CATE estimation and our numerical experiments. In the following, we respond to your concerns and explain how we improved our paper as a result (we highlighted the corresponding changes in the **revised PDF** in **blue color**).
>
> ### Response to weaknesses
> 1. > The theoretical analysis provided in the paper appears inadequate. The results for the linear model can be considered preliminary, as the simplicity of the linear setting limits the relevance of regularization—making it less critical in practice.
>
>    The theoretical analysis in the paper focuses on linear models (as in Proposition 5: Excess prediction risk for OAR/dOAR). While these results are derived in a linear setting, they **offer clear and rigorous insights that form a strong theoretical foundation** for future extensions – particularly toward nonlinear and neural network architectures. Moreover, the theoretical findings help us understand the behavior of different approaches - for example, by allowing us to reason about the relative advantages of noise regularization versus dropout (as discussed in our response to your “Question” at the end).
>
>    Our analysis highlights an important trade-off: while many theoretical properties of noise injection and dropout depend intricately on the neural network architecture, our goal was to establish a **general, model-agnostic regularization framework**. In that sense, we follow the standard approach in ML research and prioritize general results over architecture-specific findings.
>
>    > Furthermore, the numerical experiments primarily involve neural networks and kernel-based methods. While Appendix C presents theoretical results for RKHS, these should ideally be included in the main paper to highlight their importance.
>
>    **Action**. Great idea! We followed your suggestion and moved some of the main theoretical results for the OAR RKHS to the main part of the paper (see **Sec. 5.4** in the **updated PDF**).
>
>    > It would be more valuable to establish theoretical guarantees for nonparametric models, which are more representative of real-world applications. … Moreover, Appendix C primarily focuses on deriving the closed-form solution for the RKHS estimator, without providing any discussion of the statistical properties of the final estimator. This omission limits the theoretical contribution and leaves readers uncertain about the estimator’s theoretical improvement over the traditional CR RKHS estimator.
>
>    Thank you for your suggestions on how to strengthen our theory! Upon reading your comment, we realized that the results of our Proposition 5 (which only applies to linear models) can also be extended to the non-parametric case (= OAR RKHS norm).
>
>    **Action**. We have added a **new Proposition 7** to **Appendix C**, where we derived the excess prediction risk for the DR-learner with our OAR RKHS norm. Therein, we also showed, similarly to **Proposition 5**, that our OAR improves over the CR under the (i) conditional variance assumption and (ii)  low-overlap-low-heterogeneity inductive bias.

---

> > ### Author Response · Authors · 2025-11-20
> > **Rebuttal [2 / 2]**
> >
> > 2. > The simulation results are based on only 10 runs, which raises concerns. Moreover, the improvement of the proposed overlap-adaptive regularization (OAR) over CR does not appear to be statistically significant when accounting for the reported standard errors.
> >
> >    Thank you for this helpful suggestion. Previously, we reported *the standard deviations in our results for the HC-MNIST dataset, but following your suggestion, **we performed additional runs** and now highlight **significant improvements** (see **new Tables 3 and 6**).
> >
> >    **Action**. We have conducted additional runs for all the baselines for the HC-MNIST dataset (now they total to 30 runs). Hence, we **updated our Tables 3, 6, and 8**: now the Tables 3 and 6 highlight with color **only significant improvements** (or worsenings) over the baseline. Given the new results, we see that our OAR/dOAR lead to significant improvements of the CR in the **majority of combinations** (= especially, with the DR- and R-learners). This confirms the high effectiveness of our approach for the high-dimensional data.
> >
> >    > Similarly, for the synthetic data example.
> >
> >    Upon reading your comment, we realized that we should clarify the role of the synthetic dataset. The purpose of this experiment was not to report performance results, but rather to _illustrate the core motivation and intuition_ behind our approach (as shown in Figure 1). Therefore, the synthetic data results should be interpreted primarily as a sanity check, confirming the expected behavior in a controlled and low-dimensional setting.
> >
> >    To assess practical performance, we instead focus on the **large-scale benchmark evaluation** (77 ACIC 2016 datasets), where **we report significant improvements** (see Table 2). Therein, we observe **consistent and statistically significant improvements**: dOAR outperforms the baseline CR method in more than half of the 77 datasets, demonstrating the robustness and general effectiveness of our approach.
> >
> >    **Action**: We move the simulated data to **Appendix G** to improve clarity. Instead, now we present the result of the IHDP dataset in the main part of the paper.
> >
> > ### Response to questions
> >
> > - Great question! Our intuition for why the noise regularization and dropout yield different results for the synthetic dataset is related to our Propositions 2 and 3 and the corresponding interpretations of the explicit forms of the regularization. Specifically, given that we apply both the noise regularization and dropout to the middle layer of the two-layer target network, we can view the final layer as a (very approximate) linear model with the high-dimensional input. Here, we can apply our intuition: while the noise regularization penalizes all the weights equally, the dropout adaptively regularizes the weights proportionally to the variance of the input. Considering that the optimal model is zero for the CATE of the synthetic dataset, the non-adaptive variant (= noise-regularization) is arguably a better choice of the regularization, as the adaptivity of the dropout only introduces additional variance.
> >
> > We hope to have addressed all the weaknesses and questions that you mentioned! If there are any remaining concerns, please let us know.

---

### Official Review · Reviewer_uzmE · 2025-10-31

**Soundness:** 2
**Presentation:** 4
**Contribution:** 3
**Rating:** 6
**Confidence:** 4

**Summary:**

The paper proposes and studies methods for regularizing causal meta-learners so that, in a particular sense, the strength of the regularization explicitly adapts to the degree of overlap. More precisely, the regularization is stronger in regions of the covariate space in which there is limited overlap (i.e., the propensity score is close to zero or one). Some existing regularization methods like dropout have been shown to implicitly adapt to overlap in certain settings (Wager, Wang, and Liang, 2013), however the present work appears to be the first paper in the context of causal meta-learners to propose explicitly tying the degree of regularization to the degree of overlap differentially over the covariate space.

The authors consider existing regularization methods namely: dropout, noising of the regressors, and kernel ridge regularization. They propose overlap-adaptive versions of these techniques. In order to regularize adaptively, the degree of overlap, as measured by the conditional variance of the treatment, must be estimated in a first stage. This conditional variance function constitutes a high-dimensional nuisance parameter and the authors suggest a first-order bias correction so that the corrected objective function is Neyman Orthogonal to this new nuisance parameter in addition to the other nuisance parameters.

The authors examine the performance of their methods relative to benchmarks on a range of synthetic and semi-synthetic datasets. The adaptive approach leads to significant improvements on some of these benchmarks and generally depends on a) the degree of regularization, b) which regularization method is used, and c) which meta-learning objective is regularized. The most consistent improvement appears in the context of the HC-MNIST data. At least in my reading, no one particular version of their approach (i.e., dropout vs. noising, choice of base meta-learner, debiased vs. not debiased) seems to dominate the others.

**Strengths:**

I found the paper to be clear and well-explained. I find the proposed methods intuitively very reasonable, and the debiasing of the adaptively regularized objective makes the problem quite non-trivial. The discussion of the interpretation of the regularization methods in linear models seemed to me to be quite helpful. The empirical analysis of the performance of the methods is quite thorough.

**Weaknesses:**

While the method leads to significant improvements over baselines in some cases, the results are somewhat mixed. This is likely inevitable because, as the authors note, adaptively regularizing is only likely to be effective when the CATE is smooth in around regions with limited overlap. Nonetheless, given the mixed results, empirical practitioners may find it rather difficult to choose whether or not to regularize adaptively and additionally, which adaptive method to use. I wonder if there could be some more guidance on this front?

Figure 2 seems to suggest that performance is monotonically improving with the degree of regularization almost uniformly for all methods. The same is true for the results in Table 3. For the synthetic data this is not surprising given that the true CATE is constant (equal to zero) so one would ideally one would regularize as much as possible. In order to make the comparison meaningful, the authors compare methods for a given average regularization strength, but I do not see why this is the correct comparison to make: in practice practitioners would presumably choose the regularization parameters by a data driven method like cross-validation, and may get very different parameters for the different methods. Personally, I think the constant CATE in the simulated data makes it unsuitable for comparing the methods, and for the HC-MNIST data, it might be helpful to see how performance compares when the regularization parameters are chosen empirically or to compare performance under the best choice of regularization parameter for a given method (which would require a comparison for small enough regularization parameters such that performance is no longer monotonically improving.

**Questions:**

More a suggestion than a question, but I wonder whether there may be some way to get the best of both worlds. That is to allow for some compromise between constant and adaptive regularization. For example, suppose we let $\lambda(v):= (4v)^{-\gamma}-\gamma$ where $\gamma\geq0$ is an additional regularization parameter to be selected by the researcher. Then in the limiting case of $\gamma= 0$ we recover constant regularization and with $\gamma=1$ you recover $\lambda_n(v)$.

---

> ### Author Response · Authors · 2025-11-20
> **Rebuttal [1 / 2]**
>
> We are grateful for your detailed and comprehensive review! Thank you for recognizing that our paper is “the first paper in the context of causal meta-learners to propose explicitly tying the degree of regularization to the degree of overlap differentially over the covariate space”. Also, we are happy to hear that you found our work clear and well-explained. In the following, we respond to your concerns and explain how we improved our paper as a result (we highlighted the corresponding changes in the **revised PDF** in **blue color**).
>
> ### Response to weaknesses
>
> - Thank you. Indeed, we found mixed results for different combinations of our OAR/dOAR with meta-learners. For example, based on the synthetic dataset (i.e., Fig. 2), the combination of OAR/dOAR noise regularization with the IVW-learner turned out to be not very effective, while the combination of **OAR/dOAR noise regularization / dropout with DR-learner** appeared to have significant improvement over the baseline for multiple values of the average regularization parameter.
>
>    We found the latter (= **OAR/dOAR noise regularization / dropout with DR-learner**) to be a consistently good combination across **all the benchmarks**. The main reason for this is that OAR/dOAR, in combination with the DR-learner, achieve just the right balance between the high-variability of the pseudo-outcome and the regularization strength (as suggested by the assumptions of Proposition 5). On the other hand, R- and IVW-learners in combination with our OAR/dOAR might over-regularize the low-overlap areas, as the overlap already downscales the error term of the target loss.
>
>    **Action**. We added the discussion above to **Sec. 6** (see our **revised PDF**).
>
> - > Figure 2 seems to suggest that performance is monotonically improving with the degree of regularization, almost uniformly for all methods. The same is true for the results in Table 3. For the synthetic data, this is not surprising given that the true CATE is constant (equal to zero), so one would ideally one would regularize as much as possible.
>
>    The ground-truth CATE for the HC-MNIST dataset is a complex periodic curve, which, however, can be well-approximated by a linear function $2\phi + 2$ (see Appendix F.4 for the details on the data-generating mechanism). Therefore, larger (average) values of the regularization lead to better overall performance. This is true for the HC-MNIST benchmarking dataset, but not necessarily for other settings (e.g., the ACIC dataset collection).
>
>    >  In order to make the comparison meaningful, the authors compare methods for a given average regularization strength, but I do not see why this is the correct comparison to make: in practice practitioners would presumably choose the regularization parameters by a data driven method like cross-validation, and may get very different parameters for the different methods.
>
>    Thank you for asking this question and for giving us the opportunity to clarify our rationale behind why the comparison is meaningful. One _may_ think that a simple approach to a data-driven method to choose hyperparameters for the target model, but this will essentially run into the same problem of low overlap. As noted by [1], there is no silver bullet solution for the CATE model selection, and selection criteria suffer from a so-called congenialty bias (if we would, e.g., use a DR-learner loss as a selection criterion, it would primarily select DR-learners as the best models). This also applies to our low overlap setting: We would not expect some model selection criterion to consistently select “the best” hyperparameter value across all the possible benchmarks. Therefore:
>
>    - In our work, we step back from trying to select a regularization hyperparameter in a data-driven way: We rather suggest working with the imperfect but fixed amount of the average regularization (e.g., even if we over-regularize the target model, the worst we can get is the ATE estimator, which can still be useful for decision making).
>
>   - For the majority of the experiments, we thus compared the methods using the average regularization strength (informally, it guided how much “budget” we can dedicate to a learner to fit the heterogeneity of the CATE around the ATE).
>
>   - We also compared our OAR\dOAR with other methods, where no direct comparison in terms of average regularization strength is possible (see **Tables 7 and 8** in **Appendix G.3**). There, our method proved itself effective in comparison with trimming and balancing.

---

> > ### Author Response · Authors · 2025-11-20
> > **Rebuttal [2 / 2]**
> >
> > > Personally, I think the constant CATE in the simulated data makes it unsuitable for comparing the methods, and for the HC-MNIST data, it might be helpful to see how performance compares when the regularization parameters are chosen empirically or to compare performance under the best choice of regularization parameter for a given method (which would require a comparison for small enough regularization parameters such that performance is no longer monotonically improving.
> >
> >    Thank you. We are happy to discuss (1) the role of the simulated dataset and (2) using your new strategy for performance comparison. We answer both in the following:
> >
> > 1. _Simulated dataset_.  Our synthetic benchmark with the constant CATE can be considered too simple. The main motivation for the synthetic dataset was to offer a motivational example in Fig. 1 and as a sanity check before other, more challenging benchmarks. Upon reading your comment, we realized that the simulated data is not as clear in demonstrating our method as we hoped for, and, hence, we moved it to the appendix.
> >
> >    **Action**: We move the simulated data to **Appendix G** to improve clarity. Instead, now we present the result of the IHDP dataset in the main part of the paper.
> >
> > 2. _Strategy for performance comparison on HC-MNIST_: We followed your suggestion closely and implemented your second idea (= to compare performance under the best choice of regularization parameter). The results can be found in **Table 8 of Appendix G.3**. For transparency, we had the results already in the supplements during the initial revision, but realized that they were too hidden and thus now reference the results more visibly in the main paper. Therein, we compare different existing approaches to handle low overlap, even when the average amounts of the regularization are not comparable (e.g., balancing and our OAR), and our method proved itself effective in comparison with trimming and balancing. Interestingly, the performance of the trimming baseline was also monotonically improving (this can be attributed to a special, approximately linear structure of ground-truth CATE of the HC-MNIST dataset).
> >
> >    **Action**: We now emphasize our performance comparison directly in the main paper, where we point to the results in **Table 8 of Appendix G.3**.
> >
> > ### Response to questions
> >
> > - That is an excellent idea! We already implemented it as a part of the rescaling for OAR/dOAR (see **Appendix E.1**). Therein, we introduced an adaptivity coefficient $\gamma \in [0, 1]$. Here, $\gamma = 1$ leads to a full \OAR, and $\gamma = 0$ is a constant regularization.
> >
> > We hope to have addressed all the weaknesses and questions that you mentioned! If there are any remaining concerns, please let us know.
> >
> > ### References
> > - [1] Curth, Alicia, and Mihaela Van Der Schaar. "In search of insights, not magic bullets: Towards demystification of the model selection dilemma in heterogeneous treatment effect estimation." International conference on machine learning. PMLR, 2023.

---

### Official Review · Reviewer_B5nJ · 2025-10-31

**Soundness:** 3
**Presentation:** 2
**Contribution:** 3
**Rating:** 6
**Confidence:** 3

**Summary:**

This paper proposes overlap-adaptive regularizers for the second stage of meta-learners, gives parametric instantiations plus and RKHS variant, adds a debiased version.

**Strengths:**

- The concept is interesting, addresses a gap in overlapping issue in causal inference.
- Definitions and the family of $\lambda$ functions are precise.
- Provide some solid theory for parametric part. RKHS part is useful but assumes strong overlap.

**Weaknesses:**

- The paper says the noise/dropout (implicit) and explicit forms are “equivalent,” but the exact equivalence is proved only for linear target classes and generally holds as a first-order approximation for NNs. This is stated too broadly.
- Proposition 6 requires that $P(\epsilon < \pi(X) < 1-\epsilon)=1$, which makes the explicit solution neat but excludes the core part the paper aims to help (for regions with very low overlap).
- Several IFs scale like $1/\nu(x)^k$ can blow up when it's approaching 0. Please discuss the integrability/robustness.

**Questions:**

See Weaknesses.

---

> ### Author Response · Authors · 2025-11-20
> **Rebuttal**
>
> We are grateful for your review! We are glad that you found our idea interesting and our theoretical results solid. In the following, we respond to your concerns and explain how we improved our paper as a result (we highlighted the corresponding changes in the **revised PDF** in **blue color**).
>
> ### Response to weaknesses
> - **Equivalence of the explicit and implicit forms**. You are right, we only proved the equivalence of the implicit and explicit forms for the linear models, and, in general, the equivalence is only approximate (e.g., first-order for deep neural networks).
>
>    **Action**. In our **revised PDF**, we have narrowed the statement about the equivalence of the implicit and explicit forms. Also, we noted that Definition 1 can be equivalently formulated for implicit regularizations without the explicit form. By doing so, we can preserve our core idea: making **any regularization** proportional to the degree of overlap.
>
> - **Overlap assumption**. We appreciate this insightful comment. Indeed, we assume,  in Proposition 6, that  $\varepsilon$ can be arbitrarily small (but not zero). However, this version of the overlap is actually a standard estimability assumption in the causal ML literature. For example, both semi-parametric efficiency for ATE [3] and quasi-oracle efficiency of the CATE Neyman-orthogonal learners [1, 2] are in general **only possible with this version of the overlap**.
>
>    In practice, both our theory and empirical results indicate that OAR and dOAR remain effective even for very small $\varepsilon$, i.e., in regions of near–non-overlap (e.g., see left panel in Fig. 1). Hence, we believe our work makes an important contribution to improving CATE estimation in real-world low-overlap settings - for example, in medicine, where treatment assignment often depends on patient risk profiles [4], leading to limited overlap between treatment groups.
>
> - **Scale of the IFs**. True, the debiasing terms might be overly sensitive to low overlap weights (e.g., for the squared multiplicative regularization function). Therefore, in our experiments, we trimmed too large debiasing terms (e.g., when their absolute value was larger than $\alpha = 1$). This trick has helped us to stabilize the training with dOAR and to achieve robustness wrt. too large values of the debiasing terms. We refer to Algorithm 1 for further details.
>
>    **Action**. We added a remark to **Sec. 5.5** that highlights this important implementation detail.
>
> We hope to have addressed all the weaknesses that you mentioned! If there are any remaining concerns, please let us know.
>
> ### References:
> - [1]  Morzywolek, Pawel, Johan Decruyenaere, and Stijn Vansteelandt. "On a general class of orthogonal learners for the estimation of heterogeneous treatment effects." arXiv preprint arXiv:2303.12687 (2023).
> - [2] Aaron Fisher. Inverse-variance weighting for estimation of heterogeneous treatment effects. In International Conference on Machine Learning, 2024.
> - [3] D’Amour, Alexander, et al. "Overlap in observational studies with high-dimensional covariates." Journal of Econometrics 221.2 (2021): 644-654.
> - [4] Westreich, Daniel, and Stephen R. Cole. "Invited commentary: positivity in practice." American journal of epidemiology 171.6 (2010): 674-677.

---

### Official Review · Reviewer_Yknh · 2025-11-01

**Soundness:** 2
**Presentation:** 2
**Contribution:** 2
**Rating:** 6
**Confidence:** 2

**Summary:**

This paper addresses the challenge of conditional average treatment effect (CATE) estimation under low-overlap conditions. The authors propose Overlap-Adaptive Regularization (OAR), which adjusts the regularization strength according to estimated overlap weights. The paper also introduces a debiased version that preserves Neyman-orthogonality.

**Strengths:**

1.	The paper addresses a well-known and practically important issue in CATE estimation—poor performance under low overlap conditions.

2.	Introducing overlap-adaptive regularization is a straightforward and intuitive idea that appears novel in the context of CATE estimation.

3.	The proposed OAR framework can be applied to various existing meta-learners (DR-, R-, and IVW-learners) and accommodates both parametric and non-parametric models.

4.	The inclusion of a debiased version that preserves Neyman-orthogonality enhances theoretical soundness and robustness to nuisance estimation errors.

**Weaknesses:**

1.	The method’s performance is sensitive to the accuracy of estimated overlap weights, which may be unstable in practice.

2.	Theoretical analysis focuses mainly on orthogonality preservation; further insights into convergence or generalization behavior under varying overlap levels could add depth.

3.	Adaptive regularization requires computing overlap-dependent weights for each sample, which may increase overhead for large-scale or high-dimensional settings.

4.	A structural issue is that Section 3 contains material that could be moved to the appendix, while several important technical details and proofs are placed in the appendix instead of the main text, which may hinder readability and comprehension.

**Questions:**

1.	How sensitive is OAR/dOAR to the choice of the regularization function? Are there guidelines for choosing one over the others in practice?


2.	Since the method relies on estimated overlap weights, how robust is the approach to mis-specification or noise in the propensity score? Does the debiased version fully mitigate this sensitivity?

3.	How does per-instance adaptive regularization scale with large sample sizes or high-dimensional covariates, especially in neural network settings?

---

> ### Author Response · Authors · 2025-11-20
> **Rebuttal [1 / 2]**
>
> Thank you for your positive evaluation of our work! We appreciate that you found our method practically important and novel. Below, we elaborate on your comments and how we improved our paper as a result. We highlighted all changes in our **revised PDF** in **blue color**.
>
> ### Response to weaknesses
> 1. Thank you for this question. To the best of our knowledge, this is the case for **every other Neyman-orthogonal learner** targeting at CATE (i.e., DR-/R-/IVW-learners). We are not aware of any Neyman-orthogonal approach that is insensitive to the propensity score estimation errors (or, equivalently, the errors of estimating the overlap weights), given that we use the non-experimental data. The propensity estimation errors appear (i) either as a part of a doubly-robust term (e.g., for the DR-learner) or (ii) as a part of the doubly-robust term and a higher-order term (e.g., for the R-/IVW-learners, our dOAR) [1].
>
> 2. Thank you. While we dedicated extensive effort on deriving the Neyman-orthogonal version of our OAR (=dOAR), we also provide theoretical insights for the DR-learner and dropout regularization for both OAR (=non-orthogonal) and dOAR (=orthogonal) in  **Proposition 5**. Upon reading your comment, we realized that we should give more interpretation of the proposition. Therein, we compared excess prediction risks (= the main notion of generalization quality/convergence speed of meta-learners [2]) of our OAR/dOAR and the constant regularization (CR). Importantly, our **Proposition 5** holds for **any** level of overlap. Furthermore, specifically in the case of **low-overlap values** (= large $1/\nu(x)$), we also show how our OAR/dOAR can be beneficial over the CR. In this case, it is reasonable to assume a _conditional variance assumption_, namely that the conditional variance of the outcome is constant, and, thus, the conditional variance of the pseudo-ouctome is proportional to the inverse overlap (see the full formulation of Proposition 5 in Appendix D).
>
>    **Action**. Upon reading your comment, we realized that we need to provide additional discussion after our **Proposition 5**. Thus, we added a **small remark** in **Sec. 5.3** that explains the implications of **Proposition 5** depending on the level of overlap.
>
> 3. Overlap-dependent weights are functions of a one-dimensional quantity (= propensity score prediction). Therefore, their computation can be done in linear time depending on the mini-batch size (see **lines 7-12** in **Algorithm 1** for further details).
>
>    **Action**: We added a corresponding explanation in the implementation details (see **Appendix E**).
>
> 4. Thank you for the suggestion!
>
>    **Action**. We moved some of the material from Section 3 to the appendix. We also added **high-level explanations of the proofs** to our main paper.  Further, following your and Reviewer EKxX’s comments, we **expanded Sec. 5.4** to include the theory on the non-parametric OAR.

---

> ### Author Response · Authors · 2025-11-20
> **Rebuttal [2 / 2]**
>
> ### Response to questions
>
> 1. Thank you for the question! We have 2 theoretical and 1 experimental justification of why a **multiplicative regularization function** might be the regularization of choice in practice.
>
>    a. (theoretical) In our proof of **Proposition 5** in **Appendix D** (i.e., step 4), we showed that, assuming the _conditional variance assumption_ (the conditional variance of the outcome is constant), our OAR with the weights proportional to the **inverse overlap** yields a lower variance term than the constant regularization. Upon reading your question, we revisited **step 4** of the proof: The exact proportionality is of power $-1/3$ (we updated **step 4** in the proof of **Proposition 5** to now show this explicitly). Note, however, that this result holds only for the linear model and the DR-learner. Still, considering that $x^{-1/3}$ is increasing faster than $-log(x)$ (=logarithmic regularization) but slower than $x^{-1}$ (=multiplicative regularization) for $x \to 0$, the later (**multiplicative regularization**) would be a viable, but more conservative alternative to $x^{-1/3}$. This choice towards more conservative regularization is justified, as for non-linear models, the variance of estimated weights generally grows as we increase the model size (and, thus, we want to regularize at the even stronger rate that is suggested by **Proposition 5**  $x^{-1/3}$ ).
>
>    b. (theoretical) We also provided some intuition on how the **multiplicative regularization** function might be preferable for the non-parametric target models. Specifically, in **Corollary 2** of **Appendix C**, we made a connection between OAR + DR-learner and CR + R-learner for kernel ridge regressions (they are equivalent, given the ground-truth nuisance functions and the same intercept). Given the wide adoption of the R-learner, this Corollary suggests the effectiveness of the **multiplicative regularization**.
>
>    c. (experimental) Considering the numerous experiments from Sec. 6 and Appendix G, it seems that the **multiplicative regularization** is the best choice across different benchmarks.
>
>    **Action**. We added a new discussion to Sec. 6, summarizing why a **multiplicative regularization function** might be the regularization of choice in practice.
>
> 2.  Thank you for the question! Our dOAR is **by construction first-order insensitive** to errors in the propensity score estimation (see **Proposition 4**). This property was further used to derive the remainder terms in **Proposition 5**. Removing the first-order bias is a necessary and often sufficient technique to construct asymptotically optimal meta-learners [2]. This was also confirmed in numerous experiments: the dOAR achieved better performance than the OAR (e.g., see ACIC 2016 results in Table 2 or the HC-MNIST results in Table 3). This hints at the lower sensitivity of our dOUR wrt. the propensity score errors.
>
> 3. Our OAR scales linearly wrt. dataset size and does not depend on the covariates' dimensionality, given that we have already estimated the propensity scores (all the Neyman-orthogonal learners have to estimate the propensity scores anyway).  This is the case, as the overlap-dependent weights are functions of a one-dimensional quantity (= propensity score prediction). Regarding dOAR, we additionally require the evaluation of the gradient wrt. the target model inputs. Yet, this operation also scales linearly wrt. the minibatch size. Therefore, both OAR/dOAR can be evaluated in linear time depending on the minibatch size (see **lines 7-12** in **Algorithm 1** for further details).
>
>    **Action**: We added a corresponding explanation to the implementation details (see **Appendix E**).
>
> We hope to have addressed all the weaknesses and questions that you mentioned! If there are any remaining concerns, please let us know.
>
>
> ### References
> - [1] Morzywolek, Pawel, Johan Decruyenaere, and Stijn Vansteelandt. "On a general class of orthogonal learners for the estimation of heterogeneous treatment effects." arXiv preprint arXiv:2303.12687 (2023).
> - [2] Jin, Jikai, Lester Mackey, and Vasilis Syrgkanis. "It's Hard to Be Normal: The Impact of Noise on Structure-agnostic Estimation." arXiv preprint arXiv:2507.02275 (2025).

---

### Author Response · Authors · 2025-11-20
**Rebuttal**

We are grateful for the reviews of our paper, as well as the constructive and helpful comments!  We addressed all of your comments below. We also uploaded a **revised version of our paper**, where we highlight key changes colored in $\color{blue}{\text{blue}}$).

Here are our main improvements:
- **New theoretical results**. We provided additional results on the excess prediction risk for the non-parametric target models with our OAR RKHS norm (we have added a **new Proposition 7** to **Sec. 5.4**).
- **New experimental results**. We have conducted additional runs for the HC-MNIST dataset and  **updated our Tables 3 and 6** to report significant improvements over the baseline.
- **Extended discussions**. In **Sec. 6**, we added important discussions (i) on how to choose a regularization function for our OAR/dOAR, and (ii) on the best combinations of our OAR/dOAR and different meta-learners.
- **Improved clarity**. We followed multiple suggestions of the reviewers to improve the clarity (e.g., we added high-level explanations of the proofs, a discussion on the computational cost of our method, etc.)

We are confident that the feedback has greatly improved the quality of our paper, and we believe that the paper is a good fit for the ICLR 2026 conference.

---

### Meta-Review · Area_Chair_JNAB · 2026-01-07

**Summary:**

This paper studies the problem of estimating conditional average treatment effects (CATE) in low-overlap settings. It proposes an Overlap-Adaptive Regularization (OAR) approach that tailors the regularization strength based on estimated overlap weights, along with a debiased variant designed to preserve Neyman orthogonality. The reviewers view the problem as novel and compelling, and consider the work a meaningful contribution to the causal inference literature. The reviewers raise concerns about the limited and sometimes overly broad theoretical analysis, particularly its focus on linear models, orthogonality preservation, most of which has been well addressed in the rebuttal. I recommend acceptance of this paper as a poster paper.

**Reviewer Concerns:**

I think most of the concerns are addressed by the rebuttal. The only exception is the concern of Reviewer Yknh on the reliance on accurate propensity score or overlap weight estimation. In my opinion, it is a common issue in the causal effect estimation community and orthogonal to the main contribution of this work.

**Reviewer Scores:**

For Reviewer Yknh, B5nJ, and uzmE, I think the rebuttal could properly answer their question about this work, and they would keep the high score. For Reviewer EKxX, I think the concerns are completely addressed, with extra experiments and discussion added in the revised paper. I think Reviewer EKxX would have changed the score to 6.

---

### Decision · Program_Chairs · 2026-01-26

Accept (Poster)